# Mass balance and hydrological modeling of the Hardangerjøkulen ice cap in south-central Norway

Trude Eidhammer[1], Adam Booth[2], Sven Decker[3], Lu Li[4], Michael Barlage[1,5], David Gochis[1], Roy Rasmussen[1], Kjetil Melvold[6], Atle Nesje[7], Stefan Sobolowski[4]

[1]National Center for Atmospheric Research, P.O. Box 3000, Boulder, CO 80307, USA
[2]School of Earth and Environment, University of Leeds, UK
[3]Department of Geoscience, University of Oslo, Norway
[4]NORCE Norwegian Research Centre, Bjerknes Centre for Climate Research, Bergen, Norway
[5]Now at: National Center for Environmental Prediction, NOAA, College Park, MD, USA
[6]Norwegian Water Resource and Energy Directorate, Oslo, Norway
[7]Department of Earth Science, University of Bergen, Norway

*Correspondence to*: Trude Eidhammer (trude@ucar.edu)

**Abstract.**

A detailed, physically based, one dimensional column snowpack model (Crocus) has been incorporated into the hydrological model WRF-Hydro to allow for direct surface mass balance simulation of glaciers and subsequent modeling of meltwater discharge from glaciers. The new system (WRF-Hydro/Glacier) system is only activated over a priori designated glacier areas. This glacier area is initialized with observed glacier thickness and assumed to be pure ice (with corresponding ice density). This allows for melt of the glacier to continue after all accumulated snow has melted. Furthermore, the simulation of surface albedo over the glacier is more realistic as surface albedo is represented by snow where there is accumulated snow, and glacier ice when all accumulated snow is melted. To evaluate the WRF-Hydro/Glacier system over a glacier in Southern Norway, WRF atmospheric model simulations were downscaled to 1 km grid spacing. This provided meteorological forcing data to the WRF-Hydro/Glacier system at 100 m grid spacing for surface and streamflow simulation. Evaluation of the WRF downscaling showed a well comparison with in situ meteorological observations for most of the simulation period. The WRF-Hydro/Glacier system reproduced the glacier surface winter/summer and net mass balance, snow depth, surface albedo and glacier runoff well compared to observations. The improved estimation of albedo has an appreciable impact on the discharge from the glacier during frequent precipitation periods. We have shown that the integrated snow pack system allows for improved glacier surface mass balance studies as well as hydrological studies.

## 1. Introduction

Glaciers provide natural storage of water supply to rivers. In Norway, these rivers can then contribute water for domestic and industrial consumption, irrigation and hydropower (Sorg et al., 2012; Laghari, 2013, Kaser et al., 2010). Glaciers are among the first indicators of climate variability and change, and thus glacier retreat and changes and associated streamflow effects

can impact human water supplies depending on the modifications to glacier melt timing and amount (Immerzeel et al., 2013; Bolch et al., 2012). It is imperative to understand how glaciers and associated hydrological processes respond to a changing climate to better inform communities that rely on glaciers for their livelihoods and well-being.

It is the surface mass balance on glaciers that impacts the subsequent glacier-fed streamflow. Mass balance changes on glaciers in Norway are largely controlled by accumulation-season precipitation and ablation-season temperature. This was determined by comparing measured glacier mass balance from stake measurements with meteorological station data (winter precipitation and summer temperature (e.g. Andreassen and Winsvold, 2012)). However, elevation gradients and complex topography in many glaciated regions lead to large variations in temperature, precipitation and winds (and thereby transport and deposition of dry snow during the accumulation season) and net radiative exchange across the glacier (e.g Ayala et al., 2015; Liston and Sturm, 1998). Therefore, the proper simulation of the non-homogenous, non-stationary evolution of a glacier requires atmospheric processes at much finer resolution than typical global or regional climate models can provide (Aas et al., 2016).

Obtaining distributed meteorological forcing data of temperature, precipitation and wind is complicated by the spatial and temporal scarcity of observations, topographic complexity, and by the coarseness of the atmospheric models used for downscaling. As a result, there are major gaps in our knowledge regarding the behavior of glaciers at local to regional scales and the processes that control their variability (Immerzeel et al., 2010; Bolch et al., 2012). Several studies have used dynamical downscaling of regional climate models, on the order of 10-18 km (e.g. Machguth et al., 2009; Kotlarski et al., 2010a; Van Pelt et al., 2012). However, they still do not resolve many sub-grid processes and the studies have therefore required statistical corrections to the downscaling as well. Recently, several studies applied much higher resolution (0.5-2.5 km) regional climate models to provide the heterogeneous forcing required over glaciers (e.g. Collier et al., 2013; Collier et al. 2015; Aas et al., 2016; Molg and Kaser 2011; Bonekamp et al., 2019; Vionnet et al., 2019). Indeed, Lundquist et al. (2020) argue that in many instances in mountain terrain, high-resolution atmospheric models produce better estimates of annual precipitation than the information we can gain from observation networks. Physical based downscaling based on the linear model by Smith and Barstad (2004) has also been applied over complex terrain for glacier studies (e.g. Jarosch et al., 2012). Glacier mass balance parameterizations have been implemented in atmospheric models such as the regional climate model (REMO, Kotlarski et al. 2010b) and a climate mass balance model with feedback to the atmosphere was implemented into WRF by Collier et al. (2013).

The regional 'atmosphere-only' models typically do not have detailed information about runoff routing processes, which are important components in the glacier hydrological cycle, although these models can provide input to detailed offline snowpack and hydrological models. Glacier melt contributes to discharge, especially during summer when the magnitude of the summer peak river flow often depends on the contribution of melt water from snow and ice to the total river flow. This contribution from glaciers to total flow plays a key role in the glacier-fed rivers in populated regions in Norway in which summer flows are crucial for irrigation, human consumption and energy production. In the studies here, we will use the detailed hydrological model - the Weather and Research Forecasting - Hydro (WRF-Hydro) modelling system (Gochis et al., 2015; Senatore et al.,

2015; Arnault et al., 2018; Fersch et al., 2019; Rummler et al., 2019) for streamflow modeling. However, WRF-Hydro does not explicitly include glaciers within the model system, which results in a large uncertainty and underestimation of discharge during the melt season in glacial fed areas like the Himalaya (Li et al., 2017), and parts of the Andes, Scandinavia and North America. The Noah multi-parameterization (Noah-MP, Niu et al., 2011) land surface model (LSM) is often used in the existing WRF-Hydro system, which includes only a glacier land surface category. When snow is accumulated, Noah-MP uses a three-layer snow model to represent the evolution of the snow pack. However, when the seasonal accumulated snow melts off in the summer, the underlying surface, for albedo purposes, is assumed to be old snow (snow packed glacier), while not allowing for areas of bare ice and melting ice. Furthermore, the glacier is also represented in the soil layer with a two-meter layer of ice/water at the bottom of the column. This layer can melt, and refreeze, but this layer does not provide runoff to WRF-Hydro.

By linking a surface mass balance glacier model to the WRF-Hydro system that interacts with the underlying land-surface/hydrological components, the coupled interactions between the energy, water and mass balance budgets over glaciated river basins can be better depicted for projecting future impacts. For this purpose, we chose the Crocus snow model as our starting point to build the system WRF-Hydro/Glacier. Crocus is a one-dimensional, column energy and mass model of snow and ice cover, that uses meteorological conditions as input data and was initially developed for operational avalanche forecasting and simulation of Alpine snow (Brun et al., 1989, 1992). It is a physically-based model, in which the snow depth can be divided into a user defined maximum levels and where the default maximum is 50 layers. A principal strength of the model is the detailed description of the metamorphism process for different types of snow, which allows for a more accurate calculation of snow surface albedo. The Crocus model was first used for glacier mass balance studies by Gerbaux et al (2005) and recently used for glacier surface mass balance studies within the French Surfex model by Reveillet et al. (2018), Revuelto et al. (2018) and Vionnet et al. (2019).

Norway is home to some of the best-observed glaciers in the world. Its National Water Resource and Energy Directorate (NVE), regularly monitors and assesses the mass balance and length changes of Norwegian glaciers (Andreassen et al., 2020). This paper will focus on the Hardangerjøkulen ice cap, located in south-central Norway. Hardangerjøkulen is the sixth largest glacier on the mainland of Norway, and is located at the main water divide between eastern and western Norway. The glacier covers an area of approximately 71 km² and the highest point on the glacier is 1,863 m a.s.l. (Andreassen and Winsvold, 2012). Hardangerjøkulen is a plateau glacier and has several outlet glaciers, of which Blåisen (not shown) and Midtdalsbreen facing east/northeast and Rembesdalskåka in the west are the best known (Figure 1). The ice cap has a volume of about 10.64 km³, and the mean ice thickness is about 150 meters with maximum ice thickness of more than 380 m (Melvold et al., 2011).

In this paper we present the WRF-Hydro/Glacier system in which Crocus is coupled to the WRF Hydro model, with the Crocus model representing the glacier. In section 2 we explain the Crocus implementation while in section 3 we describe the

experimental design and in section 4 observational data are discussed. The results are presented in section 5 and finally, conclusions and future work are included in section 6.

## 2.    Crocus Implementation

### *2.1.  Crocus Description and Original WRF/Hydro Glacier Treatment*

One of the main reasons to use Crocus for glacier mass balance modeling and the subsequent streamflow modeling is its use of physical parameterization and ease of implementing non-flowing glacier (ice) layers. By comparison, the Noah-MP has only a maximum of three-layers in its snow pack, depending on the total snow depth (Niu et al., 2011). In Noah-MP, the snow albedo option used in this study is calculated based on the snow age through an empirical function.  Noah-MP has its own glacier module with the effect that when there is snow, the Noah MP snow module is active and albedo is represented by the

Noah-MP 3-layer snow model, and the minimum albedo of snow is set to 0.55. However, when all snow has melted, the surface is represented by the bare glacier land surface category which has an albedo of 0.675 (0.8 in the visible and 0.55 in the near infrared spectral bands) and roughness lengths and heat conductivity typical for glaciers with old snow. Furthermore, this exposed glacier ice cannot melt as the glacier is only a land surface category (though the glacier is represented in the soil layer with a two-meter layer of water/ice but does not provide runoff to WRF-Hydro). Finally, as a result of these limitations in the

Noah-MP glacier formulation, the glacier cannot decrease in mass and extent.

Crocus is an energy and mass-transfer snowpack model, initially developed for avalanche forecasting (Brun et al., 1989; 1992). In this work we use the version that was implemented into the French Surfex model V8.0 (Vionnet et al., 2012). This version has several updates from older versions of Crocus, such as impacts of wind-drift.

The Crocus snowpack model is a multilayered physically-based snow model that explicitly calculates snow grain properties

in each snow layer and how these properties change over time. The grain properties of dendricity, sphericity and size are prognosed in Crocus through metamorphism, compaction and impacts of wind drift. Furthermore, the snow albedo is calculated based on the snow grain properties from the top 3 cm of the snowpack (Vionnet et al., 2012) and is calculated in three spectral bands (0.3-0.8, 0.8-1.5 and 1.5-2.5 µm). Impurities in aging snow is parameterized in the UV and visible spectral band (0.3-0.8 µm) from the age of the snow with a time constant of 60 days. See Vionnet et al., (2012) for detailed description of the

albedo calculations. The albedo over ice is constant in all spectral bands and are 0.38, 0.23 and 0.08 for the spectral bands 0.3-0.8, 0.8-1.5 and 1.5-2.5 µm. The sensible and latent heat are parameterized with an effective roughness length over snow and ice (see Vionnet et al. (2012) for further details). Here we use 1 mm over snow and 100 mm over ice.

In the Crocus model, it is possible to divide the snow into a user defined maximum numbers of dynamically evolving layers. As new snow is accumulated, a new active layer is added. As different snow layers become similar (based upon the number

of user-set layers, the thickness of the snow layers and the snow grain characteristics), these snow layers will merge into single snow layers.

The Crocus module is added to the Noah MP land surface model inn WRF-Hydro to act as a glacier mass balance model. Over designated glacier grid points, the Crocus snow model represents both snow and ice, while outside of the designated glacier grid points, the regular three-layer snow model in Noah-MP is used. Since the current Crocus implementation in WRF-Hydro only acts over designated glacier grid points, we follow Gerbaux et al. (2005), and assume that the temperature at the bottom of the glacier and the ground below are both at 0 ºC. Note that we have no yet incorporated fluxes between the glacier and the ground below, thus there is a constant-temperature boundary conditions.

Both Crocus and Noah-MP (for the non-glacier grid points) output runoff from snowmelt (and precipitation). This runoff is provided to the terrain routing models in the hydrological model system WRF-Hydro. WRF-Hydro is a model-coupling framework designed to link multiscale process models of the atmosphere and terrestrial hydrology (Gochis et al., 2015; Yucel et al., 2015). In coupled mode it includes the full functionality of the atmospheric Weather Research and Forecasting (WRF) modelling system. WRF-Hydro enables simulation of land surface hydrology and energy states and fluxes at high spatial resolutions (typically 1 km or less) using a variety of physics-based and conceptual approaches (Yucel et al., 2015; Senatore et al., 2015). It contains horizontal routing processes and water management modules and is linked to the Noah-MP land surface module (among others). The added capability of running Crocus as a glacier mass balance module in WRF-Hydro is called the "WRF-Hydro/Glacier" system from here onward.

Note that our implementation of Crocus as glacier mass balance model does not address glacier movement (i.e. plastic flow) nor lateral wind (re)distribution of snow. Being a relatively flat dome glacier, the Hardangerjøkulen glacier that we focus on here does not move much in a year, and therefore the lack of dynamical movement of the glacier is not expected to have a major impact on the results in this paper, as we only consider four simulation years. On the other hand, the lateral snowdrift and wind-driven redistribution of snow on Hardangerjøkulen can be significant, and our results are likely impacted by the lack of this physical process in the model. It is worth mentioning that including lateral movement of snow due to snowdrift in the model system is not a trivial task, and is therefore not currently included. However, there are two options to include impacts on the snow due to wind. One of the options impacts the snow density during blowing snow events (Brun et al., 1997). This option is important in polar environment (Brun et al., 1997), and we found it necessary in our simulations as well. The other option is the sublimation due to snow drift, which was implemented by Vionnet et al. (2012) and which is in the Crocus version that is used in this study. This option was also crucial to include in order to accurately simulate the glacier mass balance over Hardangerjøkulen. It is especially important for reproduction of the observed heterogeneous snow distribution.

## 2.2. Initialization of glacier module in WRF-Hydro/Glacier

To run the WRF-Hydro/Glacier system, the glacier to be evaluated must be initialized with its thickness and extent. Here we focus on the Hardangerjøkulen; its extent and height were obtained from the NVE (Melvold et al., 2011). Figure 1 shows the glacier thickness and extent of Hardangerjøkulen at 100 m grid spacing (for which the entire WRF-Hydro/Glacier model is run). At initialization, it is assumed that the glacier consists of only ice, and the density is that of pure ice (900 kg m$^{-3}$). In the simulations presented here, the user-defined maximum layers are set to 40 layers, and the glacier is initialized with all the layers having the same assumed density and snow grain properties. As new snow accumulates during the simulations, the layers representing the glacier will start to merge since all 40 layers are occupied with the initialized ice. Here, the thickest parts of the glacier merged to an average of 8 layers after 5 months of simulation and remained fairly constant for the remainder of the simulation. Revuelto et al. (2018) also used Crocus for surface mass balance studies. They initialized their glacier with the same glacier thickness (40 m) over the entire glacier and in the six lowest layers with the thickness progressively increasing with depth. In contrast to this study, they reinitialized the glacier to 40 m every new season (August 1) so that the glacier would not decrease in extent, while here the glacier is only initialized once at the beginning of the simulation with the observed glacier thickness.

As implemented, if the glacier completely melts over a user defined glacier grid point, the original Noah-MP module is used from this point on. Therefore, as currently implemented, the glacier cannot grow horizontally in extent, it can only decrease in extent, as no dynamic response of the ice mass is included in the model. Over short model time periods, the lack of increase in glacier extent might impact a few grid points at the edges of the glacier. However, given the expected increase in temperature in the future we do not expect that limiting glacier horizontal growth will have a major impact over most studied glaciers as most are likely to decrease in mass and extent.

## 3. Experimental Description

For meteorological input data to the WRF-Hydro/Glacier system, dynamically downscaled data from the WRF model version 3.9 (Skamarock et al., 2008) was created over Southern Norway. WRF was run with an outer domain (Domain1) with a 3 km grid spacing and an inner domain (Domain 2) with a 1 km grid spacing (see Figure 2, top) with 51 stretched vertical levels (lowest prognostic level is 25 m). The ECMWF Re-Analysis Interim (ERA-I) dataset was used for input and boundary conditions and the model was run from August 1, 2014 to January 1, 2019. The microphysics scheme used was the Thompson-Eidhammer aerosol aware scheme (Thompson and Eidhammer, 2014), the Yonsei University (YSU, Hong et al., 2006) scheme

for the boundary layer (Hong et al., 2006), the rapid radiative transfer model (RRTMG) for longwave and shortwave radiation calculations (Iacono et al., 2008), and the Noah-MP land surface model (Niu et al., 2011). See Table 1 for the configuration.

We did not use any lake models, thus, in WRF, the skin temperature of lakes is typically set to the same temperature as the nearest grid point that is defined as sea surface. With this setting, the lakes will not reach freezing temperatures since the oceans surrounding southern Norway typically do not freeze in the winter. To rectify this problem, we assign a 10-day moving average skin temperature from the associated ERA-Interim grid point onto the lake grid points to allow a representation of freezing lakes. The fjords still use the sea surface temperature from ERA-Interim. We acknowledge that a large step from ~75 km (ERA-I) to 3km in WRF is of concern. However, we follow findings by Liu et al., (2016) where they state: "Tests showed that one-way nesting WRF, at 4-km grid spacing, with the ~75 km reanalysis was an adequate configuration without the need for a coarse grid that intermediates the ERA-Interim data and the WRF domain." What is important is that the area of interest must be sufficient large enough for mesoscale spin up. Our domain of interest (Domain 2) is slightly closer to the boundary than what is in Liu et al., (2016). However, as shown below, the model results are quite reasonable, thus we believe that the jump from ~75 km to 3 km is adequate.

The 1 km WRF (inner-nest) simulation results are used as input to run WRF-Hydro/Glacier. The WRF-Hydro/Glacier domain has 100 m grid spacing (Figure 2, bottom) and covers a smaller area compared to that of the 1 km domain. The precipitation, 2 m temperature, 10 m wind speed, 2 m water mixing ratio, surface pressure, and long and shortwave radiation outputs from the WRF 1 km simulations were bi-linearly re-gridded to the 100 m grid spacing in the high-resolution domain. We note that we did not account for variability in terrain in the re-gridding process. Thus the atmospheric forcing is still "smooth" as regards to a 100 m grid. However, the region of interest (Hardangerjøkulen and surrounding terrain) is an open, mostly flat area and we therefore believe that for this specific case, disregarding the variation in terrain does not have much impact on mass balance calculations. For partition of rain/snow from the input precipitation, WRF-Hydro/Glacier use the rain/snow partition from Jordan (1991). The streamflow routing is run at the same resolution as Crocus (i.e. at 100 m). Finally, no calibrations were applied to the routing model.

## 4. Glacier observations

Hardangerjøkulen is a well observed glacier, with several decades of mass balance observations, and several field campaigns. In the following, data from field observations, the ongoing mass balance observations and remote sensing are used to evaluate the WRF-Hydro/Glacier system.

### 4.1. Glacier mass balance

Glacier mass balance is the amount of mass a glacier gains or loses over a year (sum of accumulation and ablation). NVE has gathered winter, summer and annual (net) mass balance (winter + summer mass balance (where summer mass balance is

negative)) observations over Rembesdalskåka since 1963 (e.g. Andreassen et al., 2020) where Rembesdalskåka is the west/south-west glacier outlet of Hardangerjøkulen (see Figure 1). The observations are gathered at several locations on the glacier, from the lower to the upper parts (1066-1854 m a.s.l.) by using stakes (aluminum poles inserted into the glacier) and in spring by probing (using thin metal rods to measure snow depth) to the previous year's summer surface. Winter mass balance is found from the observed snow depth (by stakes and rod soundings at approximately 60 locations (see Figure 1)) and from snow density measurements at one location on the glacier. To determine summer balance (at 4 locations), the observations are usually conducted at the end of the main melt season around the time period September - October, while the winter balance is usually determined at the end of the accumulation season May - June. Often summer balance observations are conducted when new snow has accumulated on the upper part of the glacier. However, the summer surface can be identified in shallow snow pits. Therefore, to determine mass balance from the WRF-Hydro/Glacier simulations, we use the date with the smallest simulated glacier mass to determine the end of the summer and start of the winter season instead of using the actual date for when observations were gathered. Details about the annual mass balance observations are found in NVEs report series Glaciological investigations in Norway (Kjøllmoen et al., 2016, 2017, 2018 and 2019).

### 4.2. Radar-derived snow thickness

Variations in snow accumulation were measured over Hardangerjøkulen in April 2017 and 2018, using ground-penetrating radar (GPR). In 2017, surveys were conducted with a MALA Geosciences GPR system; in 2018, this system was not available hence a Sensors & Software pulseEKKO PRO model was used instead. However, data from these two systems are directly compatible since both were acquired with antennas of 200 MHz center-frequency. The GPR systems were towed behind a snowmobile at ~15-20 km/h. The interval between successive GPR recordings is ~0.2 s, giving a distance sampling interval of ~ 1 m (regularized to exactly 1 m in processing). A GPS system was also mounted on the snowmobile, recording positions every 1-2 s, to locate the GPR recordings. The positional accuracy of the GPS is ~ 5 m. A total of 116 km and 27.4 km of measurements were acquired in 2017 and 2018, respectively, and both acquisitions featured numerous crossing-points such that the internal consistency of accumulation estimates could be ensured.

GPR systems record the travel-time of a radar pulse through the ground, therefore estimates of winter accumulation requires some measure of the GPR propagation velocity to covert time to depth. This was obtained using so-called common midpoint (CMP) data (e.g., Booth et al., 2011, 2013), in which GPR responses suggested an average velocity of $0.218 \pm 0.001$ m/ns for the upper ~ 2.8 m of the snowpack. With no other velocity control available, this value is applied to convert all GPR travel-time estimates to a snow depth.

The base of winter snow accumulation was taken to be the first prominent reflective horizon within the GPR record. This is straightforward in the Hardangerjøkulen ablation zone, typically at elevations $\prec$ 1600 m, where winter snow directly overlies the glacier surface, typically at a depth of 2-3 m. Here, the only significant reflection is from the glacier surface itself.

Consequently, depth errors are expected to be less than ± 0.1 m. However, areas of firn accumulation have a more complex pattern of reflectivity and it is not always possible to guarantee accurate snow thicknesses, and errors may here be up to ± 1 m. However, given the crossing points in the GPR records, depth estimates are at least internally-consistent and errors are expected to vary systematically across the entire record.

### 4.3. MODIS snow albedo

The Crocus model computation of snow albedo depends on the physical properties of the snow grains, while the formulation used by Noah-MP model uses only a time dependent empirical formulation. To evaluate the modeled snow albedo, we use the NASA Moderate Resolution Imaging Spectrometer (MODIS) daily snow albedo product version 6 from Aqua (MYD10A1, Hall and Riggs, 2016a) and Terra (MOD10A1, Hall and Riggs, 2016b). These products are reported with 500 m grid spacing.

### 4.4. Streamflow

Discharge measurements were obtained at two rivers. One river (here named "Middalselvi") is fed by meltwater from the Midtdalsbreen (a glacier arm of Hardangerjøkulen) where the catchment is about 12 $km^2$ and 60% glacierized (see Figure 2). The other river is Finseelvi, where the catchment is about 16 $km^2$ and 14.7% glacierized, and not much impacted by glacier melt. Two Hobo Water Level loggers were installed in each catchment in Fall of 2016 and we have data until November 2018.

## 5. Results

### 5.1. WRF standalone verification

The WRF model simulations were validated by using observations from 21 Automated Weather Stations (AWS) operated by the Norwegian Meteorological Institute (Figure 3). These data were compared to the output of the 1 km simulations (Domain 2). The locations of the stations are given in Figure 3 (top left panel), along with the location of Hardangerjøkulen. We note that there exist additional stations in the south-west corner of the domain that were excluded in our evaluation because they were too close to the border of the domain. Figure 3 shows the total precipitation for the mass balance years 2015, 2016, 2017 and 2018 with observations given as colored circles. Here we define a mass balance year from October 1 in the previous year through September 30 of the current year. For example, the mass balance year 2015 ranges from October 1, 2014 to September 30, 2015. As can be seen in Figure 3, the model captures the spatial precipitation distribution of the observed precipitation with maximum precipitation near the coast and minimum to the lee of the mountains. The locations of some of these observations are in or near narrow fjords, such as the Ullensvang, Eidfjord, Skulafossen, Kvamsøy and Øystese stations. These locations tend to underpredict precipitation by nearly 20% (see Figure 4), which is a larger bias than stations further away from the fjords. The values shown in Figure 4 are obtained from finding the closest grid point to the actual location, then from there take the 4 closest model grid points relative to the selected grid point closest to actual AWS elevation. Furthermore, stations

that are located in the model over 100 m above the actual elevation are not included. Three of the stations (Finse, Midtstova and Geilo) are located at high-altitude exposed locations. At these stations there is a large under-catch of observed snow in the wintertime when the snow can blow past the precipitation gauges instead of falling into the gauges (Rasmussen et al., 2012). The stronger the wind, the larger the under-catch. The data obtained from the Norwegian Meteorology Institute for these

stations have not been corrected for any under-catch. Figure 5 show the effect of under-catch of snow at Finse, which is the station that is located closest to Hardangerjøkulen, about 4 km north-northeast of the edge of Midtdalsbreen and about 11 km from Rembesdalskåka. In Figure 5, the accumulated precipitation and temperature for the 2016 summer and winter season is shown at both Finse, Fet (a station about 25 km south west of Finse, but about 14 km away from Rembesdalskåka) and Evanger (a non-exposed inland station with little snow). Similar results are seen for 2015, 2017 and 2018 (not shown). As can be seen,

WRF compares well with observations at Evanger and Fet almost the entire period (Figure 5a-5d). WRF precipitation also compares well with observations at Finse in the summer season (Figure 5f), but has much more precipitation than the observations in the winter season (Figure 5e) where wind speeds are often more than 5 m s$^{-1}$ (Figure 5i) and temperatures are below freezing. Furthermore, WRF does simulate the storm sequences well as seen in Figure 5g and 5h. Thus we attribute the low observed precipitation compared to WRF during the winter season at Finse to the under-catch problem as precipitation

modeled at Fet compares great with observations. Note that during September to 17 to September 26, the Finse station did not provide any data (Figures 5f, h and j). However, during this time period, WRF did not predict any precipitation, and Fet did not observe any precipitation. Thus, the cumulative precipitation shown in Figure 5f is still valid.

The World Meteorology Organization (WMO) Solid Precipitation Intercomparison Experiment (SPICE) was set up to evaluate the under-catch of snow and develop transfer functions to correct for the under-catch of solid precipitation.  (Smith et al.,

2020). One location for these studies is Haukeliseter, which is about 20 km from Røldal (see Figure 2) and is within the 1 km domain. In these studies, several different precipitation gauges and wind shield combinations were used. The Double Fence Automated Reference (DFAR) was deployed as the reference and is used as the "truth" precipitation. We compared the WRF model results with the DFAR data (Smith et al. 2019), and WRF is predicting more precipitation compared to these observations, with a bias typically at ~30% (not show). About 10% of this bias could potentially be attributed to

underestimation with the DFAR (Rasmussen et al. 2012). The bias in WRF is opposite and higher compared to what is found at locations with little impact of snow. In regards to transfer functions (correcting for under catch in observations), Smith et al (2020) stated this is their study: "Although the application of transfer functions is necessary to mitigate wind bias in solid precipitation measurements, especially at windy sites and for unshielded gauges, the inconsistency in the performance metrics among sites suggests that the functions be applied with caution."  We are therefore not adjusting the observed observations on

Finse for our evaluation, and rather stress the well comparison between model and observations at Fet and summer season precipitation on at Finse.

We conclude, as shown above, most of the non-exposed inland stations are relatively well simulated by the WRF-model and the seasonal cycle of precipitation is captured. We note, however, one time period where WRF is underpredicting the precipitation relative to locations near Finse (Hardangerjøkulen). Several stations near Finse do not catch a precipitation period in the middle of January 2017, which has an impact on Finse as well (this underprediction in WRF is difficult to directly evaluate at Finse, but the two stations near Finse (Fet and Eidfjord) clearly have an underprediction at this time period (not shown). We note that the observed storm sequences were captured in the simulation, wind direction was well simulated as well as the wind speed during this precipitation event (not shown), just not the precipitation amount. The effect of this precipitation time period will be discussed in relation to the mass balance and streamflow results is Section 5.

Figure 6 shows a scatterplot of observed and simulated 2 m temperature at the Finse AWS. As can be seen, the modeled temperature compares well with the observed, but with a small negative bias in the winter. At the very low observed temperatures ( < -15ºC), WRF often has a positive bias. This is likely due to WRF not capturing the strong inversions that often occurs in the winter months (Mölders and Kramm, 2010; Hines et al., 2011). Figure 5e also shows this positive bias at the very low temperatures. However, in general, WRF compares well with observations with a correlation coefficient near 0.9.

Figure 7 shows the simulated and observed 10 m wind speed and direction for the entire simulation period at Finse. Although the wind direction is not an input variable in the land surface model in WRF-Hydro/Glacier, the wind direction can dictate precipitation amount and type, thus wind direction is important for obtaining correct simulation of precipitation and subsequent glacier mass balance as shown in Bhatt et al (2020)[1]. As can be seen, the simulated wind direction compares well with the observed. Mean bias is -0.13 m s$^{-1}$ and coefficient of correlation is 0.8. Overall, the major simulated wind directions and speed are simulated quite well (see also Figure 5 for wind speed).

### 5.2. Glacier mass balance and snow height

Figure 8 shows the observed glacier mass balance, new accumulated snow thickness and density from NVE versus modeled for the 2015-2018 mass balance years for Rembesdalskåka. The observed winter balance is taken from the green locations shown in Figure 1 while the summer balance is found at the 4 red locations in Figure 1. The observed mass balance is plotted as averages in intervals of 50 m. For the summer balance all values between the 4 locations are interpolated. The modeled winter (and summer) balance is plotted as averages of all grid points over Rembesdalskåka within intervals of 30 m. As can be seen, the modeled mass balance (winter, summer and net, Figure 8a, d, g, and j) is generally comparable with the observations. The observed winter mass balance shows a small decrease at the top of the glacier, with a slight increase about 200 meters below the top (see left panels). This is most likely due to redistribution of snow that is common at Finse and its surroundings, as strong winds occur often there. Since lateral redistribution of snow is not included in the Crocus and Noah-

---

[1] * Note, text citing the Bhatt paper will be removed if that paper is not accepted by time (the Bhatt paper is listed as "to be submitted" in reference list)

MP models, the modeled mass balance increases more linearly towards the top compared to the observations. The winter mass balance is about 29 % (20 %) underestimated in 2017 by Crocus (Noah-MP), which is likely due to the underprediction of winter precipitation at Finse and nearby stations in this year (discussed in section 3.1). For the three other years (2015, 2016 and 2018), the modeled winter balance is within 9.3, 13.8 and 5.1 % (8.7, 14.7 and 2 %) of the observed winter mass balance for Crocus (Noah-MP) respectively (see Figure 9). The reason for the slightly larger bias for Crocus is not known at this point. However, note that Crocus matches better with the observations at lower levels below about 1500-1600 m in 2017 and 2018 (see Figure 8 left and middle panels). This does not show up in the total mass balance comparisons since most of the mass is above 1600 m due to higher snow thickness and larger area (see Rembesdalskåka, Figure 1). Another point to add here is the improved simulation of winter mass balance of Rembesdalskåka compared to Engelhardt et al (2012). They used gridded interpolated precipitaion data from seNorge (http://senorge.no) and obtained a mean negative bias of 28% from 19 modeled winter mass balance years for Rembedalskåka, while we use high-resolution regional scale modeling to obtain horizontally distributed precipitation.

For the summer and net mass balance we only consider Crocus since Noah-MP cannot melt more than the accumulated snow amount from the model start (see Figure 8, left panels). As the winter balance, the modeled summer mass balance is also in general agreement with the observations with a bias of 6.67%, -1.5%, -7.4 % and 5.9 % in 2015, 2016, 2017 and 2018 respectively (Figure 9). However, the modeled 2018 summer balance curve in Figure 8j shows a different function of height compared to the observations with stakes. This year the summer balance observations by NVE was taken late in the fall (November 22), while the actual minimum was most likely in September based upon our model results. At this point, only the two top stakes that are used to determine mass balance were left standing and both were located above 1750 m, while the other sakes had melted out (Kjøllmoen et al., 2019). Our model results compare favorably at the top altitudes where observations were available. Below ~1750 m, there are no observations, and observed summer mass balance is instead estimated from observed temperatures at nearby stations, and the relationship between temperature and melting for the period 2012-2017 (Andreassen et al., 2020). Therefore, there is a possibility that our model results are closer in agreement to the actual melting than indicated in Figure 8j.

The modeled distribution of snow thickness and snow water equivalent (SWE) over Rembesdalskåka is comparable to measurements by NVE (Figure 10). The NVE data are the same as used in Figure 8, while not averaged in elevation transects as in Figure 8. There is more heterogeneity in the observations, as would be expected since the models does not account for lateral redistribution of snow. Despite this, Crocus does account for sublimation of snow drift, which results in more heterogeneity in the spatial snow distribution over the glacier compared to Noah-MP. The relatively large underestimation of modeled SWE (i.e. the winter balance) in 2017 is also seen in the snow thickness. This is also evident in Figure 8, middle panel, which shows the snow-depth as a function of altitude. The simulated snow-depth for 2015, 2016 and 2018 compare better with observations at altitudes above 1600 m compared to the mass balance simulations (Figure 8, left panel).

The observed densities for which the observed mass balance is based upon are 490, 481, 599 and 576 kg m$^{-3}$ for the years 2015-2018 while the modeled Crocus snow densities, on average, are lower (see Figure 8, right panel). Snow densities from Noah-MP are also lower, but higher than the Crocus snow densities. In the first modeling year of 2015 Figure 8c), the snow density with Noah-MP is uniform over all elevations. However, as the simulations continued over several seasons (Figure (8f, 8i and 8l), the snow density increases with height for the following years. The reason for this increase is that Noah-MP only has three model layers. When new snow in the winter accumulates, some of this snow is merged with higher density multiyear snow from previous years in the accumulation zone of the glacier. It is therefore difficult to estimate the actual modeled snow density of the new seasonal snow layer in the accumulation zone with Noah-MP in a multiyear simulation. The underestimation of density with Crocus is in agreement with Queno et al., (2016) who found that the bulk density in their study using Crocus also was underestimated and that Crocus tends to underestimate the snow compaction. On the other hand, since the density used to approximate the observed mass balance is taken from only one location, there are likely some uncertainties in the representability of this density over the entire glacier.

Ground penetrating radar observations were gathered in April 2017 and 2018. Note that this time period is not the same as when winter mass balance is determined with stake observations which occurred at the end or after the accumulation season (Figure 10). Figure 11 shows both modeled and observed spatially distributed accumulated snow for the respective winter season, and the snow depth as a function of height. Figure 12 shows a 2-D histogram of the same data. Although these data show some scatter, and isolated observations significantly diverging from the model output (blue), the occurrence of such points is between 10-100 times lower than those which plot close to the one-to-one comparison (red). As such, the majority of depths observed within the GPR dataset match the model outputs to within 1 m. Crocus has more variation in snow depth over the glacier compared to Noah-MP (see Figure 12), but the modeled glacier in both snow models has less heterogeneity compared to the observations (Figure 11). For 2017, we can see that there are areas over Rembesdalskåka where the model clearly underestimates the snow thickness, as can also be seen with the point observations from NVE in Figure 10. However, at other locations on Rembesdalskåka the comparison is favorable. Also, in the northern and western part of Hardangerjøkulen (the entire glacier complex) the model matches the observed snow thickness quite well. In the middle part of the glacier, Crocus estimates deeper snow depth compared to Noah-MP and is closer in agreement with the observations both in 2017 and 2018. The reason for the lower snow depth in Noah-MP is likely due to the high snow density at higher elevations (see Figure 8)

In section 2.1 we mentioned the importance of adding sublimation of blowing snow in our simulations. Figure 13 shows the snow thickness and the respective scatterplot for when the sublimation of blowing snow is not included (which is the default in the SURFEX V8.0 setup (as downloaded)). As can be seen, the simulated snow thickness is slightly higher than the observations with the GPR, and this is especially true at the eastern part of Hardangerjøkulen. During the 2017 winter season, this region had on average the strongest winds, causing more sublimation from snowdrift than at other locations on the glacier. Without including sublimation of blowing snow, the simulation overestimates snow thickness. However, for Rembesdalskåka,

the overall winter balance increases when excluding the sublimation due to blowing snow and compares slightly better with observations (not shown). The resulting streamflow from turning off sublimation of blowing snow is about a 4% increase (not shown). We need to note that Figures 11 and 13 show pixel-to-pixel variations in the Crocus output. This is not due to variations of atmospheric forcing (which has a 1 km grid spacing compared the 100 m grid spacing of the WRF-Hydro/Glacier simulations) or blowing snow. We suspect the pixel-to-pixel variations arise from small vertical resizing errors of the very

thick glacier layers, for where we relaxed some of the test requirements when resizing. This does not change the conclusions in this paper, and work is in progress to address this issue.

## 5.2 Albedo

As discussed earlier, Crocus calculates albedo based upon the modeled snow properties, while Noah-MP albedo is dependent on snow age alone. To compare modeled versus observed albedo, we use observations from MODIS-Terra and MODIS-Aqua

daily snow cover and albedo products. To investigate different regions of the glacier (accumulation versus ablation area), we picked two different locations of the glacier. Figure 14 shows the albedo near the top of the glacier, where the accumulated snow typically does not melt to bare ice during summer (see Figure 1 for location). The grey dots represent albedo from MODIS-Terra and the black dots represents MODIS-Aqua albedo. Solid line is albedo from Crocus while dashed line is albedo from Noah-MP. The albedo is shown for the months May through August for each modeled year, thus the start of the melting

season is included. Figure 15 shows the same as Figure 14 but closer to the north-north-west edge of the glacier (see Figure 1 for location), where accumulated snow often completely melts during the summer season. As can be seen, the modeled albedo at both locations line up well with the observed albedo. The decrease in albedo at the end of the accumulation season, as the snow is aging, is well captured for both Noah-MP and Crocus. This is especially evident in 2015 in Figure 15.

The rapid increase in albedo throughout the summer is due to snow events. The albedo determined with Crocus typically

decreases rapidly after an event since the albedo is based upon snow properties, the change of the snow properties over time and the thickness of the snow layer. Noah-MP albedo, on the other hand, decreases slower after each snow event since it is only dependent on the snow age. At the edge of the glacier (Figure 15), the observations show a gradual decrease in albedo as the snow starts to melt away to the bare ice (see for example July in 2016 and 2018 in Figure 15), while the Crocus model has a more abrupt decrease as the snow is completely melted away. The lowest value of albedo in Crocus over the ice is 0.35. The

Noah-MP does not have this decrease in albedo as the land surface category over glacier grid points is assumed to be that of old snow with minimum albedo of 0.675. Therefore, since snow in the three-layer snow model is allowed to have lower albedo than 0.675, when the bare land surface category (glacier) is revealed, the albedo actually increases (see 2018 in Figure 15). The correlation coefficients between modeled and observed albedo is given in Table 1. Overall, the Crocus albedo compares much better with the observations than the Noah-MP albedo in the summertime when the snow melts and the ice becomes

apparent due to the assumption of glacier land surface category in Noah-MP.

## 5.3. Discharge

Figure 2 includes the catchment areas of the Middalselvi (which is fed by Midtdalsbreen) and Finseelvi, locations where discharge measurements were gathered in 2017 and 2018. Figure 16 shows the modeled and observed discharge in Middalselvi for summer 2017 and 2018. Also shown is the daily precipitation at Finse station. The modeled precipitation at Finse AWS

station compares well with the observed, except for one precipitation event in late July 2017, where the model predicts too much precipitation at the Finse station. During the precipitation events, both the Crocus and Noah-MP simulations and observations show increases in discharge but with some variability in strength. Interestingly, during the large dry events in the end of June and through most of July of 2018, the modeled and observed discharge is very similar.

In regards to the observed peak flow at the beginning of melt season in May, the model does not predict such a peak flow.

These peaks mark the onset of the hydrologic season in the rivers. They occurred almost at the same time in the two catchments in two consecutive years under similar preconditions. Photographs taken on site prior the event indicate that water was flowing over the snow pack and was carving down to the river bed during the event. Probably some part of the peak is due to a pulsed meltwater flux and/or the associated pressure build up to that time which is subsequently released. Therefore, when evaluating the accumulated discharge, we start the accumulated period directly after the large spike in the observations. The accumulated

discharge for 2017 shows that the observations are slightly higher than the discharge simulated with WRF-Hydro/Glacier (with Crocus). They still follow closely, but WRF-Hydro/Glacier flattens out in the fall earlier compared to observations. This is likely due to lack of using the baseflow/groundwater module in these specific WRF-Hydro/Glacier simulations, which could add some water to the surface streamflow. The WRF-Hydro simulations with only Noah-MP are consistently much lower compared to both observations and WRF-Hydro/Glacier at Middalselvi. For 2018, simulations with both Noah-MP over the

glacier and Crocus over the glacier follow the observations reasonably well, however they both overpredict the discharge in the end of May and early June. Around July 30, the Crocus simulations increase slightly more than the observations while the Noah-MP simulations reduce discharge considerably at the end of August. The Crocus simulations still have discharge comparable with the observations until September. As shown in Figure 8j, Crocus has a larger negative summer mass balance compared to the observations which, for which the large discharge in early June in both Noah-MP and Crocus is likely

connected to. Thus, even though Noah-MP compares well with observations at the end of the melt season in 2018, this is most likely due to too much melt in the beginning of the melt season and not the skill of the Noah-MP simulations.

One of the reasons for the lower discharge in Middalselvi with the Noah-MP compared to Crocus is likely the lack of glacier runoff once the seasonal snowpack has melted. The Middalselvi is about 60% glacierized, while Finseelvi is only 14.7% glacierized. As can be seen in the discharge data for Finseelvi (Figure 17), the Noah-MP compares better with the observations

in the entire melt season than it does for Middalselvi (especially in 2017, in 2018 the discharge is too high in end of May and beginning of June as discussed above). Crocus is not shown in Figure 17 since, in this specific setup, the Crocus simulation

only feeds rivers downstream of Hardangerjøkulen and the WRF-Hydro/Glacier system uses the Noah-MP snow model in the Finseelvi catchment area.

We hypothesize that a second reason for the lower discharge with Noah-MP compared to Crocus is likely the albedo treatment. In 2017, there are many small snow accumulation events. This causes the Noah-MP albedo to rarely go below 0.7 at the top of the glacier (Figure 14). However, both observations and Crocus simulations have albedo closer to 0.6. At the edge of the glacier (Figure 15), the times where Crocus is close to 0.6 is longer, and in August, the albedo for both Crocus and the observations are around or below 0.4. For 2018, the albedo of Noah-MP and Crocus both compares well with the observations at the top of the glacier, due to the prolonged dry periods (no new snow events that lead to overestimation of albedo in Noah-MP). During this dry time period, the discharge from Noah-MP and Crocus are comparable. At the edge of the glacier, Noah-MP overestimate the albedo from end of July. This is when all accumulated snow is melted and the surface is bare ice. The Noah-MP albedo increases to 0.67 (that of old snow), while Crocus and observations indicate that the albedo should be that of ice. This is also the time period where streamflow from Noah-MP significantly diverges from Crocus (Figure 16, early July). As an illustration we re-ran Crocus for the 2018 season and substituted ice albedo with snow albedo. Figure 18 shows the accumulated streamflow with the sensitivity study (red curve). It is clear that the streamflow is reduced compared to the original simulation from the end of July at the time period where surface albedo at the edge of the glacier is reduced to that of ice. Bonekamp et al. (2019) found that the recharge of snow albedo from summer snow events in their simulations (WRF with Noah-MP) over glaciers in the Shimsal catchment in Karakoram had large impact on the summer mass balance. We remind that even though the final accumulated streamflow using snow albedo instead of ice albedo seem to be more in line with observations, this assumption does not actually improve the simulation as it reduces the streamflow for the wrong reason.

## 6. Conclusion

The detailed physically-based snow model Crocus was implemented into the WRF-Hydro system to act as a glacier model. This model supports a large number of snow layers with dynamic density, thickness and snow properties. Furthermore, the albedo is prognostically calculated based on the physical properties of the snow. The implementation of Crocus allows for a direct estimation of glacier surface winter, summer and net balance. The snow accumulation is already represented reasonably well within the WRF model when using Noah-MP land surface model. However, the results of comparative studies performed here show that the Crocus snow model improves the simulation of melting ice by allowing for snow to melt to bare ice, and using the albedo of ice for further melting, instead of an assumed albedo over glacier land surface category. This is critical for representation of glacier wastage during the summer season. Furthermore, the integration of Crocus with WRF-Hydro allows for the discharge to be directly affected by the melting ice. Major conclusion of the evaluation of implementation of the WRF-Hydro/Glacier system and the input forcing are summarized as follows:

- WRF produces meteorological forcing data comparable with the observations. For example, the temperature and wind speed at Finse are in excellent agreement. The 1-km grid spacing simulation results are used for input to the WRF-Hydro/Glacier system to evaluate glacier surface mass balance and subsequent streamflow.

- Observed solid precipitation on Finse is affected by under-catch due to strong winds. Using observed precipitation from this location without correcting for this under-catch will result in an underestimation of winter surface mass balance. However, the high-resolution model simulations with WRF at 1 km shows a great ability in producing winter precipitation. This can also be seen in the generally good agreement of winter mass balance.

- The simulations with Crocus are doing a reasonable job in reproducing winter, summer and annual surface mass balance with bias of less than 11% in summer and about 15 % in winter (excluding 2017 which have a bias in forcing data). Noah-MP is doing slightly better than Crocus in simulating winter mass balance, but cannot be used for directly evaluating the summer mass balance since Noah-MP can only melt accumulated snow from simulation start and there is a lack of preexisting ice in the initialization of the simulation.

- Snow depth is reproduced well, both compared with the GPR observations and observations from NVE (except 2017 compared with the NVE observations). However, the mean snow density with Crocus and Noah-MP is 5-20% underestimated compared to NVE observations, suggesting that some of the inconsistency of snow depth (little bias) and mass balance (some bias) comparison is due to the bias in modeled snow density. Also note that the snow density is only measured at one location and assumed to be the same over the entire glacier. Furthermore, some of the bias between observations and model results are also likely due to not accounting for lateral snow distribution in the model.

- By applying Crocus over a glacier, the surface albedo is better represented in the ablation region where ice (with lower albedo compared to snow albedo) becomes present. This allows for better representation of melting and subsequent discharge in a glacier dominated catchment with WRF-Hydro/Glacier.

In conclusion, both the more physically based snow model Crocus, and Noah-MP are simulating the winter mass balance and snow thickness well. The WRF-Hydro discharge simulations in the minimally glacierized, lower-elevation catchment (Finseelvi) is also reasonable with the Noah-MP model (keeping in mind that neither WRF-Hydro nor WRF-Hydro/Glacier has been calibrated for the simulated region). Despite this, WRF-Hydro/Glacier shows more realistic performance in the glacierized catchment due to the fact that it allows modeling of negative net mass balance and uses ice surface albedo where all accumulated snow is melted, both elements which Noah-MP currently does not include. We note that we have not currently included lateral movements of the glacier or any treatment of blowing snow in WRF-Hydro/Glacier. The impacts of including these additional physical processes may afford additional improvements in annual snowpack and snowmelt representation, particularly with respect to their spatial distribution. Finally, the forcing at 1 km does not account for any topographic variations

in the 100 m domain, thus snowpack evolution at 100 m scale is not included. More development and testing of WRF-Hydro/Glacier are certainly required and are currently underway. In particular the system should be applied and assessed over other glaciated regions of the world. It is a tool that can contribute to better understanding of future hydroclimate; especially in light of the continued widespread glacier retreat and decline in mass balance, which has profound implications for future water resources in many parts of the world.

## Acknowledgements

This work has been funded through the Research Council of Norway (RCN) through EvoGlac (grant no. 255049). The hydrologic fieldwork is funded by the Strategic Research Initiative 'Land Atmosphere Interaction in Cold Environments' (LATICE) of the University of Oslo. Radar fieldwork was part-funded by the European Union's Horizon 2020 INTERACT project, under grant agreement No 730938 'Snow Accumulation Patterns on Hardangerjøkulen Ice Cap (SNAP)', NERC SPHERES DTP grant NE/ L00257 and NERC Industrial CASE Studentship NE/P009492/1. Field surveys were supported by Emma Pearce and Siobhan Killingbeck (University of Leeds) and the logistics expertise of Kjell Magne Tangen (Tajo AS). Jostein Bakke (University of Bergen) supplied the GPR system for the 2017 survey. Atle Nesje received in-kind funding from Department of Earth Science at UiB. Kjetil Melvold received in-king funding from NVE. The National Center for Atmospheric Research is sponsored by the U.S. National Science Foundation (NSF). Hallgeir Elvehøy (NVE) is acknowledged for valuable help and comments with the glacier mass balance data.

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

**Tables**

**Table 1:** WRF and WRF/Hydro model configuration

| Model Features | Configuration |
| --- | --- |
| Horizontal resolutions | Domain1: 3 km (200x200 grid points) |
| | Domain2: 1 km (160x160 grid points) |
| Vertical levels | 50 |
| Topography | USGS |
| Forcing | ERA-I |
| Time step | Domain 1: 15 s |
| | Domain 2: 3 s |
| **Physics** | |
| Microphysics | Thomspon-Eidhammer aerosol aware (Thompson and Eidhammer, 2014) |
| Planetary Boundary Layer | YSU (Hong et al., 2006) |
| Radiation | RRTMG (Iacono et al., 2008) |
| Land Surface Model | Noah-MP (Niu et al., 2011) |

| WRF-Hydro | |
|---|---|
| Horizontal resolution | 100 m (500x500 gridpoinst) |
| Activated routing | Surface overland flow routing |
| | Channel routing |
| | Subsurface routing |
| Routing model time step | 15 s |



**Table 2:** Correlation coefficient between simulated and observed albedo. Values outside parenthesis are albedo from Modis Aqua while values inside parenthesis are albedo from Modis Terra. The highest correlation for each year is shown in bold font.


| Year -> Model | 2015 | 2016 | 2017 | 2018 |
|---|---|---|---|---|
| **Crocus (top)** | 0.74 (0.88) | 0.54 (0.67) | 0.69 (0.64) | 0.36 (0.42) |
| **Noah-MP (top)** | 0.59 (0.77) | 0.01 (0.12) | -0.14 (-0.27) | 0.32 (0.51) |
| **Crocus (edge)** | 0.89 (0.89) | 0.81 (0.83) | 0.89 (0.80) | 0.82 (0.85) |
| **Noah-MP (edge)** | 0.71 (0.90) | -0.33 (-0.21) | -0.06 (-0.21) | 0.15 (0.36) |




**Figures**

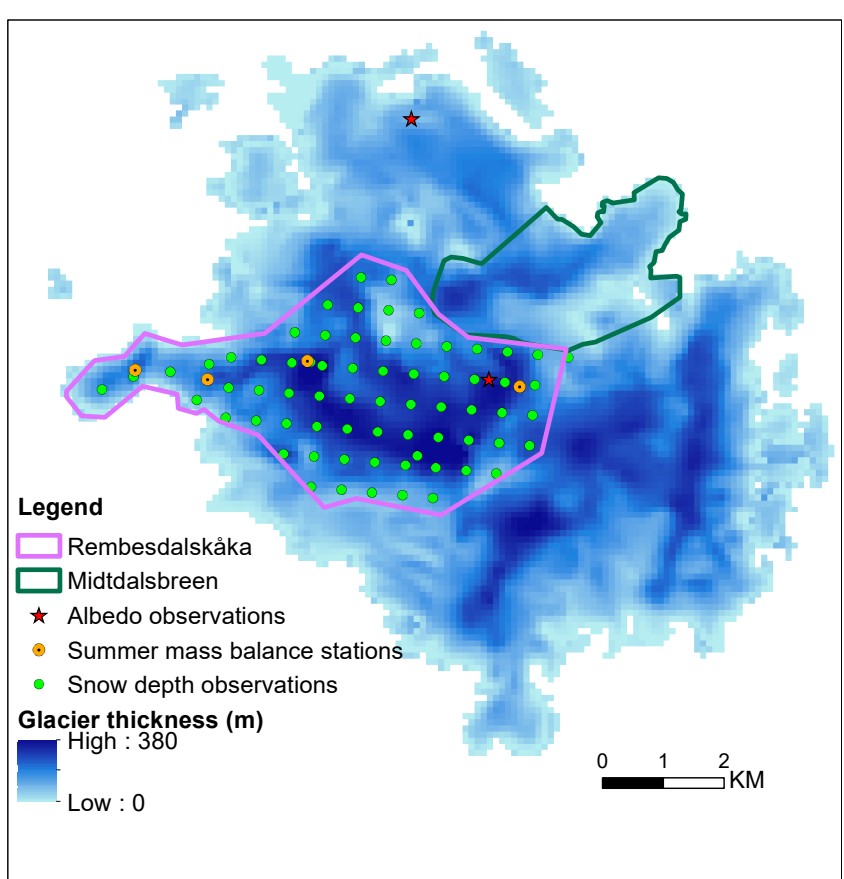

**Figure 1: Glacier thickness and present extent for Hardangerjøkulen. The outlet glaciers Rembesdalskåka, Midtdalsbreen are also**
**indicated. The green dots indicate locations where stake observations for winter balance were obtained, while orange dots indicate**
**location of observations for summer balance. Stars shows location of albedo comparison with model against observations.**

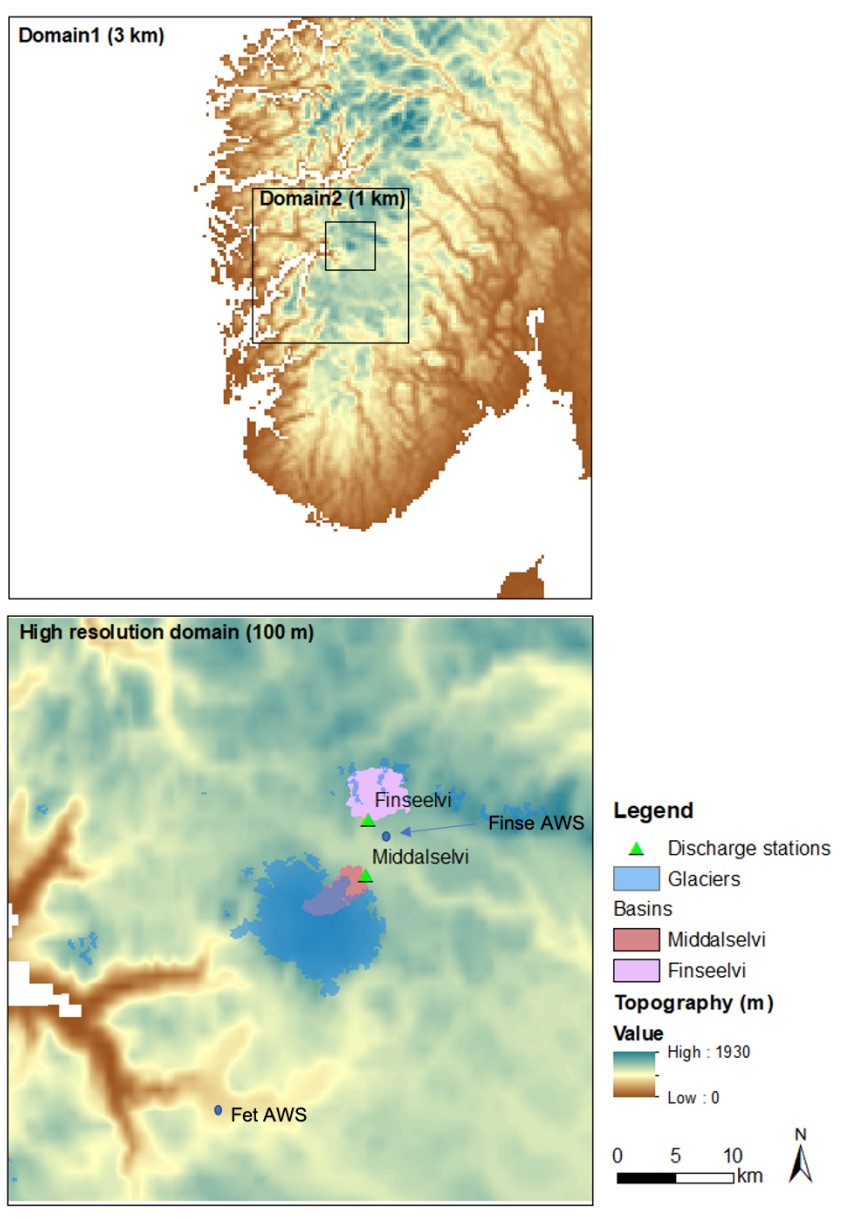

**Figure 2: Model domains. The top figure shows the WRF domains with 3 km and 1 km grid spacing. The bottom figure shows the domain of the high resolution (100 m grid spacing) used with WRF-Hydro/Glacier. Furthermore, the outline of Hardangerjøkulen and the river basins of the two rivers with discharge observations are included as well.**

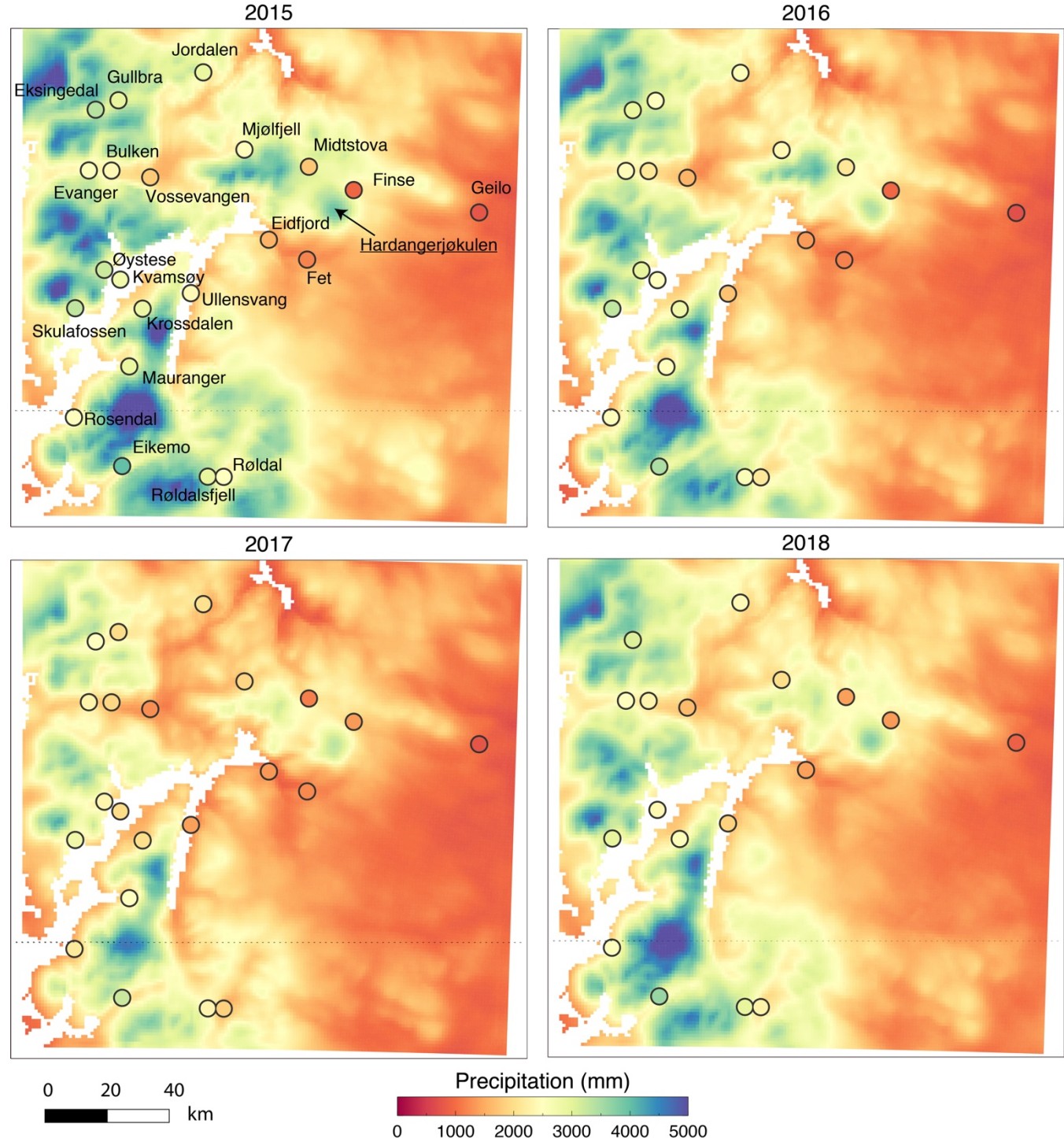

**Figure 3: Simulated and observed precipitation (October – September) for mass balance years 2015-2018. Circles represent observations from AWS. The location of Hardangerjøkulen is also indicated.**

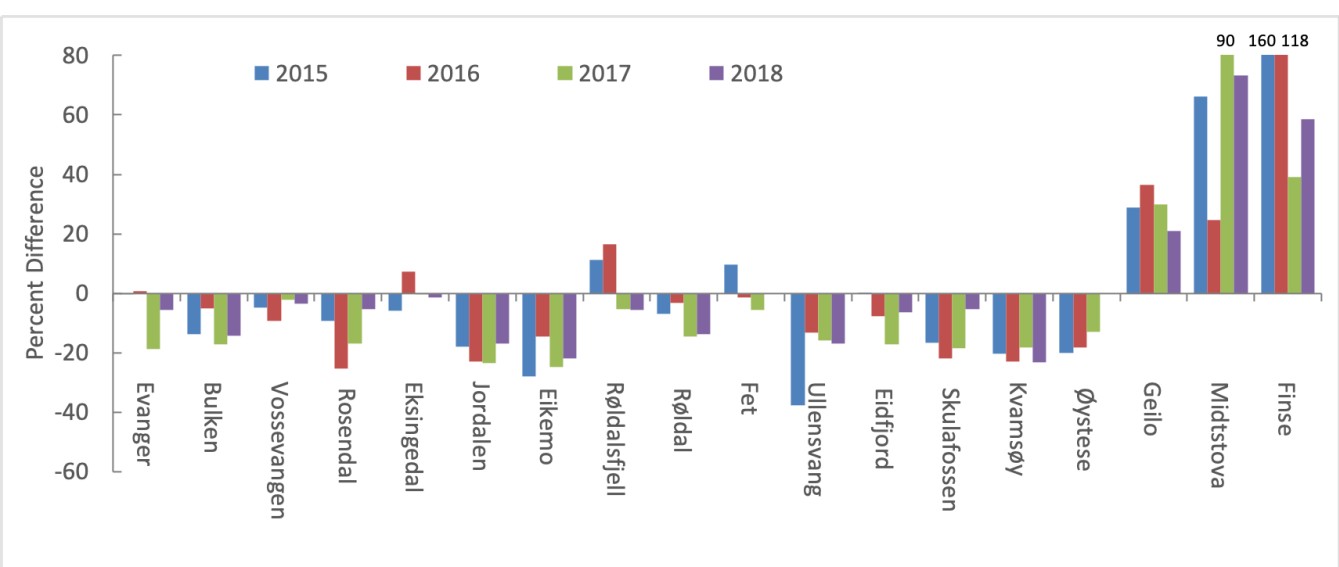

**Figure 4: Percent difference in simulated and observed precipitation at all stations for mass balance years 2015-2018. Some values above Midtstova and Finse are above the axis limit and are therefore indicated with labels.**


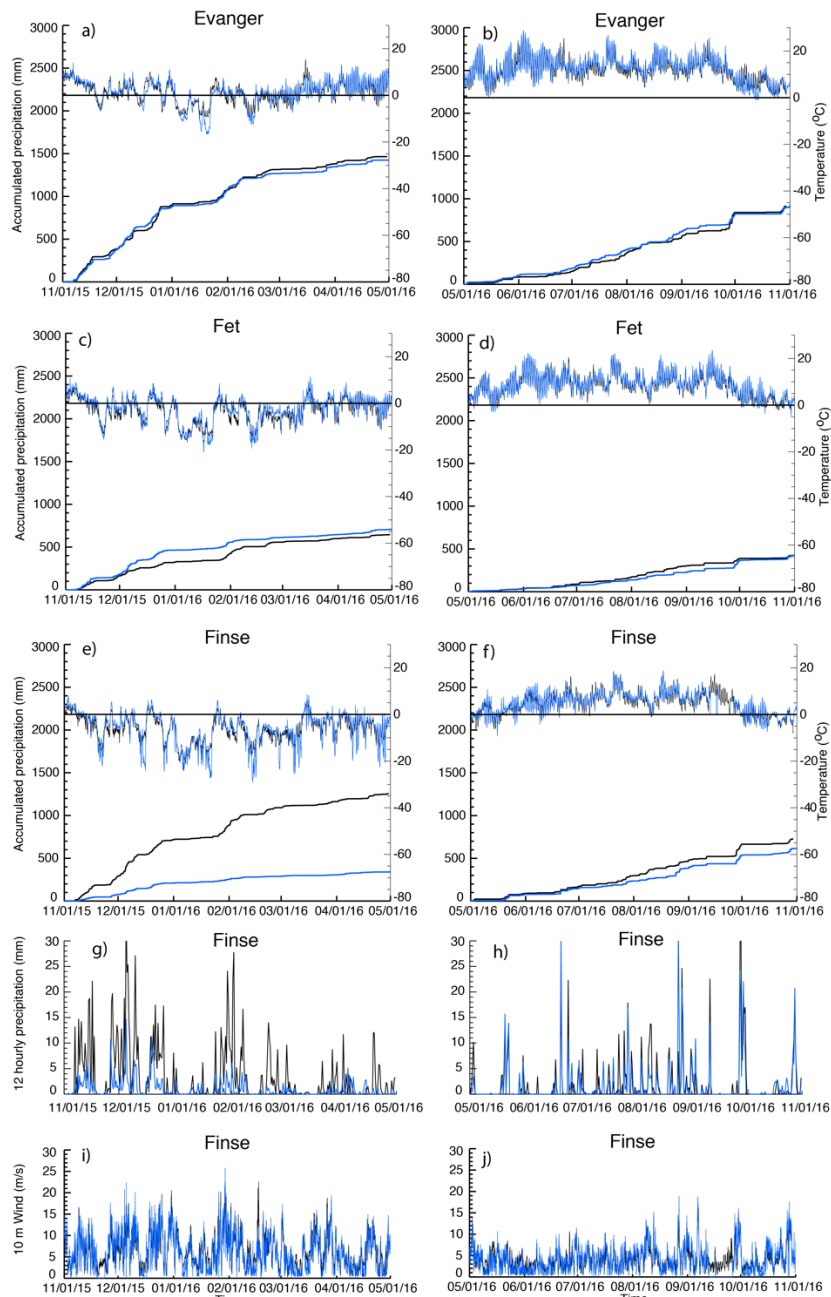

**Figure 5: Observed (blue) and modeled (black) accumulated precipitation and temperature at Evanger (a, b), Fet (c, d) and Finse (e, f) stations for the 2016 winter season (left panel) and 2016 summer season (right panel). g) and h) shows time series of precipitation. The bottom panel (i, j) shows the 10 m wind speed at Finse.**

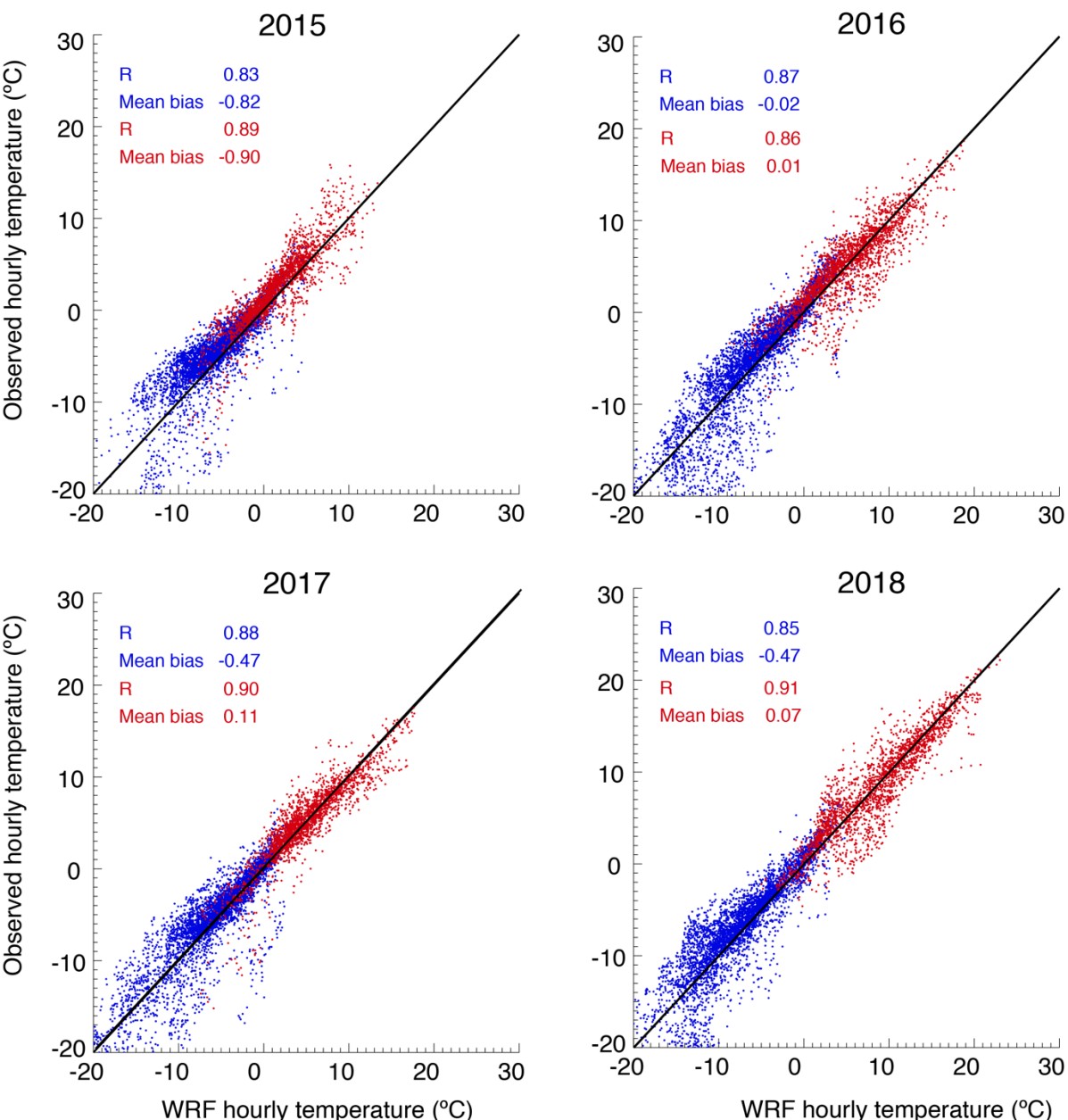

**Figure 6: Scatterplot of observed and simulated temperature for mass balance years 2015-2018 at Finse station. Blue colors represent the winter season and red colors represent summer season.**


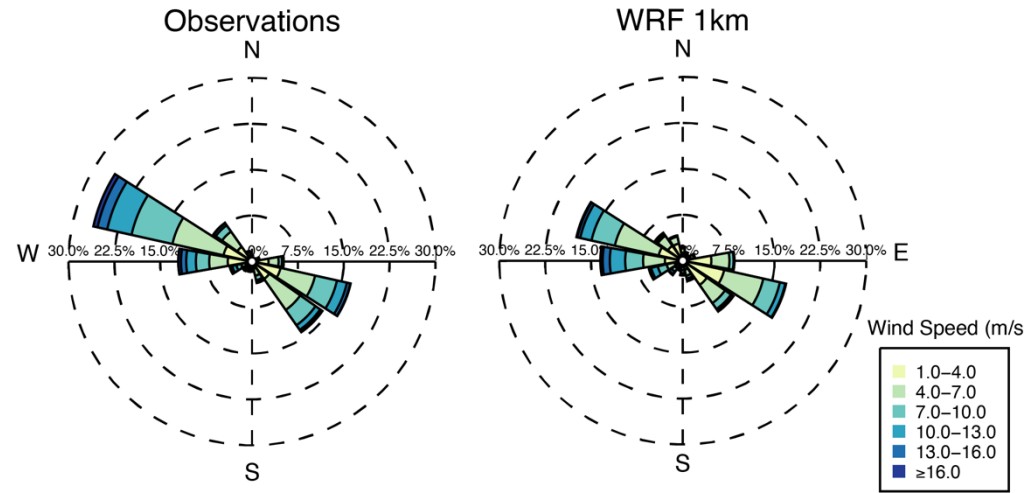

**Figure 7: Wind rose of simulated and observed 10 wind speed and direction at Finse AWS stations for the entire simulation period.**

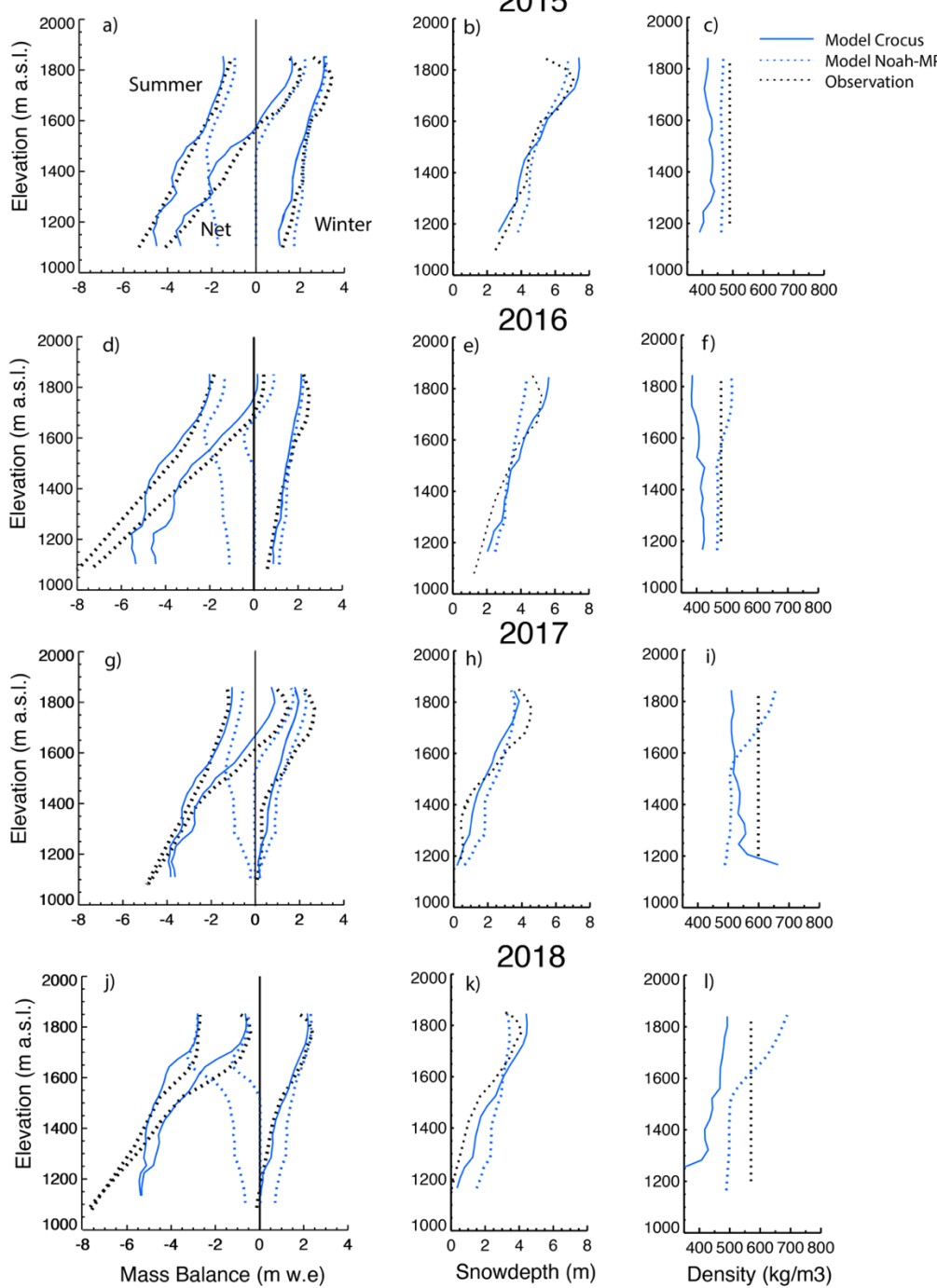

**Figure 8: Left panels: Mass balance across Rembesdalskåka with height (a, d, g and j). Middle panel (b, e, h and k): Accumulated snow thickness associated with the winter mass balance. Right panel (c, f, I and l): Snow density of the accumulated snow. Each row represents a different year.**

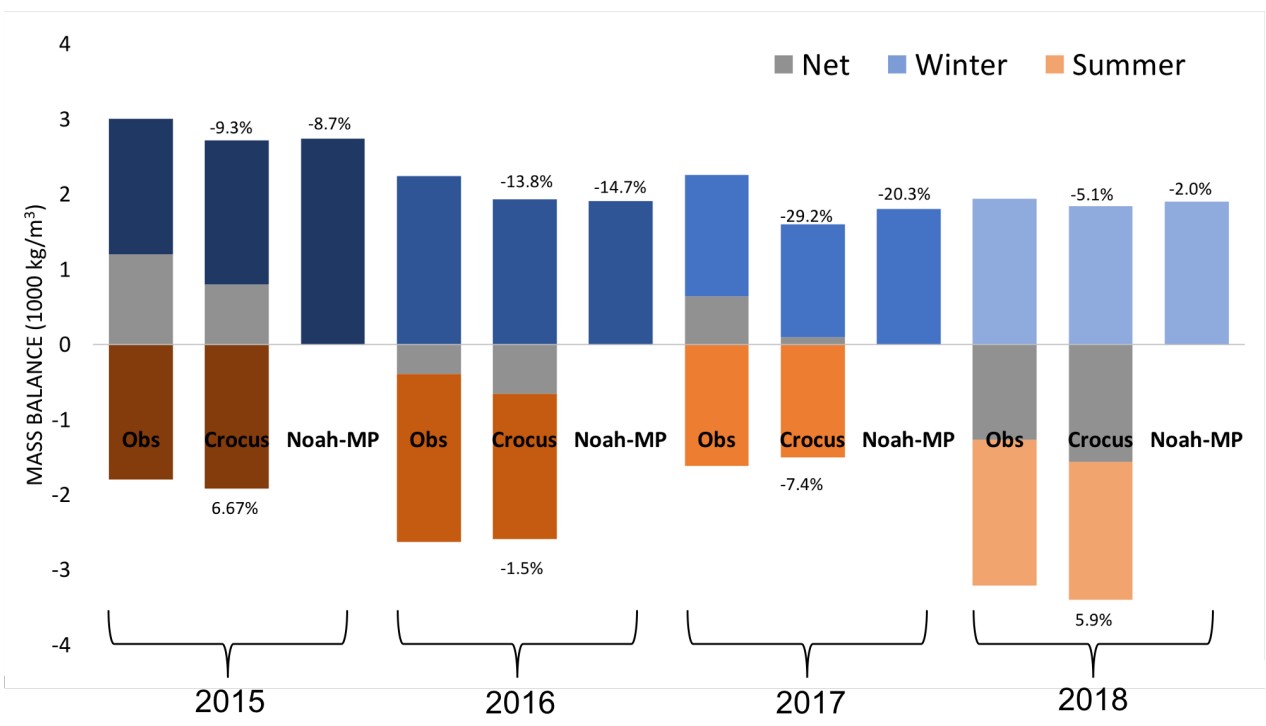

**Figure 9: Box plot of winter, summer and net balance for the mass balance years 2015-2018 for Rembesdalskåka. Only winter mass balance is shown for Noah-MP.**

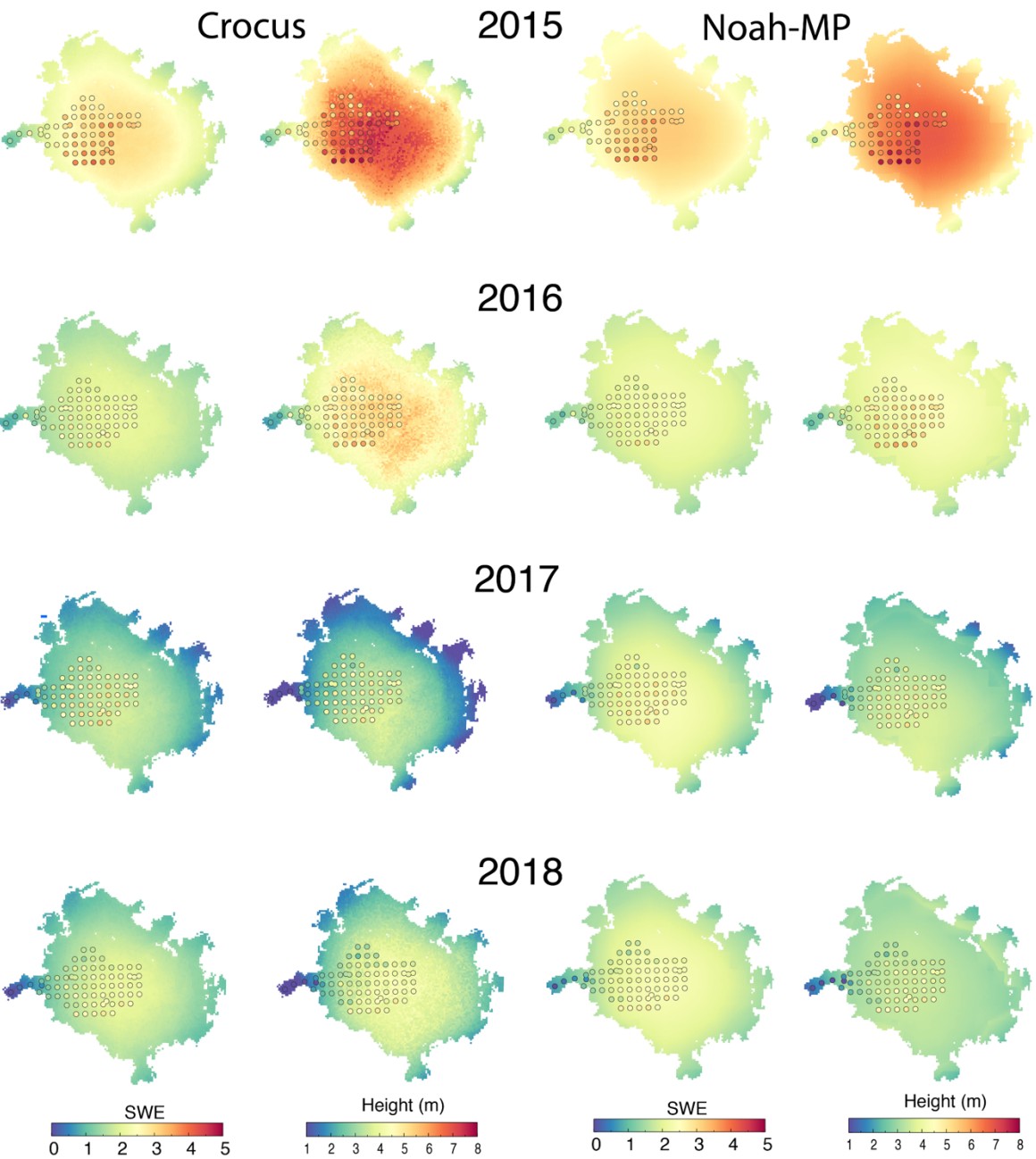

**Figure 10: Mass balance at Rembesdalskåka. Two left panels: Winter mass balance and accumulated snow height across Hardangerjøkulen as modeled with Crocus. Colored circles are observations from NVE. Two right panels: Same as the two left panels but as modeled with Noah-MP**

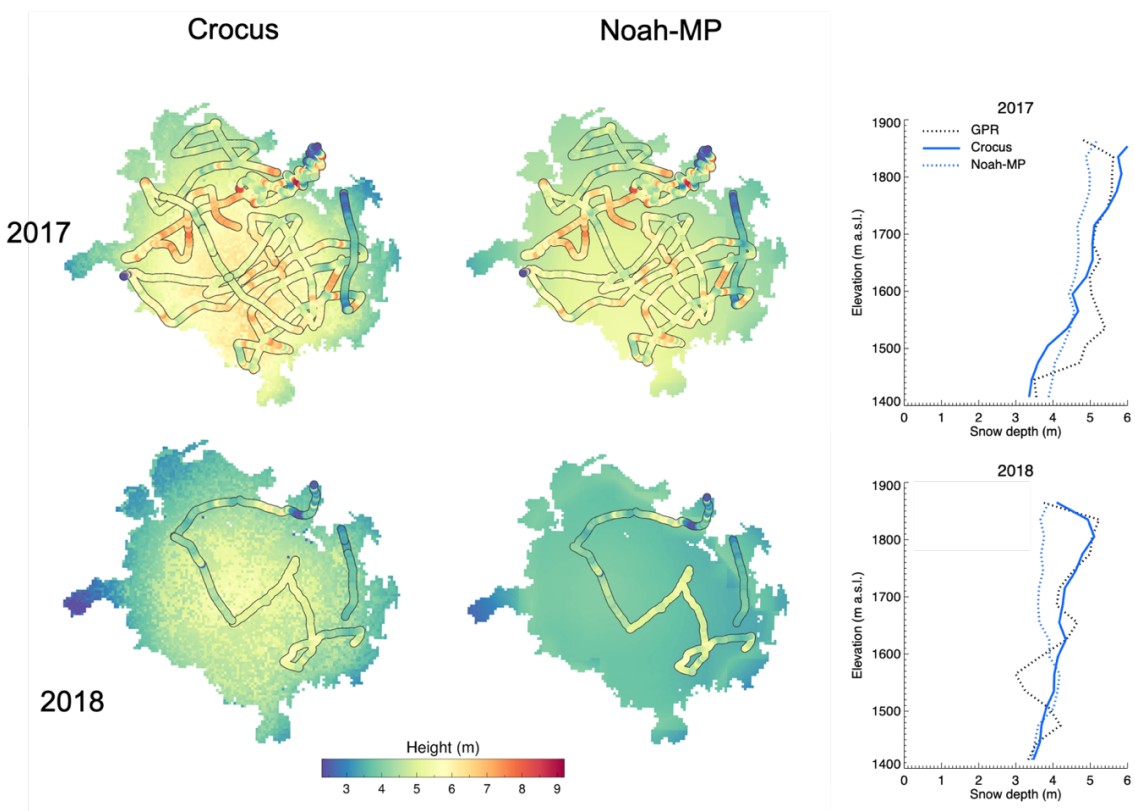

**Figure 11: Modeled and observed snow accumulation in April. Observations are with Ground Penetrating Radar. Left plots are simulation results with Crocus and middle plots are simulation results with Noah-MP. Rightmost plot shows the snow depth as a function of elevation**

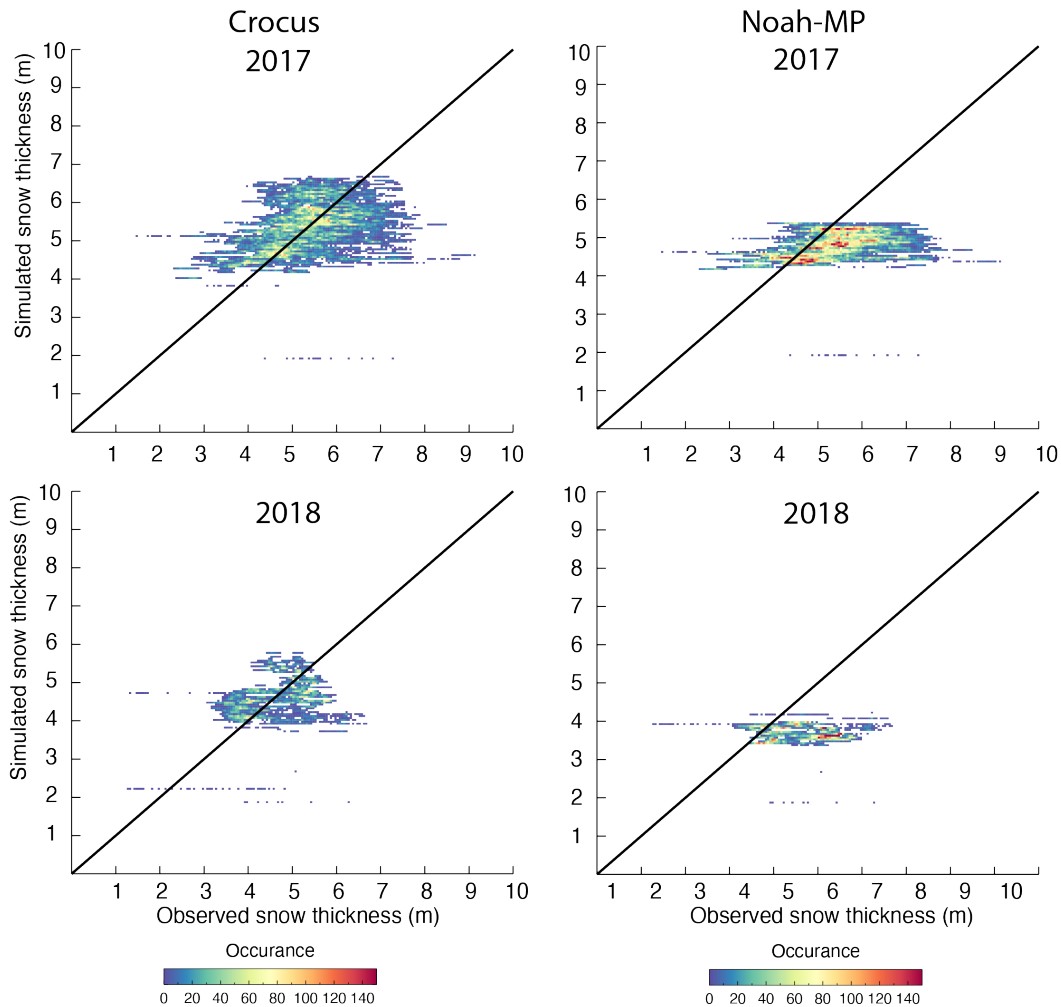

Figure 12: Scatter plot of modeled and observed snow accumulation. The observations are with Ground Penetrating Radar. Top plots are simulation results with Crocus and bottom plots are simulation results with Noah-MP.

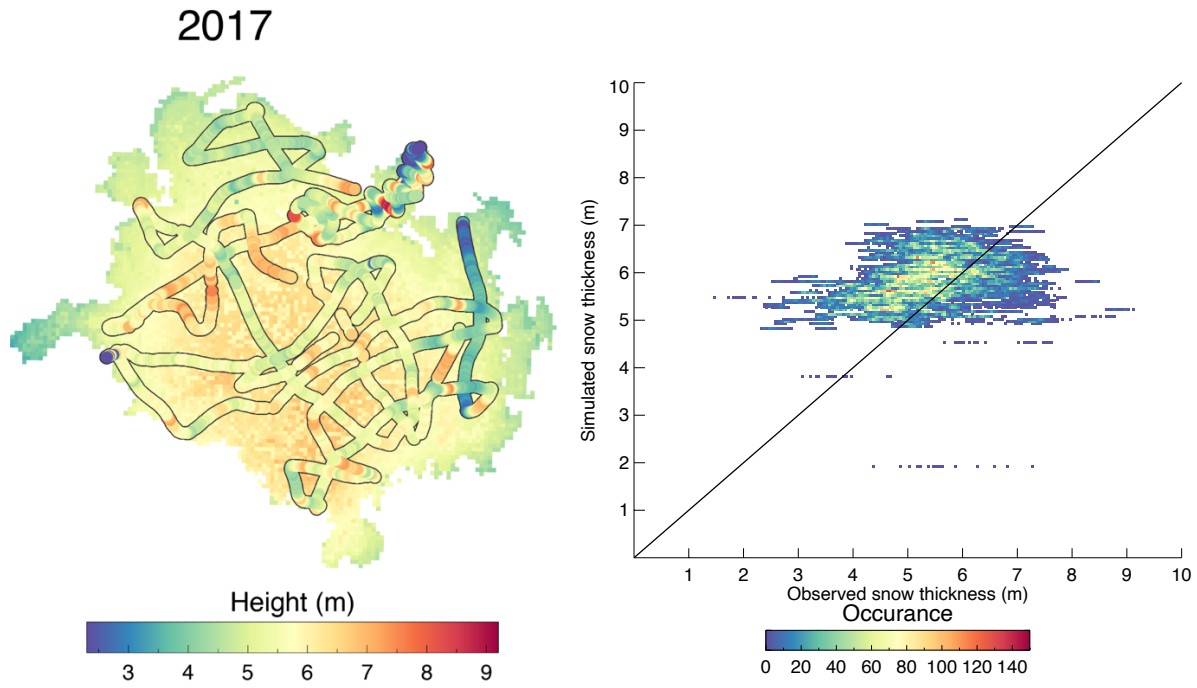

**Figure 13: Simulated and GPR observed snow depth for 2017, but with no effect of sublimation from snow drift in the simulations.**

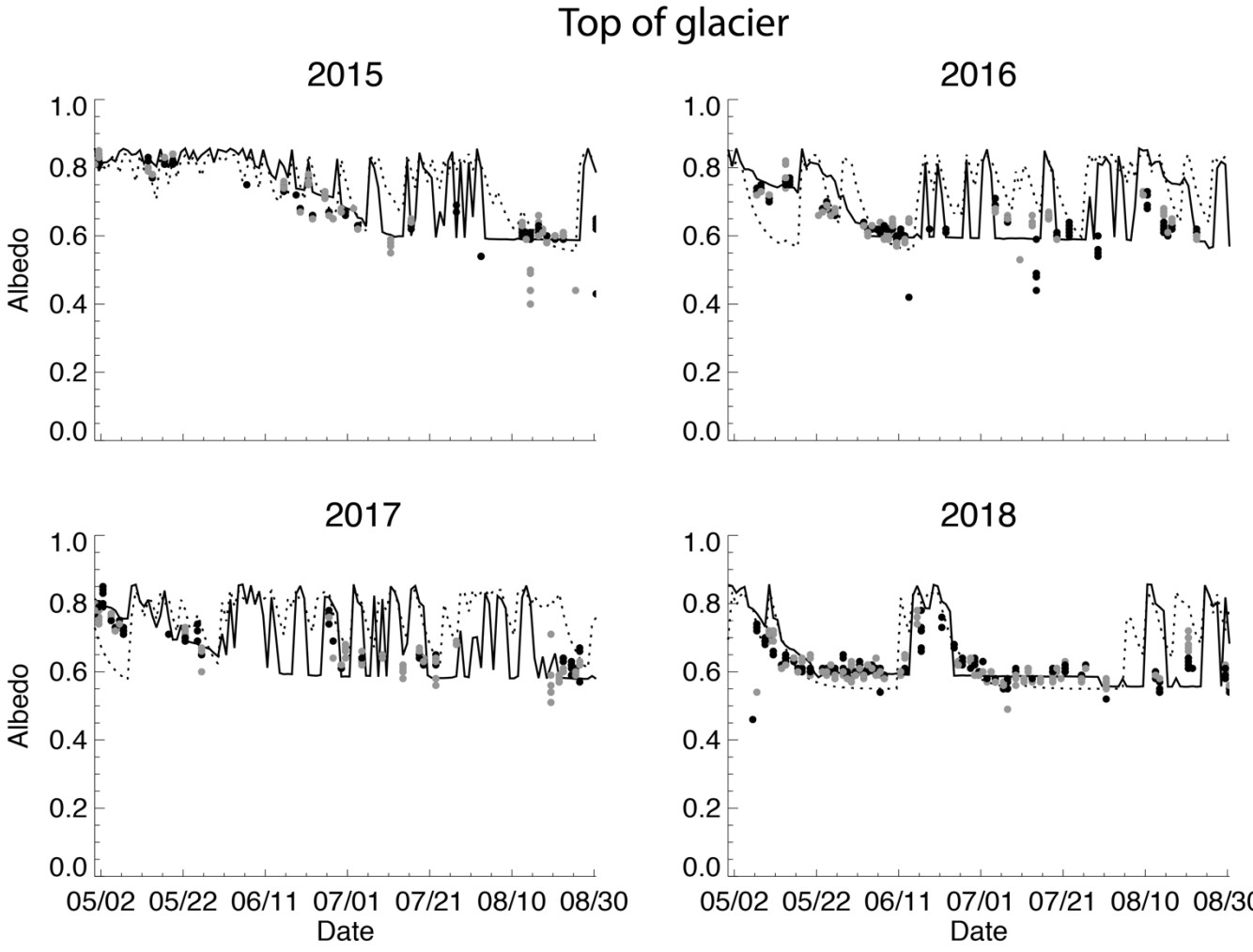

 **Figure 14: Observed and modeled albedo at the top of Hardangerjøkulen. Dots represent observed albedo from the MODIS satellite (Grey: Terra, Black: Aqua). Solid line is albedo from the Crocus snow model, dashed line is albedo from the Noah-MP snow model.**

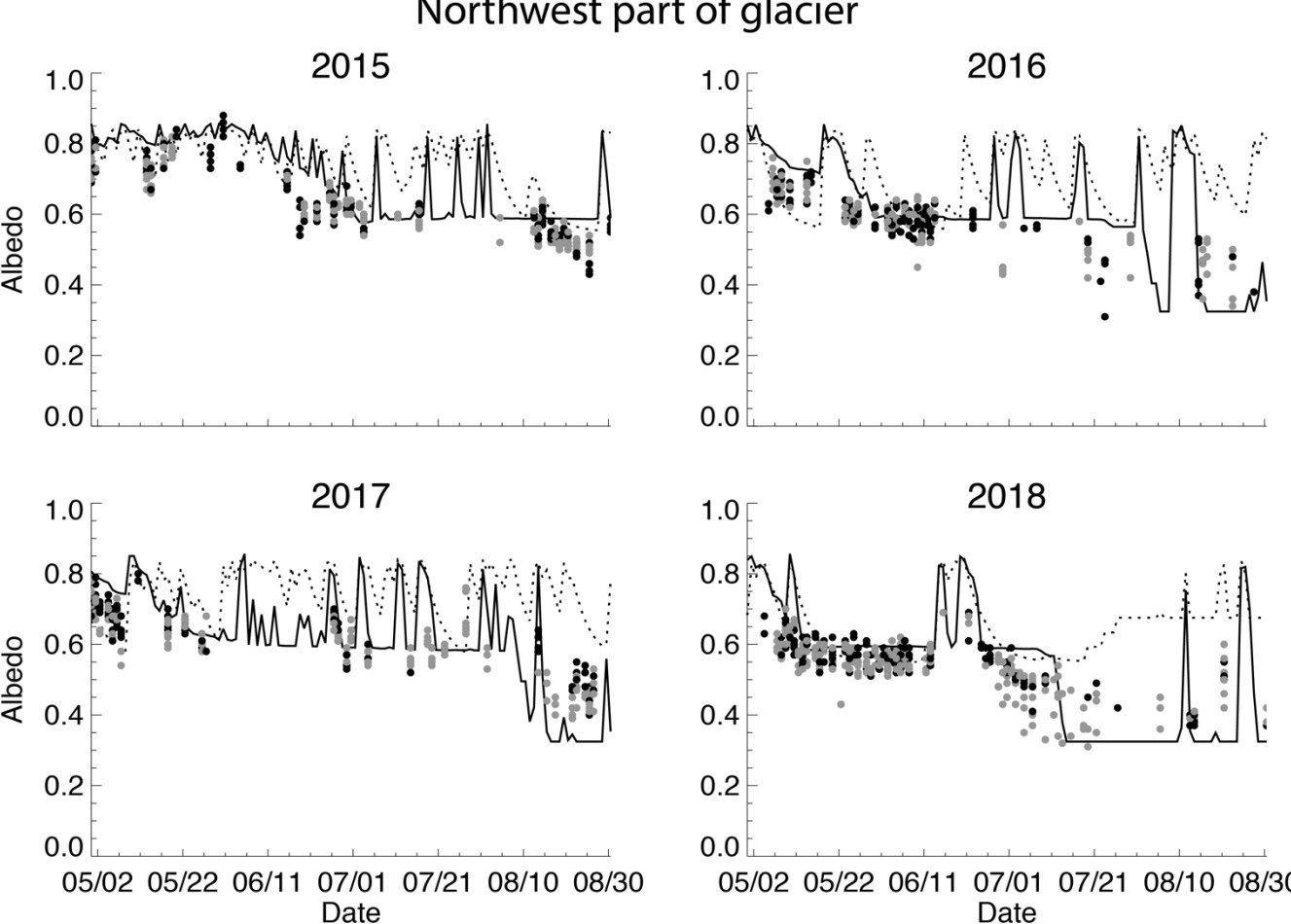

**Figure 15: Same as Figure 14 but at a Northwest part of Hardangerjøkulen, where snow at times completely melts to the ice surface.**

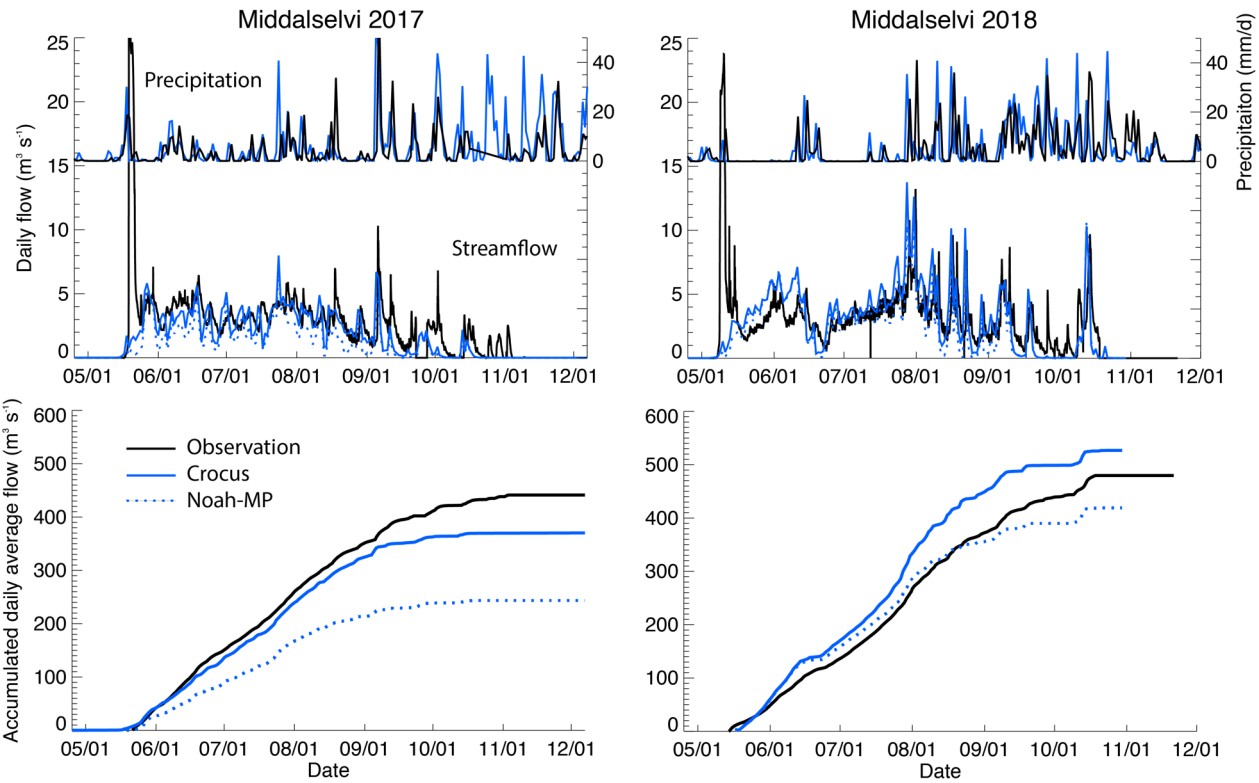


**Figure 16: Top: Observed (black line) and modeled (blue) precipitation at the Finse AWS station and daily discharge at Midtdalselvi. Bottom: Accumulated discharge. Solid blue line is with Crocus and dashed blue line is with Noah-MP.**


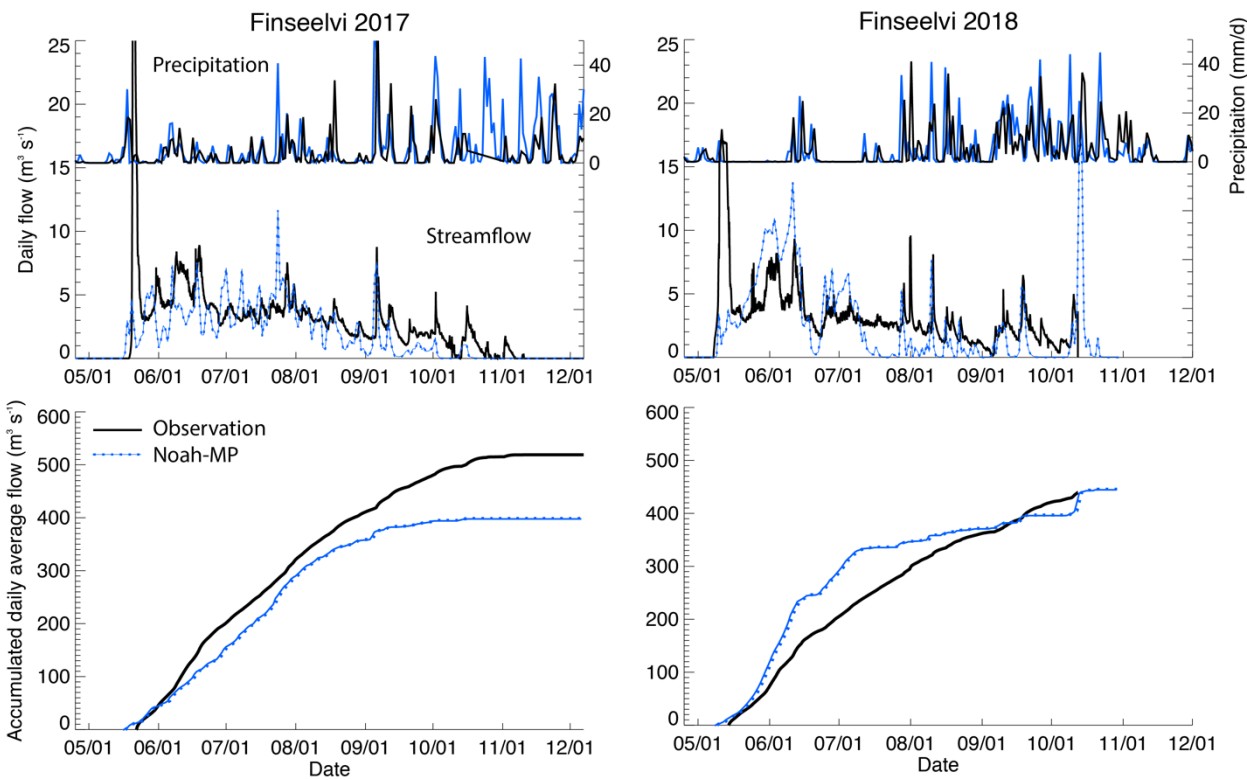

**Figure 17: Same as Figure 16, but for Finseelvi. Note that in this watershed, the WRF-Hydro/Glacier system is not using Crocus, but Noah-MP due to lack of glaciers.**




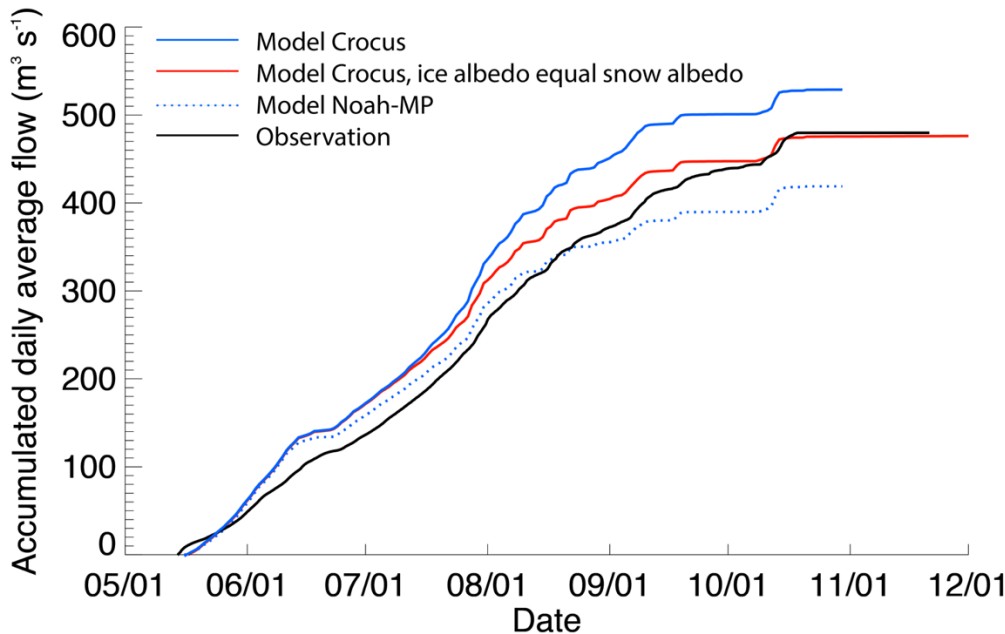

**Figure 18: Accumulated stream flow at Middalselvi in 2018. Red is the sensitivity test where snow albedo is used instead of ice albedo.**