# Peer review of "Mass balance and hydrological modeling of the Hardangerjøkulen ice cap in south-central Norway"

_Hydrology and Earth System Sciences, 2020_

## Referee Comment (RC1) · Emily Collier (Referee) · 20 Jul 2020

Eidhammer and colleagues present a new coupled modelling tool for atmospheric, glaciological and hydrological simulations, where they have integrated the snowpack model Crocus into WRF-Hydro. The authors apply the model to a multi-year, very-high-resolution simulation of Hardangerjøkulen and evaluate its performance with respect to a variety of observations. The integration of Crocus provides an important improvement in the representation of glaciers compared with the Noah-MP land surface model that will increase the utility and reliability of WRF for simulations of glacierized regions. While a small number of previous efforts have been made to improve glacier physics

in WRF, this work includes a novel linkage to detailed hydrological processes and a thorough evaluation over a multi-year time period. The manuscript is well and concisely written, and I recommend its publication in HESS after minor revisions.

Minor comments:

1. The introduction inadequately contextualizes the authors' work with regards to our previous efforts to improve the representation of glacier physics in WRF and their applications. In addition to Collier et al. (2013), there are two more relevant references:

- Collier, E., Maussion, F., Nicholson, L. I., Mölg, T., Immerzeel, W. W., and Bush, A. B. G.: Impact of debris cover on glacier ablation and atmosphere–glacier feedbacks in the Karakoram, The Cryosphere, 9, 1617–1632, https://doi.org/10.5194/tc-9-1617-2015, 2015.

- Aas, K. S., Dunse, T., Collier, E., Schuler, T. V., Berntsen, T. K., Kohler, J., and Luks, B.: The climatic mass balance of Svalbard glaciers: a 10-year simulation with a coupled atmosphere–glacier mass balance model, The Cryosphere, 10, 1089–1104, https://doi.org/10.5194/tc-10-1089-2016, 2016.

2. The authors state that glacier ice in Noah-MP cannot melt several times (Lines 134, 303, 397, 445), however my understanding of this LSM's treatment of glaciers is that the subsurface at glacierized grid points is defined as a fully saturated and initially frozen soil. This "soil ice" can and does melt, sometimes entirely. If my understanding is correct, does this treatment differ in WRF-Hydro, or is drainage of glacier melt not accounted for in the hydrological part of the model?

3. The authors provide relatively few details about the WRF simulations and could consider adding a table with basic information (e.g., grid dimensions, timesteps, physics options, any special settings) to increase the reproducibility of their study. On a related note, was WRF-Hydro/Glacier run with or without a PBL scheme?

4. Line 172: Could the authors comment on the impact of using a reanalysis with

~80-km grid spacing to directly force the outer WRF domain with 3-km grid spacing?

5. Line 188: Why was the model evaluation performed only for the 1-km domain? It looks like the 100-m domain may contain at least the Finse AWS. If so, I suggest the authors also provide a brief evaluation of near-surface variables from this domain, since these data directly force the glaciological and hydrological components.

6. Line 219: The manuscript has quite a few figures. I think the authors could remove Figure 6 and provide the R2 and mean bias in the text. Although simulated wind direction is evaluated, biases and their implications for the results are not discussed elsewhere, so Figure 7 may also be unnecessary.

7. Section 3.1: I suggest moving the model evaluation to the results section. In addition, please describe issues with the measurements and missing data (e.g., Lines 307-312, missing data at Finse visible in Figure 5) in the methods.

8. Line 247: Could the authors provide the dates they used for calculating climatic mass balance in the text or a table? How do the results compare when using the same dates as the observations?

9. Line 293: Could the authors discuss why the winter balance simulated by Noah-MP has, in general, a smaller bias at higher elevations?

10. Line 352: How were these two locations selected for comparison with MODIS?

11. Line 387: What do the authors mean by "lack of groundwater in these specific WRF-Hydro/Glacier simulations"?

12. Line 401-403: Why was Crocus not used to simulate the 14.7% glacierized area in Finseelvi?

13. Line 413: Where can the reader see that the streamflow significantly diverged?

14. Line 422: Please elaborate on model calibration in the methods section.

Technical comments:

Line 50: Please add "e.g.," to the list of citations.

Line 212 "time period"

Line 213 "do not"

Line 216: "were captured"

Line 236: Remove "surface" or change to "glacier surface mass balance"

Line 238: What does "(nve.no/hydrologi/bre)" mean?

Line 285: Please indicate which locations were used for measuring the summer mass balance.

Line 290: "redistribution of snow"?

Line 309: "stakes"

Lines 323 to 325: I suggest removing "slightly" since differences reach 20+%.

Figure 1: Please add a spatial scale.

Figure 2, bottom panel: It would be helpful to add the location of Finse, so that it's clearer where the station is relative to the study glacier.

---

## Referee Comment (RC2) · Anonymous Referee #2 · 20 Jul 2020

This paper presents the development of a new glacier component of the WRF-Hyrdo simulation platform. It relies on the multi-layer snowpack scheme Crocus that has been implemented in WRF-Hydro. Crocus is used to simulate in WRF-Hydro/Glacier the continuous evolution of snow and ice layers at pre-identified glacier points. In this paper, WRF-Hydro/Glacier is evaluated for the Hardangerjøkulen ice cap in Norway. Model outputs are compared to a large set of observations: (i) measurements of winter, summer and net glacier mass balance, (ii) snow depth measurements from a Ground Penetrating Radar (GPR), (iii) albedo measurements from MODIS and (iv) discharge measurements at two locations. The evaluation revealed improved performances compared to the default version of WRF-Hydro. In particular, the evolution of

surface albedo is better represented during the ablation season leading to better estimation of summer mass balance and improved discharge estimations for a partially glacierized catchment.

The new modelling system described in this study presents a large interest for the mountain hydrology community and constitutes an important improvement for WRF-Hydro. My main comments about the study concern (i) the downscaling of meteorological variables in WRF-Hydro, (ii) the evaluation of winter precipitation, (iii) the comparison between simulated and observer glacier mass balance and (iv) the impact of the parameterization that represent the effect of wind-induced snow transport on snowpack properties in Crocus. These questions need to be clarified prior to publication in HESS. They are listed below as general comments followed by more specific and technical comments.

General comments

1. In this study, the authors tested an offline configuration of WRF-Hydro/Glacier at 100-m resolution. This configuration is driven by an atmospheric forcing obtained with the WRF atmospheric model running at 1 km resolution. The downscaling from 1 km to 100 m corresponds to a simple bi-linear interpolation as explained at L 183-185. No correction as a function of the elevation difference between the 1-km grid and the 100-m grid is applied for example for temperature. Effect of slopes, aspects and shadowing on incoming shortwave radiation are not taken into account as well and wind speed is not corrected as a function of local topography. This leads to a "smooth" atmospheric forcing at 100-m resolution and ultimately to snowpack simulations that cannot capture the variability of snow accumulation and melt over the glacier as illustrated on Fig. 10 and 11. This absence of small scale variability is not only explained by non-simulated lateral redistribution processes in Crocus. Therefore, the absence of appropriate meteorological downscaling to sub-kilometre resolution in WRF-Hydro/Glacier should be at least discussed by the authors in terms of impact on simulated glacier mass balance.

[Figure]

2. In the paper, the evaluation of WRF winter precipitation at high-altitude exposed stations is influenced by a large wind-undercatch impacting the measurements of winter precipitation at these stations. This is well illustrated on Fig. 5. Station data were not corrected for wind-undercatch by met.no. As mentioned by the author, this limits the relevance of the comparison between model output and observations. However, winter precipitations are a key component of the glacier winter mass balance and it would be very interesting to propose in the paper an improved evaluation of WRF winter precipitation at high-altitude stations. Since wind speed measurements are available (at least at Finse), it would be very interesting if the authors could propose their own corrections of wind-undercatch using correction functions taken from the WMO SPICE project (https://www.wmo.int/pages/prog/www/IMOP/intercomparisons/SPICE/SPICE.html) and detailed information on the precipitation gauge used at Finse and at the other locations (type of gauge, type of shield, . . .). They could also quantify the uncertainties associated with these corrections. Another solution would be to use the reference precipitation data from the Haukeliseter experimental site who was the Norwegian site that contributed to the WMO SPICE project. In particular, Haukeliseter is equipped with a Double Fence precipitation gauge. High-quality precipitation data were collected at Haukeliseter during winter 2016/2017 (Schirle et al., 2019). They may be also available for the other winters covered by this study. Haukeliseter is located south of the Hardangervidda and east of Roldal and must be located within the WRF 1-km domain based on Fig 3.

3. The authors compared on Fig. 8 the simulated and observed winter, summer and net mass balance. However, they did not clearly explain in the current version of the paper (L 284-289) how the simulated winter and summer balances were computed. For the summer mass balance, did the authors extract the simulation results at the 3 to 5 locations used to compute the observed mass balance and then interpolated the results? The same question raises for the simulated winter mass balance. In addition, on Fig. 8, the elevation dependency of the observed winter mass balance (Fig 8a, d, g, j) and snow depth associated with the winter mass balance (Fig 8b, e, h, k) look

different. However, the authors explained at L 241-243 that the observed winter mass balance was derived from snow depth and the unique snow density measurements over the glacier. As a consequence, we would expect a similar elevation dependency between winter mass balance and snow depth. However, for example, in 2015 the observed snow depth showed a decrease between 1400 and 1600 m which is not present in the observed winter mass balance. This point should be clarified by the authors.

4. In the configuration of Crocus used by the authors in this study, two parameterizations are activated to represent the effect of wind-induced snow transport on snowpack properties: (i) a param. that simulates snow compaction and fragmentation of snow grains for surface layers during blowing snow events and (ii) a param. that computes mass loss due to blowing snow sublimation. At L 164-166, the authors insisted on the importance of these two parameterizations for accurate simulations of glacier mass balance. However, in the rest of the paper, the effects of these two parameterizations are never quantified. For example, how does the compaction parameterization affect the quality of the simulated snow density over the glacier? In addition, the authors suggest that the blowing sublimation parameterization in Crocus explain the spatial variability in snow depth and SWE over the glacier (L 315-320). However, it is not clear that it can explain the local differences in snow depth and SWE. Indeed, the atmospheric forcing driving Crocus in WRH-Hydro/Glacier is rather smooth (cf General comment 1) and may not create a large variability of blowing snow sublimation from one grid cell to another on the 100-m grid. I recommend the authors to compare the results of simulations with and without the blowing sublimation parameterization. One winter would be certainly sufficient.

Specific comments

Abstract L 16: The transition between the first and the second sentence of the abstract is not clear at the moment. It would be interesting to add here a sentence that explains why Crocus is suitable for glacier modelling. The fact that a multi-layer snowpack scheme can be used for direct surface mass balance simulation of glacier is not necessarily clear for a reader who is not familiar with Crocus.

Abstract L 17-18: The two different resolutions for WRH-Hdyro and WRF-Hydro/Glacier are rather confusing. Maybe mention atmospheric simulation on one hand and offline surface simulations on the other hand.

Abstract L 19-20: A sentence is missing in the abstract to explain that the study is carried out over a glacier in southern Norway. It is only mentioned in the title.

P2 L 38-40: I recommend the authors to add one or two citations for this sentence.

P2 L 48-49: the authors should add relevant citations for the application of statistical downscaling for glacier studies. In addition, physically based downscaling methods have also been developed to obtain better distributed meteorological forcing in regions of complex terrain (e.g., Jarosch et al., 2012; Fiddes et al., 2014).

P2 L 54-55: I agree with authors that regional "atmosphere-only" models do not typically include a detailed representation of glacier and their impact on streamflow generation. However, atmospheric forcing from these regional models are often used to drive more detailed models such snowpack models or hydrological models in offline mode for impact studies. I recommend to the authors to mention this approach in the introduction.

P 2 L 59: The authors should better explain this "link" between an atmospheric model and a detailed hydrological model. Indeed, are they talking about offline simulations or online simulations with feedbacks of the surface on the atmosphere dynamics? When used in offline mode as it is the case in this study, WRH-Hydro does not really differ from a more classic hydrological model such as MESH (Pietroniro et al., 2007) that takes its atmospheric forcing from an external system (GCM, RCM, . . ..), downscale them and use them to drive hydrological simulations.

P3 L 75 and P4 L 103: Crocus has a user-defined value for the maximal number

of snow layers. The value of 50 corresponds to the default value in the model. I recommend the authors to clearly explain it. Especially since they then use 40 layers in their implementation of Crocus in WRF-Hydro.

P 3 L 76-78: Note that the default version of Crocus still uses a classic bucket-type approach for liquid water percolation in the snowpack. It does not solve Richards equations and ignore preferential flows contrary to the SNOWPACK model (Wever et al., 2015, 2016). Preliminary developments have been tested in Crocus but are not available in the version of Crocus used in this paper (D'Amboise et al., 2017).

P 3 L 78-79: the reference to the paper by Reveillet et al. (2018) is missing. I also recommend the authors to refer to the work of Gerbaux et al (2005) which used the original version of Crocus.

P4 L 98-99: This sentence is not clear and should be rephrased. Indeed, the "age of snow" is not directly used in the prognostic equations for the time evolution of microstructural variables In Crocus. It is used in the albedo parameterization to compute the decay of albedo in the UV and visible band and will indirectly impact the evolution of microstructural variables. Compaction is not also directly impacting the evolution of microstructural variables.

P4 L 100-101: A description of the distinction between snow and ice albedo is missing in this paragraph. It would be interesting if the authors could use a similar description as Reveillet et al. (2018) (see Sect. 3.1.1 of this paper). The authors should also mention how the aerodynamic roughness are treated for snow and ice surfaces.

P 4 L 101-102: the justification "Due to the prognostic calculation of snow grain properties, . . ." is not clear and should be rephrased.

P4 L 106-108: did the authors consider using Crocus to represent snow over land in NOAH-MP?

P4 L 109-111: Are the authors imposing a constant-temperature of 0 degC for the

ground below the glacier or are they imposing a zero-heat flux between the deepest Crocus layer and the ground below as in Gerbaux et al. (2005)?

P5 L 126-137: It would be useful to have this paragraph at the beginning of Section 2 so that the reader could better understand how glaciers are represented in the default version of WRF-Hydro and what are the associated challenges. This would provide a very relevant justification for the use of Crocus over glaciers.

P 5 L 149-150: initializing all the layers with pure ice may influence the accuracy of the snowpack simulations when snow starts to fall on the glacier. Indeed, as mentioned by the authors, it forces the model to merge layers as soon as new snow is added. Instead, the initialization proposed by Revuelto et al. (2018) used 6 initial layers to represent the ice and a maximal number of layers set to 50 so that new snow layers can be created as soon as snow is falling on the glacier without forcing the immediate merging of ice layers underneath. This difference should be mentioned in the paper and its impact briefly discussed.

P 5 L 150: what are the snow grain properties for ice layers used in the paper?

P 6 L 162-167: I recommend the authors to move this paragraph at the end of Section 2.1 when they are mentioning the absence of lateral snow redistribution due to wind in Crocus. This would help the reader to better understand how WRH-Hydro/Glacier accounts for the impact of wind-induced snow transport.

P 6 L 167: it would be interesting to add here a brief description of the configuration of the routing model. Is the routing simulated at the same resolution as Crocus (100 m)? How were derived the routing parameters? It is also important to mention here that no calibration was applied to the routing model.

P 6 L 170: a table summarizing information about the simulation domains (size, number of grid points, ....) would be useful. This table could also include information on the 100-m simulation domain.

P6 L 171: what is the height above the ground of the lowest atmospheric prognostic level?

P 7 L 183: how is computed the rain/snow partitioning in WRF-Hydro/Glacier?

P7 L 188: which method it used to obtain the simulated values at the location of the AWS from the output of WRF 1 km? Nearest-neighbor interpolation, bilinear interpolation? Did they author consider the elevation difference between the simulated station elevation and the actual station elevation when selecting the stations used for model evaluation?

P8 L 216-217: At this stage of the paper, it is not clear that the wind direction and the wind speed are well simulated by WRF. This information is mainly confirmed by Figure 7 which is presented later.

P 9 L 239: at how many locations is measured the mass balance of the glaciers?

P 9 L 247-249: it is not clear why the authors did not use the actual dates when the observations were gathered to compute the simulated mass balance, contrary to Vionnet et al. (2019). Could they add a justification?

P 9 L 256-257: it is not clear to me how a device transported at 15-20 km/h (approx. 4.1-5.5 m/s) with a sampling interval of 1 s can generate a 1-m spacing between datapoints. Note that I am not familiar with GPR postprocessing, so ignore my comment if the 1-m spacing is obtained from data postprocessing. P 10 L 273: snow albedo in the version of Crocus used in this paper also depends on the snow age. Indeed, the snow age is used to parameterize the influence of light absorbing impurities on snow albedo in the UV and visible range. A more recent version of Crocus explicitly simulates the direct and indirect radiative impacts of light-absorbing impurities in snow (Tuzet et al., 2017). P10 L 277: what is the size of the two river catchments considered in the study?

P 12 L 335-347: Figure 10 shows that the GPR provided an excellent coverage of the glacier for winter 2017. It would be very interesting to use these data to compare the

simulated and observed elevation-snow depth relation and to compare the simulated and observed variability of snow depth per elevation bands. This would complement well Fig. 8.

P 12 L 349: the evaluation of the simulated albedo is based on a comparison at two representative points selected over the glacier. This choice should be justified in the method section when the authors are describing the MODIS albedo data. Indeed, at L 273-276 the authors mention that MODIS data are available at 500 m resolution over the glacier. This suggest that the full MODIS albedo dataset would be used for model evaluation.

P 15 L 440: I am not sure that the term "excellent" can be used to qualify the WRF winter precipitation. Indeed, the authors have shown that they cannot be directly evaluated due to wind-undercatch at the high-elevation stations which are the most relevant for this study. Maybe the authors should make here the link between the WRF winter precipitation and the winter mass balance simulated by WRF-Hydro/Glacier.

P 16 L 454-455: the author should mention here that the model at 100-m resolution cannot capture the spatial variability of the snowpack on the glacier due (i) the absence of proper meteorological downscaling and (ii) the non-representation of lateral snow redistribution. It is mentioned later in the conclusion as a perspective, but it should appear in the bullet list containing the major conclusions of the study as well.

P 16 L 467: information on the code availability would be very interesting for the readers. It could potentially serve a basis for Crocus users who want to implement the model in their own land surface model.

Technical comments

Text

P2 L 35: "ablation-season"?

P2 L 38: a parenthesis ")" is missing

P4 L 94: I recommend the authors to use "initially developed" instead of "specifically developed" Indeed, since the 90's, Crocus has been used in many different applications (https://www.umr-cnrm.fr/spip.php?article268&lang=en).

P4 L 99: use "metamorphism" instead of "metamorphosis"

P 4 L 100: Ấń Vionnet Âż instead of "Vionett".

P5 L 144: It is surprising to have a Section 2.1 without a Section 2.2.

P5 L 147: use kg mˆ{-3} instead of kg/m3

P 5 L 208: use m sˆ{-1} instead of m/s

P15 L 427: maybe add "and ice" after "physical properties of snow"

P21 L 598: the reference to Willemet (2008) is not included in the text.

Table

Table 1: what is the signification of the values appearing in bold in the table?

Figure

Figure 7: different scales are used for the observed and simulated wind data on the wind rose. Using the same scale would allow q more direct comparison between model and observations. In addition, could the authors provide simple errors metrics such as bias and RMSE for wind speed at Finse?

References

D'Amboise, C. J. L., Müller, K., Oxarango, L., Morin, S., and Schuler, T. V.: Implementation of a physically based water percolation routine in the Crocus/SURFEX (V7.3) snowpack model, Geosci. Model Dev., 10, 3547–3566, https://doi.org/10.5194/gmd-10-3547-2017, 2017.

Jarosch, A. H., Anslow, F. S., & Clarke, G. K. (2012). High-resolution precipitation and

temperature downscaling for glacier models. Climate Dynamics, 38(1-2), 391-409.

Fiddes, J., & Gruber, S. (2014). TopoSCALE v. 1.0: downscaling gridded climate data in complex terrain. Geoscientific Model Development, 7(1), 387-405.

Pietroniro, A., Fortin, V., Kouwen, N., Neal, C., Turcotte, R., Davison, B., ... & Pellerin, P. (2007). Development of the MESH modelling system for hydrological ensemble forecasting of the Laurentian Great Lakes at the regional scale. Hydrol. Earth Syst. Sci, 11, 1279-1294.

Réveillet, M., Six, D., Vincent, C., Rabatel, A., Dumont, M., Lafaysse, M., Morin, S., Vionnet, V., and Litt, M.: Relative performance of empirical and physical models in assessing the seasonal and annual glacier surface mass balance of Saint-Sorlin Glacier (French Alps), The Cryosphere, 12, 1367–1386, https://doi.org/10.5194/tc-12-1367-2018, 2018.

Schirle, C. E., S. J. Cooper, M. A. Wolff, C. Pettersen, N. B. Wood, T. S. L'Ecuyer, T. Ilmo, and K. Nygård, 2019: Estimation of Snowfall Properties at a Mountainous Site in Norway Using Combined Radar and In Situ Microphysical Observations. J. Appl. Meteor. Climatol., 58, 1337–1352, https://doi.org/10.1175/JAMC-D-18-0281.1.

Tuzet, F., Dumont, M., Lafaysse, M., Picard, G., Arnaud, L., Voisin, D., Lejeune, Y., Charrois, L., Nabat, P., and Morin, S.: A multilayer physically based snowpack model simulating direct and indirect radiative impacts of light-absorbing impurities in snow, The Cryosphere, 11, 2633–2653, https://doi.org/10.5194/tc-11-2633-2017, 2017.

Wever, N., Schmid, L., Heilig, A., Eisen, O., Fierz, C., & Lehning, M. (2015). Verification of the multi-layer SNOWPACK model with different water transport schemes. The Cryosphere, 9(2), 2655-2707.

Wever, N., Würzer, S., Fierz, C., & Lehning, M. (2016). Simulating ice layer formation under the presence of preferential flow in layered snowpacks. The Cryosphere, 10, 2731-2744.

---

## Author Comment (AC1) · 9 Oct 2020

**Response to Interactive comment by Anonymous Referee #2**

First, we want to thank the Referee for the review of our manuscript. The excellent comments and suggestions have greatly helped improve our paper. We have tried as best we can to respond to the comments and we have (or will) follow most of the suggestions. The original reviews are in black, and our responses are in blue.

This paper presents the development of a new glacier component of the WRF-Hyrdo simulation platform. It relies on the multi-layer snowpack scheme Crocus that has been implemented in WRF-Hydro. Crocus is used to simulate in WRF-Hydro/Glacier the continuous evolution of snow and ice layers at pre-identified glacier points. In this paper, WRF-Hydro/Glacier is evaluated for the Hardangerjøkulen ice cap in Nor- way. Model outputs are compared to a large set of observations: (i) measurements of winter, summer and net glacier mass balance, (ii) snow depth measurements from a Ground Penetrating Radar (GPR), (iii) albedo measurements from MODIS and (iv) discharge measurements at two locations. The evaluation revealed improved performances compared to the default version of WRF-Hydro. In particular, the evolution of surface albedo is better represented during the ablation season leading to better estimation of summer mass balance and improved discharge estimations for a partially glacierized catchment.

The new modelling system described in this study presents a large interest for the mountain hydrology community and constitutes an important improvement for WRF- Hydro. My main comments about the study concern (i) the downscaling of meteorolog- ical variables in WRF-Hydro, (ii) the evaluation of winter precipitation, (iii) the compar- ison between simulated and observer glacier mass balance and (iv) the impact of the parameterization that represent the effect of wind-induced snow transport on snow- pack properties in Crocus. These questions need to be clarified prior to publication in HESS. They are listed below as general comments followed by more specific and technical comments.

General comments

1. In this study, the authors tested an offline configuration of WRF-Hydro/Glacier at 100-m resolution. This configuration is driven by an atmospheric forcing obtained with the WRF atmospheric model running at 1 km resolution. The downscaling from 1 km to 100 m corresponds to a simple bilinear interpolation as explained at L 183-185. No correction as a function of the elevation difference between the 1km grid and the 100 m grid is applied for example for temperature. Effect of slopes, aspects and shadowing on incoming shortwave radiation are not taken into account as well and wind speed is not corrected as a function of local topography. This leads to a "smooth" atmospheric forcing at 100-m resolution and ultimately to snowpack simulations that cannot capture the variability of snow accumulation and melt over the glacier as illustrated on Fig. 10 and 11. This absence of small scale variability is not only explained by non-simulated lateral redistribution processes in Crocus. Therefore, the absence of appropriate meteorological downscaling to sub-kilometre resolution in WRF-Hydro/Glacier should be at least discussed by the authors in terms of impact on simulated glacier mass

It is true that we are missing some small-scale variability. However, it should be noted that the area of interest is on a plateau with large open areas (see picture below). Yes, there are some shading, which is mainly prevalent at very low sun angles.  We therefore believe that the bilinear interpolation from 1 km to 100 m is not of a too large concern for this specific case. In a rugged terrain, such as in the Alps, the bilinear interpolation might be a larger problem. The slope of the glacier is not that steep either, with and increase with about 100 m per km.

We added this text in the manuscript

*We note that we did not account for variability in terrain in the re-gridding process. Thus the atmospheric forcing is still "smooth" as regards to a 100 m grid. However, the region of interest (Hardangerjøkulen and surrounding*

*terrain) is an open mostly flat area and we therefore believe that for this specific case, disregarding the variation in terrain does not have much impact on mass balance calculations.*

[Figure]

2. In the paper, the evaluation of WRF winter precipitation at high-altitude exposed stations is influenced by a large wind-undercatch impacting the measurements of winter precipitation at these stations. This is well illustrated on Fig. 5. Station data were not corrected for wind undercatch by met.no. As mentioned by the author, this limits the relevance of the comparison between model output and observations. However, winter precipitations are a key component of the glacier winter mass balance and it would be very interesting to propose in the paper an improved evaluation of WRF winter precipitation at high-altitude stations. Since wind speed measurements are available (at least at Finse), it would be very interesting if the authors could propose their own corrections of wind-undercatch using correction functions taken from the WMO SPICE project (https://www.wmo.int/pages/prog/www/IMOP/intercomparisons/SPICE/SPICE.html) and detailed information on the precipitation gauge used at Finse and at the other locations (type of gauge, type of shield, . . .). They could also quantify the uncertainties associated with these corrections. Another solution would be to use the reference precipitation data from the Haukeliseter experimental site who was the Norwegian site that contributed to the WMO SPICE project. In particular, Haukeliseter is equipped with a Double Fence precipitation gauge. High-quality precipitation data were collected at Haukeliseter during winter 2016/2017 (Schirle et al., 2019). They may be also available for the other winters covered by this study. Haukeliseter is located south of the Hardangervidda and east of Roldal and must be located within the WRF 1-km domain based on Fig 3.

Indeed, Haukeliseter is located in the WRF 1-km domain and it is a great suggestion to use data from this location to evaluate modeled winter precipitation. We have data from 2014 and 2015, and we will look into obtaining the data from 2016/2017 as well. As regards to conducting a wind correction at Finse based on findings from the SPICE project, this is something we will look into. The precipitation gauge at Finse is a single – Alter -shield.

Although the SPICE project has allowed for suggested transfer functions to correct for the under-cathment we note this finding in Smith et al 2020 "Evaluation of the WMO-SPICE transfer functions for adjusting the wind bias in solid precipitation measurements" Hess that they state:

"Although the application of transfer functions is necessary to mitigate wind bias in solid precipitation measurements, especially at windy sites and for unshielded gauges, the inconsistency in the performance metrics among sites suggests that the functions be applied with caution."

Just as Haukeliseter is a windy location, so if Finse. Thus it is worth looking into using published transfer functions from Haukeliseter. Thus we will take the suggestion into consideration while updating the manuscript, but with the notion that transfer functions should be applied with caution.

3. The authors compared on Fig. 8 the simulated and observed winter, summer and net mass balance. However, they did not clearly explain in the current version of the paper (L 284-289) how the simulated winter and summer balances were computed. For the summer mass balance, did the authors extract the simulation results at the 3 to 5 locations used to compute the observed mass balance and then interpolated the results? The same question raises for the simulated winter mass balance. In addition, on Fig. 8, the elevation dependency of the observed winter mass balance (Fig 8a, d, g, j) and snow depth associated with the winter mass balance (Fig 8b, e, h, k) look different. However, the authors explained at L 241-243 that the observed winter mass balance was derived from snow depth and the unique snow density measurements over the glacier. As a consequence, we would expect a similar elevation dependency between winter mass balance and snow depth. However, for example, in 2015 the observed snow depth showed a decrease between 1400 and 1600 m which is not present in the observed winter mass balance. This point should be clarified by the authors.

For the modeled winter and summer balance we take the average balance for each grid point within a certain elevation level (40 m)

We have added this sentence:

*The modeled winter (and summer) balance is plotted as averages of all grid points over Rembesdalskåka within intervals of 40 m*

Regarding the difference in plottet SWE versus height, the SWE values were taken from the values shared in the reports by Andreassen et al., (2016), (2017), (2019) and Kjøllmoen et al., (2018) (see Figure 1) while the snowdepth was taken from the original data. One of the authors of these reports stated that the data in the reports were an arithmetic mean of height and SWE withing 50 m as a base for subjective smoothed SWE curves. In the manuscript we will be consistent with which data we use for the SWE and snowdepth (if we end up showing both). Note that there are only 6 observation points below 1600 m, so the variations are larger in this region (as seen in the snowdepth curve in the manuscript.

[Figure]

**Figure 6-11**
Specific (left) and volume (right) winter, summer and annual balance at Rembesdalskåka in 2015.
Specific summer balance at five stakes is shown (o).

**Figure 1.** Example of SWE versus height from the Reports by Andreassen et al., (2016), (2017), (2019) and Kjøllmoen et al., (2018)

4. In the configuration of Crocus used by the authors in this study, two parameterizations are activated to represent the effect of wind-induced snow transport on snowpack properties: (i) a param. that simulates snow compaction and fragmentation of snow grains for surface layers during blowing snow events and (ii) a param. that computes mass loss due to blowing snow sublimation. At L 164-166, the authors insisted on the importance of these two parameterizations for accurate simulations of glacier mass balance. However, in the rest of the paper, the effects of these two parameterizations are never quantified. For example, how does the compaction parameterization affect the quality of the simulated snow density over the glacier? In addition, the authors suggest that the blowing sublimation parameterization in Crocus explain the spatial variability in snow depth and SWE over the glacier (L 315-320). However, it is not clear that it can explain the local differences in snow depth and SWE. Indeed, the atmospheric forcing driving Crocus in WRH-Hydro/Glacier is rather smooth (cf General comment 1) and may not create a large variability of blowing snow sublimation from one grid cell to another on the 100-m grid. I recommend the authors to compare the results of simula- tions with and without the blowing sublimation parameterization. One winter would be certainly sufficient.

We actually did include a section describing the sublimation of blowing snow issue and had a figure. However, to reduce the paper and number of figures, the editor suggested we initially remove this section. We will go over the manuscript again and either add the figure or rewrite the text and give some examples.

The modeled wind did have some variations over the glacier on the large scale with the north east part of the glacier experiencing the highest windspeeds (Figure 2). This is also where you have the largest differences in mass balance (see figure 3 where the left plot show without sublimation due to wind drift and the right shows with).

**2017**

[Figure]

Windspeed (m/s)

0   1   2   3   4   5   6

**Figure 2** Average wind speed winter 2017

[Figure]

**Figure 3** Modeled versus observed snow thickness. Left plot shows without sublimation of blowing snow and right plot shows with sublimation of blowing snow.

Specific comments

Abstract L 16: The transition between the first and the second sentence of the ab- stract is not clear at the moment. It would be interesting to add here a sentence that explains why Crocus is suitable for glacier modelling. The fact that a multi-layer snowpack scheme can be used for direct surface mass balance simulation of glacier is not necessarily clear for a reader who is not familiar with Crocus.

We have moved the part of the abstract that describes why Crocus is suitable for glacier model further up in the abstract.

Abstract L 17-18: The two different resolutions for WRH-Hdyro and WRF-Hydro/Glacier are rather confusing. Maybe mention atmospheric simulation on one hand and offline surface simulations on the other hand.

The sentence now reads:

"WRF atmospheric model simulations were downscaled to 1 km grid spacing to provide meteorological forcing data to the WRF-Hydro/Glacier system at 100 m grid spacing for surface simulation."

Abstract L 19-20: A sentence is missing in the abstract to explain that the study is carried out over a glacier in southern Norway. It is only mentioned in the title.

The sentence now reads:

"To evaluate the new system (WRF-Hydro/Glacier) over a glacier in Southern Norway,"

P2 L 38-40: I recommend the authors to add one or two citations for this sentence.

We are adding the citation of Ayala, A, Pellicciotti, F, and Shea, JM (2015), Modeling 2 m air temperatures over mountain glaciers: Exploring the influence of katabatic cooling and external warming. J. Geophys. Res. Atmos., 120, 3139– 3157. doi: 10.1002/2015JD02313 and citations within this paper.

P2 L 48-49: the authors should add relevant citations for the application of statistical downscaling for glacier studies. In addition, physically based downscaling methods have also been developed to obtain better distributed meteorological forcing in regions of complex terrain (e.g., Jarosch et al., 2012; Fiddes et al., 2014).

The papers by Machguth et al., 2009; Kotlarski et al., 2010; Van Pelt et al., 2012 already describe the statistical downscaling they are conducting; thus we do not think there is a need to add additional citations for statistical downscaling for glacier studies.

We will add the suggested citations of Jarosch et al. and Fiddes et al.

P2 L 54-55: I agree with authors that regional "atmosphere-only" models do not typically include a detailed representation of glacier and their impact on streamflow generation. However, atmospheric forcing from these regional models are often used to drive more detailed models such snowpack models or hydrological models in offline mode for impact studies. I recommend to the authors to mention this approach in the introduction.

We agree, and we will add a few sentences regarding this topic

P 2 L 59: The authors should better explain this "link" between an atmospheric model and a detailed hydrological model. Indeed, are they talking about offline simulations or online simulations with feedbacks of the surface on the atmosphere dynamics? When used in offline mode as it is the case in this study, WRH-Hydro does not really differ from a more classic hydrological model such as MESH (Pietroniro et al., 2007) that takes its atmospheric forcing from an external system (GCM, RCM, . . ..), downscale them and use them to drive hydrological simulations.

This is correct, and we are reformulating

P3 L 75 and P4 L 103: Crocus has a user-defined value for the maximal number of snow layers. The value of 50 corresponds to the default value in the model. I recommend the authors to clearly explain it. Especially since they then use 40 layers in their implementation of Crocus in WRF-Hydro.

Sentences now read:

"It is a physically-based model, in which the snow depth can be divided into a user defined maximum levels and where the default maximum is 50 layers."

"In the Crocus model, it is possible to divide the snow into a user defined maximum numbers of dynamically evolving layers."

P 3 L 76-78: Note that the default version of Crocus still uses a classic bucket-type approach for liquid water percolation in the snowpack. It does not solve Richards equations and ignore preferential flows contrary to the SNOWPACK model (Wever et al., 2015, 2016). Preliminary developments have been tested in Crocus but are not available in the version of Crocus used in this paper (D'Amboise et al., 2017).

Thanks for this comment. We are removing the statement of retention since Crocus still uses a classic bucket-type approach.

P 3 L 78-79: the reference to the paper by Reveillet et al. (2018) is missing. I also recommend the authors to refer to the work of Gerbaux et al (2005) which used the original version of Crocus.

Thank you for drawing attention to Reveillet et al. (2018). We have added this paper to our citations:

"The Crocus model was first used for glacier mass balance by Gerbaux et al (2005) and recently used for glacier surface mass balance studies within the French Surfex model by Reveillet et al. (2018), Revuelto et al. (2018) and Vionnet et al. (2019). "

P4 L 98-99: This sentence is not clear and should be rephrased. Indeed, the "age of snow" is not directly used in the prognostic equations for the time evolution of microstructural variables In Crocus. It is used in the albedo parameterization to compute the decay of albedo in the UV and visible band and will indirectly impact the evolution of microstructural variables. Compaction is not also directly impacting the evolution of microstructural variables.

Thank you for the clarification. We have removed "age of the snow"

P4 L 100-101: A description of the distinction between snow and ice albedo is missing in this paragraph. It would be interesting if the authors could use a similar description as Reveillet et al. (2018) (see Sect. 3.1.1 of this paper). The authors should also mention how the aerodynamic roughness are treated for snow and ice surfaces.

This is a good suggestion and we will add some more information regarding the distinction between ice and snow albedo and the values for snow and ice roughness (we used the same ratio between ice and snow as in the original code). For ice albedo we ended up using the values used in Surfex (and not the ones from Gerbaux et al, as used in Reveillet.

P 4 L 101-102: the justification "Due to the prognostic calculation of snow grain properties, . . ." is not clear and should be rephrased.

This sentence is begin removed

P4 L 106-108: did the authors consider using Crocus to represent snow over land in NOAH-MP?

Currently the fluxes between the glacier surface and the ground is not incorporated (the engineering of passing fluxes in SURFEX compared to Noah-MP is somewhat different and is not yet incorporated). Under glacier surface it is assumed that the temperature is the same as soil and therefore there are no fluxes added. We would like to incorporate the fluxes between Crocus and the soil soon to be able to use Crocus over the entire domain.

P4 L 109-111: Are the authors imposing a constant-temperature of 0 degC for the ground below the glacier or are they imposing a zero-heat flux between the deepest Crocus layer and the ground below as in Gerbaux et al. (2005)?

We are imposing a constant-temperature of 0 degC, as the heat-fluxes between the soil and surface is not yet incorporated. We removed the statement that there are no fluxes of heat pass trough and instead state that no fluxes have been incorporated between the glacier and the ground below.

P5 L 126-137: It would be useful to have this paragraph at the beginning of Section 2 so that the reader could better understand how glaciers are represented in the default version of WRF-Hydro and what are the associated challenges. This would provide a very relevant justification for the use of Crocus over glaciers.

Agree. We will move this paragraph (or parts of it to the beginning of Section 2)

P 5 L 149-150: initializing all the layers with pure ice may influence the accuracy of the snowpack simulations when snow starts to fall on the glacier. Indeed, as mentioned by the authors, it forces the model to merge layers as soon as new snow is added. Instead, the initialization proposed by Revuelto et al. (2018) used 6 initial layers to represent the

ice and a maximal number of layers set to 50 so that new snow layers can be created as soon as snow is falling on the glacier without forcing the immediate merging of ice layers underneath. This difference should be mentioned in the paper and its impact briefly discussed.

The reviewer is correct, and in future studies the glacier should be initialized in less layers than the maximum layers. Note, we allow 2 months for spin-up time before using October 1st for calculating winter balance. At this time the glacier has started to merge. By January 1 the glacier is merged down to about 6-8 layers and remain as such for the rest of the simulation. Will add a short discussion regarding this point in the paper.

P 5 L 150: what are the snow grain properties for ice layers used in the paper?

We use the Brun 92 scheme. We are not sure where to explicitly find the grain properties for ice layers in the code, but Snowgrain1=99 and Snowgrain2 = 0.003415 in our output files.

P 6 L 162-167: I recommend the authors to move this paragraph at the end of Section 2.1 when they are mentioning the absence of lateral snow redistribution due to wind in Crocus. This would help the reader to better understand how WRH-Hydro/Glacier accounts for the impact of wind-induced snow transport.

We have followed the suggestion and moved the paragraph.

P 6 L 167: it would be interesting to add here a brief description of the configuration of the routing model. Is the routing simulated at the same resolution as Crocus (100 m)? How were derived the routing parameters? It is also important to mention here that no calibration was applied to the routing model.

We will add a brief description on the routing model (WRF-Hydro). However, instead of adding this description in section 2.1, Crocus initialization, we will add it to section 3, Experimental Description. And yes, routing is at 100m, the same resolution as Crocus.

P 6 L 170: a table summarizing information about the simulation domains (size, number of grid points, ....) would be useful. This table could also include information on the 100-m simulation domain.

This was suggested by the other reviewer as well, and we will add a table summarizing the simulations

P6 L 171: what is the height above the ground of the lowest atmospheric prognostic level?

The lowest model level height is about 25 m

P 7 L 183: how is computed the rain/snow partitioning in WRF-Hydro/Glacier?

The rain/snow partitioning in WRF-Hydro/Glacier is the same as one of the Noah-MP options, which is called the Jordan Scheme. Snowfraction (SF) = 0 above 2.5C.  SF = 0.6 between 2 and 2.5C. SF = 1 below 0.5C. Between 0.5 and 2C, SF is a linear function. We will add a description regarding the rain/snow partitioning.

P7 L 188: which method it used to obtain the simulated values at the location of the AWS from the output of WRF 1 km? Nearest-neighbor interpolation, bilinear interpolaion? Did they author consider the elevation difference between the simulated station elevation and the actual station elevation when selecting the stations used for model evaluation?

We used nearest neighbor. We did not consider elevation difference. For most of these stations, the model elevation in the grid point is higher than the observation elevation, and at times about 150 m higher. The Finse elevation difference between model grid point and actual elevation is only 16 m.

P8 L 216-217: At this stage of the paper, it is not clear that the wind direction and the wind speed are well simulated by WRF. This information is mainly confirmed by Figure 7 which is presented later.

We will move up the discussion regarding wind direction and speed at Finse. However, we might remove the figure (and only describe the results) due to suggestions from the other reviewer to reduce the number of figures.

P 9 L 239: at how many locations is measured the mass balance of the glaciers?

For the winter balance, all the green dots in Figure 1 is used (about 60 locations). For the Summer balance, there are 4 locations.

P 9 L 247-249: it is not clear why the authors did not use the actual dates when the observations were gathered to compute the simulated mass balance, contrary to Vionnet et al. (2019). Could they add a justification?

For the winter balance we did. For the summer balance we did not because the observations are adjusted for summer surface, not when observations are conducted. We will clarify the text.

P 9 L 256-257: it is not clear to me how a device transported at 15-20 km/h (approx. 4.1-5.5 m/s) with a sampling interval of 1 s can generate a 1-m spacing between data- points. Note that I am not familiar with GPR postprocessing, so ignore my comment if the 1-m spacing is obtained from data postprocessing.

Thanks for point out this error. We checked our survey coordinates, and there is indeed a GPR trace every ~1 m; it is instead the GPS system which records a positional datapoint every 1 s. The wording has been corrected in the manuscript:

*"The GPR systems were towed behind a snowmobile, at ~15-20 km/h. The interval between successive GPR recordings is ~0.2 s, giving a distance sampling interval of ~ 1 m (regularized to exactly 1 m in processing). A GPS system was also mounted on the snowmobile, recording positions every 1-2 s, to locate the GPR recordings"*

P 10 L 273: snow albedo in the version of Crocus used in this paper also depends on the snow age. Indeed, the snow age is used to parameterize the influence of light absorbing impurities on snow albedo in the UV and visible range. A more recent version of Crocus explicitly simulates the direct and indirect radiative impacts of light-absorbing impurities in snow (Tuzet et al., 2017).

We will include this information either on the line referred to in this comment, or earlier in the manuscript when discussing the albedo

P10 L 277: what is the size of the two river catchments considered in the study?

The catchment size of Fineseelvi is about 16 km$^2$ and Middalselvi is about 12 km$^2$. We will add this information in the manuscript.

P 12 L 335-347: Figure 10 shows that the GPR provided an excellent coverage of the glacier for winter 2017. It would be very interesting to use these data to compare the simulated and observed elevation-snow depth relation and to compare the simulated and observed variability of snow depth per elevation bands. This would complement well Fig. 8.

We will consider this suggestion. However, we have gotten several suggestions to reduce the number of figures (by the editor and the other reviewer)

P 12 L 349: the evaluation of the simulated albedo is based on a comparison at two representative points selected over the glacier. This choice should be justified in the method section when the authors are describing the MODIS

albedo data. Indeed, at L 273-276 the authors mention that MODIS data are available at 500 m resolution over the glacier. This suggest that the full MODIS albedo dataset would be used for model evaluation.

This is a good point. We will add a clarification in the methods section about how we use MODIS to compare with the model.

P 15 L 440: I am not sure that the term "excellent" can be used to qualify the WRF winter precipitation. Indeed, the authors have shown that they cannot be directly evaluated due to wind undercatch at the high-elevation stations which are the most relevant for this study. Maybe the authors should make here the link between the WRF winter precipitation and the winter mass balance simulated by WRF-Hydro/Glacier.

We will rephrase the sentence, reflecting the reviewer's comment.

P 16 L 454-455: the author should mention here that the model at 100-m resolution cannot capture the spatial variability of the snowpack on the glacier due (i) the absence of proper meteorological downscaling and (ii) the non-representation of lateral snow redistribution. It is mentioned later in the conclusion as a perspective, but it should appear in the bullet list containing the major conclusions of the study as well.

We will mention the non-representation of lateral snow redistribution. Regarding the absence of proper meteorological downscaling, since the region is not very complex, we believe for this study, the downscaling is proper.

P 16 L 467: information on the code availability would be very interesting for the readers. It could potentially serve a basis for Crocus users who want to implement the model in their own land surface model.

Currently we are implementing the Crocus into the National Water Model (a version of WRF-Hydro). So at the time just before publishing we plan on deciding where to add the code for users to use and refer to this site.

Technical comments
Text
P2 L 35: "ablation-season"? done
P2 L 38: a parenthesis ")" is missing  done

P4 L 94: I recommend the authors to use "initially developed" instead of "specifically developed" Indeed, since the 90's, Crocus has been used in many different applications (https://www.umr-cnrm.fr/spip.php?article268&lang=en). done

P4 L 99: use "metamorphism" instead of "metamorphosis" done
P 4 L 100: Ân´ Vionnet Âz˙ instead of "Vionett". done
P5 L 144: It is surprising to have a Section 2.1 without a Section 2.2. Agreed, and we will move the text before Section 2.1 into a new Section 2.1 and moce Section 2.1 to Section 2.2 We will do the same with Section 3.1 (since there are no section 3.2

P5 L 147: use kg mˆ{-3} instead of kg/m3 done
P 5 L 208: use m sˆ{-1} instead of m/s done
P15 L 427: maybe add "and ice" after "physical properties of snow" done
P21 L 598: the reference to Willemet (2008) is not included in the text. This is removed
Table

Table 1: what is the signification of the values appearing in bold in the table? The bold font represents the case with the highest correlation for each year and each location. We will mention this in the text

Figure 7: different scales are used for the observed and simulated wind data on the wind rose. Using the same scale would allow q more direct comparison between model and observations. In addition, could the authors provide simple errors metrics such as bias and RMSE for wind speed at Finse?

We completely agree. The author tried to make the scale the same, but with the software used, it was not possible. According to the other reviewer, we might remove this figure. If now, we will try and create a figure with similar scale. We will add bias and RMSE for the wind speed.

References

D'Amboise, C. J. L., Müller, K., Oxarango, L., Morin, S., and Schuler, T. V.: Implemen- tation of a physically based water percolation routine in the Crocus/SURFEX (V7.3) snowpack model, Geosci. Model Dev., 10, 3547–3566, https://doi.org/10.5194/gmd- 10-3547-2017, 2017.

Jarosch, A. H., Anslow, F. S., & Clarke, G. K. (2012). High-resolution precipitation and temperature downscaling for glacier models. Climate Dynamics, 38(1-2), 391-409.

Fiddes, J., & Gruber, S. (2014). TopoSCALE v. 1.0: downscaling gridded climate data in complex terrain. Geoscientific Model Development, 7(1), 387-405.

Pietroniro, A., Fortin, V., Kouwen, N., Neal, C., Turcotte, R., Davison, B., ... & Pellerin, P. (2007). Development of the MESH modelling system for hydrological ensemble fore- casting of the Laurentian Great Lakes at the regional scale. Hydrol. Earth Syst. Sci, 11, 1279-1294.

Réveillet, M., Six, D., Vincent, C., Rabatel, A., Dumont, M., Lafaysse, M., Morin, S., Vionnet, V., and Litt, M.: Relative performance of empirical and physical models in as- sessing the seasonal and annual glacier surface mass balance of Saint-Sorlin Glacier (French Alps), The Cryosphere, 12, 1367–1386, https://doi.org/10.5194/tc-12-1367-2018, 2018.

Schirle, C. E., S. J. Cooper, M. A. Wolff, C. Pettersen, N. B. Wood, T. S. L'Ecuyer, T. Ilmo, and K. Nygård, 2019: Estimation of Snowfall Properties at a Mountainous Site in Norway Using Combined Radar and In Situ Microphysical Observations. J. Appl. Meteor. Climatol., 58, 1337–1352, https://doi.org/10.1175/JAMC-D-18-0281.1.

Tuzet, F., Dumont, M., Lafaysse, M., Picard, G., Arnaud, L., Voisin, D., Lejeune, Y., Charrois, L., Nabat, P., and Morin, S.: A multilayer physically based snowpack model simulating direct and indirect radiative impacts of light-absorbing impurities in snow, The Cryosphere, 11, 2633–2653, https://doi.org/10.5194/tc-11-2633-2017, 2017.

Wever, N., Schmid, L., Heilig, A., Eisen, O., Fierz, C., & Lehning, M. (2015). Verifica- tion of the multi-layer SNOWPACK model with different water transport schemes. The Cryosphere, 9(2), 2655-2707.

Wever, N., Würzer, S., Fierz, C., & Lehning, M. (2016). Simulating ice layer formation under the presence of preferential flow in layered snowpacks. The Cryosphere, 10, 2731-2744. Interactive comment on Hydrol. Earth Syst. Sci. Discuss., https://doi.org/10.5194/hess-2020- 119, 2020.

---

## Author Response (AR1)

**Response to Interactive comment By Emily Collier**

First, we want to thank Emily Collier for the kind review of our manuscript. The comments and suggestions have helped improve our paper. We have tried as best we can to respond to the comments and we have followed most of the suggestions. The original reviews are in black, and our responses are in blue.

Eidhammer and colleagues present a new coupled modelling tool for atmospheric, glaciological and hydrological simulations, where they have integrated the snowpack model Crocus into WRF-Hydro. The authors apply the model to a multi-year, very-high- resolution simulation of Hardangerjøkulen and evaluate its performance with respect to a variety of observations. The integration of Crocus provides an important improvement in the representation of glaciers compared with the Noah-MP land surface model that will increase the utility and reliability of WRF for simulations of glacierized regions. While a small number of previous efforts have been made to improve glacier physics in WRF, this work includes a novel linkage to detailed hydrological processes and a thorough evaluation over a multi-year time period. The manuscript is well and concisely written, and I recommend its publication in HESS after minor revisions.

Minor comments:

1. The introduction inadequately contextualizes the authors' work with regards to our previous efforts to improve the representation of glacier physics in WRF and their ap- plications. In addition to Collier et al. (2013), there are two more relevant references:

- Collier, E., Maussion, F., Nicholson, L. I., Mölg, T., Immerzeel, W. W., and Bush, A. B. G.: Impact of debris cover on glacier ablation and atmosphere–glacier feedbacks in the Karakoram, The Cryosphere, 9, 1617–1632, https://doi.org/10.5194/tc-9-1617- 2015, 2015.

- Aas, K. S., Dunse, T., Collier, E., Schuler, T. V., Berntsen, T. K., Kohler, J., and Luks, B.: The climatic mass balance of Svalbard glaciers: a 10-year simulation with a coupled atmosphere–glacier mass balance model, The Cryosphere, 10, 1089–1104, https://doi.org/10.5194/tc-10-1089-2016, 2016.

Thank you for advising us on these two more papers. We have included them into the paper.

2. The authors state that glacier ice in Noah-MP cannot melt several times (Lines 134, 303, 397, 445), however my understanding of this LSM's treatment of glaciers is that the subsurface at glacierized grid points is defined as a fully saturated and initially frozen soil. This "soil ice" can and does melt, sometimes entirely. If my understanding is correct, does this treatment differ in WRF-Hydro, or is drainage of glacier melt not accounted for in the hydrological part of the model?

In Noah-MP the glacier is represented with a two-meter layer of ice at the bottom of the column (it is not frozen soil, but 100% water). This layer can melt and refreeze, but it does not run off the grid (so will not contribute to the hydrology). This is true both for regular WRF and in WRF-Hydro.

*We now state this in the paper: When snow is accumulated, Noah-MP uses a three-layer snow model to represent the evolution of the snow pack. However, when the seasonal accumulated snow melts off in the summer, the underlying surface for albedo purposes is assumed to be old snow (snow packed glacier), while not allowing for areas of bare ice. Furthermore, the glacier is also represented in the soil layer with a two-meter layer of ice/water at the bottom of the column. This layer can melt, and refreeze, but this layer does not provide runoff to WRF-Hydro. "*

3. The authors provide relatively few details about the WRF simulations and could consider adding a table with basic information (e.g., grid dimensions, timesteps, physics options, any special settings) to increase the reproducibility of their study. On a related note, was WRF-Hydro/Glacier run with or without a PBL scheme?

We have added a table with the settings used in the WRF simulations (this was also suggested by the other reviewer). The WRF-Hydro part was run without a PBL scheme since WRF-Hydro was not directly coupled with the atmosphere.

4. Line 172: Could the authors comment on the impact of using a reanalysis with ~80-km grid spacing to directly force the outer WRF domain with 3-km grid spacing?

We followed the procedure following Liu et al (2016, Continental-scale convection-permitting modeling of the current and future climate of North America, *Clim. Dyn*), where it is stated that "Tests showed that one-way nesting WRF, at 4-km grid spacing, with the ~75 km reanalysis was an adequate configuration without the need for a coarse grid that intermediates the ERA-Interim data and the WRF domain. "

Basically, the area of interest must be sufficient large enough for mesoscale spinup. Now, our domain of interest (Domain 2) is perhaps slightly closer to the boundary than what is in Liu et al. However, our model results are quite reasonable, thus we do not think that the jump from ~80 km to 3 km introduce a large issue. We have added a sentence discussing this issue

*We acknowledge that a large step from ~75 km (ERA-I) to 3km in WRF is of concern. However, we follow findings by Liu et al., (2016) where they state: "Tests showed that one-way nesting WRF, at 4-km grid spacing, with the ~75 km reanalysis was an adequate configuration without the need for a coarse grid that intermediates the ERA-Interim data and the WRF domain." What is important is that the area of interest must be sufficient large enough for mesoscale spin up. Our domain of interest (Domain 2) is slightly closer to the boundary than what is in Liu et al., (2016). However, as shown below, the model results are quite reasonable, thus we believe that the jump from ~75 km to 3 km is adequate.*

5. Line 188: Why was the model evaluation performed only for the 1-km domain? It looks like the 100-m domain may contain at least the Finse AWS. If so, I suggest the authors also provide a brief evaluation of near-surface variables from this domain, since these data directly force the glaciological and hydrological components.

We did not run WRF on the 100 m domain. The 1 km output (wind, temperature etc) were interpolated to the 100 m domain (line 185 in original document) so we could force the WRF-Hydro part with wind, temperature from the 1 km domain. The other reviewer had some questions about interpolating from 1km to 100 m, and you can see the response in the reply for reviewer #2

6. Line 219: The manuscript has quite a few figures. I think the authors could remove Figure 6 and provide the R2 and mean bias in the text. Although simulated wind direction is evaluated, biases and their implications for the results are not discussed elsewhere, so Figure 7 may also be unnecessary.

Thanks for the suggestion. We did consider remove these figures, but decided to keep these figures. We have included a section that describe the importance of adding sublimation due to wind drift of snow, and therefore believe that the wind rose should remain in the manuscript.

7. Section 3.1: I suggest moving the model evaluation to the results section. In addition, please describe issues with the measurements and missing data (e.g., Lines 307-312, missing data at Finse visible in Figure 5) in the methods.

Agree. We have moved the evaluation section to the result section and we have addressed the missing data. We have included these sentences:

*Note that during September to 17 to September 26, the Finse station did not provide any data (Figures 5f, h and j). However, during this time period, WRF did not predict any precipitation, and Fet did not observe any precipitation. Thus, the cumulative precipitation shown in Figure 5f is still valid.*

8. Line 247: Could the authors provide the dates they used for calculating climatic mass balance in the text or a table? How do the results compare when using the same dates as the observations?

For the end of the winter season, we used the end dates when observed. For the end of the summer season, we used the time when the SWE was the lowest. The reason for this is that the observations might be taken when snow has already started to accumulate and observations are adjusted for this. When we use the same date as observations, the comparison is worse. We have clarified why we do not use the observed date for the summer balance.

*Often summer balance observations are conducted when new snow has accumulated on the upper part of the glacier. However, the summer surface can be identified in shallow snow pits. Therefore, to determine mass balance from the WRF-Hydro/Glacier simulations, we use the date with the smallest simulated glacier mass to determine the end of the summer and start of the winter season instead of using the actual date for when observations were gathered.*

9. Line 293: Could the authors discuss why the winter balance simulated by Noah-MP has, in general, a smaller bias at higher elevations?

Currently we do not know the exact reason for the higher bias with Crocus and we added this sentence:

*The reason for the slightly larger bias for Crocus is not known at this point.*

10. Line 352: How were these two locations selected for comparison with MODIS?

We picked one location representative of the ablation area (Northwest location) and one location representative of accumulation area (Top of the glacier location). Furthermore, we used 9 model grid points.

We added this sentence:

*To investigate different regions of the glacier (accumulation versus ablation area), we picked two different locations of the glacier.*

11. Line 387: What do the authors mean by "lack of groundwater in these specific WRF-Hydro/Glacier simulations"?

We did not run with the groundwater module on for these simulations.

12. Line 401-403: Why was Crocus not used to simulate the 14.7% glacierized area in Finseelvi?

In hindsight we could have done it. In the initial and boundary files we only added the specific glacier area that is needed to run Crocus over Hardangerjøkulen. We should have added the 14.7% glarier area too.

13. Line 413: Where can the reader see that the streamflow significantly diverged? 14. Line 422: Please elaborate on model calibration in the methods section.

Figure 15 shows that the streamflow starts to diverge more in mid-July 2018. We added the date and reference to Figure 15 in the text. We have also noted in the methods section that WRF-Hydro is not calibrated.

Technical comments:

Line 50: Please add "e.g.," to the list of citations.

Done

Line 212 "time period"

Done

Line 213 "do not"

Done

Line 216: "were captured"

Done

Line 236: Remove "surface" or change to "glacier surface mass balance"

Done

Line 238: What does "(nve.no/hydrologi/bre)" mean?

This should have been https://www.nve.no/hydrologi/bre. However, we changed to a more proper citation and use Andreassen et al., 2020 instead.

Line 285: Please indicate which locations were used for measuring the summer mass balance.

Done

Line 290: "redistribution of snow"?

Done

Line 309: "stakes"

Done?

Lines 323 to 325: I suggest removing "slightly" since differences reach 20+%.

Done

Figure 1: Please add a spatial scale.

The spatial scale will be added in the final figure. Currently there are some issues with the ArcGIS license. This should be fixe before we submit final figures.

Figure 2, bottom panel: It would be helpful to add the location of Finse, so that it's clearer where the station is relative to the study glacier.

Done

**Response to Interactive comment by Anonymous Referee #2**

First, we want to thank the Referee for the review of our manuscript. The excellent comments and suggestions have greatly helped improve our paper. We have tried as best we can to respond to the comments and we have followed most of the suggestions. The original reviews are in black, and our responses are in blue.

This paper presents the development of a new glacier component of the WRF-Hyrdo simulation platform. It relies on the multi-layer snowpack scheme Crocus that has been implemented in WRF-Hydro. Crocus is used to simulate in WRF-Hydro/Glacier the continuous evolution of snow and ice layers at pre-identified glacier points. In this paper, WRF-Hydro/Glacier is evaluated for the Hardangerjøkulen ice cap in Nor- way. Model outputs are compared to a large set of observations: (i) measurements of winter, summer and net glacier mass balance, (ii) snow depth measurements from a Ground Penetrating Radar (GPR), (iii) albedo measurements from MODIS and (iv) discharge measurements at two locations. The evaluation revealed improved performances compared to the default version of WRF-Hydro. In particular, the evolution of surface albedo is better represented during the ablation season leading to better estimation of summer mass balance and improved discharge estimations for a partially glacierized catchment.

The new modelling system described in this study presents a large interest for the mountain hydrology community and constitutes an important improvement for WRF- Hydro. My main comments about the study concern (i) the downscaling of meteorolog- ical variables in WRF-Hydro, (ii) the evaluation of winter precipitation, (iii) the compar- ison between simulated and observer glacier mass balance and (iv) the impact of the parameterization that represent the effect of wind-induced snow transport on snow- pack properties in Crocus. These questions need to be clarified prior to publication in HESS. They are listed below as general comments followed by more specific and technical comments.

General comments

1. In this study, the authors tested an offline configuration of WRF-Hydro/Glacier at 100-m resolution. This configuration is driven by an atmospheric forcing obtained with the WRF atmospheric model running at 1 km resolution. The downscaling from 1 km to 100 m corresponds to a simple bilinear interpolation as explained at L 183-185. No correction as a function of the elevation difference between the 1km grid and the 100 m grid is applied for example for temperature. Effect of slopes, aspects and shadowing on incoming shortwave radiation are not taken into account as well and wind speed is not corrected as a function of local topography. This leads to a "smooth" atmospheric forcing at 100-m resolution and ultimately to snowpack simulations that cannot capture the variability of snow accumulation and melt over the glacier as illustrated on Fig. 10 and 11. This absence of small scale variability is not only explained by non-simulated lateral redistribution processes in Crocus. Therefore, the absence of appropriate meteorological downscaling to sub-kilometre resolution in WRF-Hydro/Glacier should be at least discussed by the authors in terms of impact on simulated glacier mass

It is true that we are missing some small-scale variability. However, it should be noted that the area of interest is on a plateau with large open areas (see picture below). Yes, there are some shading, which is mainly prevalent at very low sun angles.  We therefore believe that the bilinear interpolation from 1 km to 100 m is not of a too large concern for this specific case. In a rugged terrain, such as in the Alps, the bilinear interpolation might be a larger problem. The slope of the glacier is not that steep either, with and increase with about 100 m per km.

We added this text in the manuscript

*We note that we did not account for variability in terrain in the re-gridding process. Thus the atmospheric forcing is still "smooth" as regards to a 100 m grid. However, the region of interest (Hardangerjøkulen and surrounding*

*terrain) is an open mostly flat area and we therefore believe that for this specific case, disregarding the variation in terrain does not have much impact on mass balance calculations.*

[Figure]

2. In the paper, the evaluation of WRF winter precipitation at high-altitude exposed stations is influenced by a large wind-undercatch impacting the measurements of winter precipitation at these stations. This is well illustrated on Fig. 5. Station data were not corrected for wind undercatch by met.no. As mentioned by the author, this limits the relevance of the comparison between model output and observations. However, winter precipitations are a key component of the glacier winter mass balance and it would be very interesting to propose in the paper an improved evaluation of WRF winter precipitation at high-altitude stations. Since wind speed measurements are available (at least at Finse), it would be very interesting if the authors could propose their own corrections of wind-undercatch using correction functions taken from the WMO SPICE project (https://www.wmo.int/pages/prog/www/IMOP/intercomparisons/SPICE/SPICE.html) and detailed information on the precipitation gauge used at Finse and at the other locations (type of gauge, type of shield, . . .). They could also quantify the uncertainties associated with these corrections. Another solution would be to use the reference precipitation data from the Haukeliseter experimental site who was the Norwegian site that contributed to the WMO SPICE project. In particular, Haukeliseter is equipped with a Double Fence precipitation gauge. High-quality precipitation data were collected at Haukeliseter during winter 2016/2017 (Schirle et al., 2019). They may be also available for the other winters covered by this study. Haukeliseter is located south of the Hardangervidda and east of Roldal and must be located within the WRF 1-km domain based on Fig 3.

Indeed, Haukeliseter is located in the WRF 1-km domain and it is a great suggestion to use data from this location to evaluate modeled winter precipitation. We have obtained the DFAR observations from 2015 trough 2017. The DFAR data is used as the "truth" and several correction functions has been created based on these data (Smith et al 2020). Thus we compared the WRF 1km simulation with these observations to further evaluate WRF. However, as can be seen in the figures below, WRF still shows higher precipitation during most of the observation periods. Black lines indicate time periods where wind speed is above 10 m/s and grey lines indicates where the temperature is below zero degree Celsius. Typically, the model shows about 30-40% (but also up to 100% at the highest) more precipitation compared to the observations. Note that most of these observations are from the winter season. When looking at stations that are not affected much by snow, WRF typically underestimate some (up to about 20 %, see figure 4 in original manuscript). Thus, even with the DFAR observations (if DFAR is be taken as the truth), WRF overestimate precipitation on Haukeliseter by a large margin and this is opposite to locations that are not affected much by snow. It is therefore difficult to even use the DFAR observations to evaluate the WRF simulation.

Although the SPICE project has allowed for suggested transfer functions to correct for the under-catch we note this finding in Smith et al 2020 "Evaluation of the WMO-SPICE transfer functions for adjusting the wind bias in solid precipitation measurements" Hess that they state:

"Although the application of transfer functions is necessary to mitigate wind bias in solid precipitation measurements, especially at windy sites and for unshielded gauges, the inconsistency in the performance metrics among sites suggests that the functions be applied with caution."

Note that the Fet station is about 15 km from Rembedalskåka while the Finse station is about 12 km away. Instead of using any transfer-functions, we are adding more emphasis on the well comparison between the model and the observations from Fet, since Fet is also fairly close to Rembedalskåka.

We have added this paragraph in the manuscript:

*The World Meteorology Organization (WMO) Solid Precipitation Intercomparison Experiment (SPICE) was set up to evaluate the under-catch of snow and develop transfer functions to correct for the under-catch of solid precipitation. (Smith et al., 2020). One location for these studies is Haukeliseter, which is about 20 km from Røldal (see Figure 2) and is within the 1 km domain. In these studies, several different precipitation gauges and wind shield combinations were used. The Double Fence Automated Reference (DFAR) was deployed as the reference and is used as the "truth" precipitation. We compared the WRF model results with the DFAR data (Smith et al. 2019), and WRF is still predicting more precipitation compared to these observations, with bias typically at 40%, but also as high as 100% (not shown). This is opposite and much higher compared to what is found at locations with little impact of snow. Furthermore, Smith et al (2020) stated this is their study: "Although the application of transfer functions is necessary to mitigate wind bias in solid precipitation measurements, especially at windy sites and for unshielded gauges, the inconsistency in the performance metrics among sites suggests that the functions be applied with caution." We are therefore not adjusting the observed observations on Finse for our evaluation, and rather stress the well comparison between model and observations at Fet and summer season precipitation on at Finse.*

[Figure]

[Figure]

[Figure]

[Figure]

Figure 1 Observed and modeled precipitation at Haukeliseter

3. The authors compared on Fig. 8 the simulated and observed winter, summer and net mass balance. However, they did not clearly explain in the current version of the paper (L 284-289) how the simulated winter and summer balances were computed. For the summer mass balance, did the authors extract the simulation results at the 3 to 5 locations used to compute the observed mass balance and then interpolated the results? The same question raises for the simulated winter mass balance. In addition, on Fig. 8, the elevation dependency of the observed winter mass balance (Fig 8a, d, g, j) and snow depth associated with the winter mass balance (Fig 8b, e, h, k) look different. However, the authors explained at L 241-243 that the observed winter mass balance was derived from snow depth

and the unique snow density measurements over the glacier. As a consequence, we would expect a similar elevation dependency between winter mass balance and snow depth. However, for example, in 2015 the observed snow depth showed a decrease between 1400 and 1600 m which is not present in the observed winter mass balance. This point should be clarified by the authors.

For the modeled winter and summer balance we take the average balance for each grid point within a certain elevation level (40 m)

We have added this sentence:

*The modeled winter (and summer) balance is plotted as averages of all grid points over Rembesdalskåka within intervals of 40 m*

Regarding the difference in plottet SWE versus height, the SWE values were taken from the values shared in the reports by Kjøllmoen et al., (2016, 2017, 2018 and 2019) (see Figure 1) while the snowdepth was taken from the original data. One of the authors of these reports stated that the data in the reports were an arithmetic mean of height and SWE withing 50 m as a base for subjective smoothed SWE curves. We are now using only the reported values from the Reports the Figure.

[Figure]

**Figure 6-11**
Specific (left) and volume (right) winter, summer and annual balance at Rembesdalskåka in 2015.
Specific summer balance at five stakes is shown (o).

**Figure 2.** Example of SWE versus height from the Reports by Kjøllmoen et al., (2016, 2017, 2018 and 2019)

4. In the configuration of Crocus used by the authors in this study, two parameterizations are activated to represent the effect of wind-induced snow transport on snowpack properties: (i) a param. that simulates snow compaction and fragmentation of snow grains for surface layers during blowing snow events and (ii) a param. that computes mass loss due to blowing snow sublimation. At L 164-166, the authors insisted on the importance of these two parameterizations for accurate simulations of glacier mass balance. However, in the rest of the paper, the effects of these two parameterizations are never quantified. For example, how does the compaction parameterization affect the quality of the simulated snow density over the glacier? In addition, the authors suggest that the blowing sublimation parameterization in Crocus explain the spatial variability in snow depth and SWE over the glacier (L 315-320). However, it is not clear that it can explain the local differences in snow depth and SWE. Indeed, the atmospheric forcing driving Crocus in WRH-Hydro/Glacier is rather smooth (cf General comment 1) and may not create a large variability of blowing snow sublimation from one grid cell to another on the 100-m grid. I recommend the authors to

compare the results of simula- tions with and without the blowing sublimation parameterization. One winter would be certainly sufficient.

We actually did include a section describing the sublimation of blowing snow issue and had a figure. However, to reduce the paper and number of figures, the editor suggested we initially remove this section. We have included a paragraph in section 5, showing the result without sublimation.

The modeled wind did have some variations over the glacier on the large scale with the north east part of the glacier experiencing the highest windspeeds (Figure 3). This is also where you have the largest differences in mass balance (see figure 3 where the left plot show without sublimation due to wind drift and the right shows with).

[Figure]

**Figure 3** Average wind speed winter 2017

[Figure]

**Figure 4** Modeled versus observed snow thickness. Left plot shows without sublimation of blowing snow and right plot shows with sublimation of blowing snow.

New paragraph:

*In section 2.1 we mentioned the importance of adding sublimation of blowing snow in our simulations. Figure 21 shows the snow thickness and the respective scatterplot for when the sublimation of blowing snow is not included (which is the default in the SURFEX V8.0 setup (as downloaded)). As can be seen, the simulated snow thickness is slightly higher than the observations with the GPR, and this is especially true at the eastern part of Hardangerjøkulen. During the 2017 winter season, this region had on average the strongest winds, causing more sublimation from snowdrift than at other locations on the glacier. Without including sublimation of blowing snow, the simulation overestimates snow thickness. However, for Rembesdalskåka, the overall winter balance increases when excluding the sublimation due to blowing snow and compares slightly better with observations (not shown). The resulting streamflow from turning of sublimation of blowing snow is about a 4% increase (not shown).*

Specific comments

Abstract L 16: The transition between the first and the second sentence of the ab- stract is not clear at the moment. It would be interesting to add here a sentence that explains why Crocus is suitable for glacier modelling. The fact that a multi-layer snowpack scheme can be used for direct surface mass balance simulation of glacier is not necessarily clear for a reader who is not familiar with Crocus.

We have moved the part of the abstract that describes why Crocus is suitable for glacier model further up in the abstract.

Abstract L 17-18: The two different resolutions for WRH-Hdyro and WRF-Hydro/Glacier are rather confusing. Maybe mention atmospheric simulation on one hand and offline surface simulations on the other hand.

The sentence now reads:

*WRF atmospheric model simulations were downscaled to 1 km grid spacing to provide meteorological forcing data to the WRF-Hydro/Glacier system at 100 m grid spacing for surface simulation.*

Abstract L 19-20: A sentence is missing in the abstract to explain that the study is carried out over a glacier in southern Norway. It is only mentioned in the title.

The sentence now reads:

*To evaluate the new system (WRF-Hydro/Glacier) over a glacier in Southern Norway,*

P2 L 38-40: I recommend the authors to add one or two citations for this sentence.

We added the citation of Ayala, A, Pellicciotti, F, and Shea, JM (2015), Modeling 2 m air temperatures over mountain glaciers: Exploring the influence of katabatic cooling and external warming. J. Geophys. Res. Atmos., 120, 3139– 3157. https://doi.org/10.1002/2015JD023137, and Liston, G., & Sturm, M. (1998). A snow-transport model for complex terrain. Journal of Glaciology, 44(148), 498-516. doi:10.3189/S0022143000002021

P2 L 48-49: the authors should add relevant citations for the application of statistical downscaling for glacier studies. In addition, physically based downscaling methods have also been developed to obtain better distributed meteorological forcing in regions of complex terrain (e.g., Jarosch et al., 2012; Fiddes et al., 2014).

The papers by Machguth et al., 2009; Kotlarski et al., 2010; Van Pelt et al., 2012 already describe the statistical downscaling they are conducting; thus we do not think there is a need to add additional citations for statistical downscaling for glacier studies.

We added this sentence in the manuscript. *Physical based downscaling based on the linear model by Smith and Barstad (2004) has also been applied over complex terrain for glacier studies (e.g. Karosch et al., 2012)*

P2 L 54-55: I agree with authors that regional "atmosphere-only" models do not typically include a detailed representation of glacier and their impact on streamflow generation. However, atmospheric forcing from these regional models are often used to drive more detailed models such snowpack models or hydrological models in offline mode for impact studies. I recommend to the authors to mention this approach in the introduction.

We agree, and we added this sentence:

*though these models provide input to detailed offline snowpack and hydrological models.*

P 2 L 59: The authors should better explain this "link" between an atmospheric model and a detailed hydrological model. Indeed, are they talking about offline simulations or online simulations with feedbacks of the surface on the atmosphere dynamics? When used in offline mode as it is the case in this study, WRH-Hydro does not really differ from a more classic hydrological model such as MESH (Pietroniro et al., 2007) that takes its atmospheric forcing from an external system (GCM, RCM, . . ..), downscale them and use them to drive hydrological simulations.

This is correct, and we have reformulated:

*In the studies where, we will use detailed hydrological model - the Weather and Research Forecasting - Hydro (WRF-Hydro) modelling system (Gochis et al., 2015; Senatore et al., 2015; Arnault et al., 2018; Fersch et al., 2019; Rummler et al., 2019) for streamflow modeling.*

P3 L 75 and P4 L 103: Crocus has a user-defined value for the maximal number of snow layers. The value of 50 corresponds to the default value in the model. I recommend the authors to clearly explain it. Especially since they then use 40 layers in their implementation of Crocus in WRF-Hydro.

Sentences now read:

*It is a physically-based model, in which the snow depth can be divided into a user defined maximum levels and where the default maximum is 50 layers.*

*In the Crocus model, it is possible to divide the snow into a user defined maximum numbers of dynamically evolving layers.*

P 3 L 76-78: Note that the default version of Crocus still uses a classic bucket-type approach for liquid water percolation in the snowpack. It does not solve Richards equations and ignore preferential flows contrary to the SNOWPACK model (Wever et al., 2015, 2016). Preliminary developments have been tested in Crocus but are not available in the version of Crocus used in this paper (D'Amboise et al., 2017).

Thanks for this comment. We removed the statement of retention since Crocus still uses a classic bucket-type approach.

P 3 L 78-79: the reference to the paper by Reveillet et al. (2018) is missing. I also recommend the authors to refer to the work of Gerbaux et al (2005) which used the original version of Crocus.

Thank you for drawing attention to Reveillet et al. (2018). We have added this paper to our citations:

*The Crocus model was first used for glacier mass balance by Gerbaux et al (2005) and recently used for glacier surface mass balance studies within the French Surfex model by Reveillet et al. (2018), Revuelto et al. (2018) and Vionnet et al. (2019).*

P4 L 98-99: This sentence is not clear and should be rephrased. Indeed, the "age of snow" is not directly used in the prognostic equations for the time evolution of microstructural variables In Crocus. It is used in the albedo parameterization to compute the decay of albedo in the UV and visible band and will indirectly impact the evolution of microstructural variables. Compaction is not also directly impacting the evolution of microstructural variables.

Thank you for the clarification. We have removed "age of the snow"

P4 L 100-101: A description of the distinction between snow and ice albedo is missing in this paragraph. It would be interesting if the authors could use a similar description as Reveillet et al. (2018) (see Sect. 3.1.1 of this paper). The authors should also mention how the aerodynamic roughness are treated for snow and ice surfaces.

This is a good suggestion and we have added some more information regarding the distinction between ice and snow albedo and the values for snow and ice roughness. For ice albedo we ended up using the values used in Surfex (and not the ones from Gerbaux et al, as used in Reveillet.

We now state this in the manuscript (italic shows added text):

Furthermore, the snow albedo is calculated based on the snow grain properties from the top 3 cm of the snowpack (Vionnet et al., 2012) and is calculated in three spectral bands (0.3-0.8, 0.8-1.5 and 1.5-2.5 µm). *Impurities in aging snow is parameterized inn the visible spectral band (0.3-0.8 µm) from the age of the snow with a time constant of 60 days. See Vionnet et al., (2012) for detailed description of the albedo calculations. The albedo over ice is constant in all spectral bands and are 0.38, 0.23 and 0.08 for the spectral bands 0.3-0.8, 0.8-1.5 and 1.5-2.5 µm. The sensible*

*and latent heat are parameterized with an effective roughness length over snow and ice (see Vionnet et al., (2012) for further details). Here we use 1 mm over snow and 100 mm over ice.*

P 4 L 101-102: the justification "Due to the prognostic calculation of snow grain properties, . . ." is not clear and should be rephrased.

This sentence is removed

P4 L 106-108: did the authors consider using Crocus to represent snow over land in NOAH-MP?

Currently the fluxes between the glacier surface and the ground is not incorporated (the engineering of passing fluxes in SURFEX compared to Noah-MP is somewhat different and is not yet incorporated). Under glacier surface it is assumed that the temperature is the same as soil and therefore there are no fluxes added. We would like to incorporate the fluxes between Crocus and the soil soon to be able to use Crocus over the entire domain.

P4 L 109-111: Are the authors imposing a constant-temperature of 0 degC for the ground below the glacier or are they imposing a zero-heat flux between the deepest Crocus layer and the ground below as in Gerbaux et al. (2005)?

We are imposing a constant-temperature of 0 degC, as the heat-fluxes between the soil and surface is not yet incorporated. We removed the statement that there are no fluxes of heat pass trough and instead state that no fluxes have been incorporated between the glacier and the ground below.

P5 L 126-137: It would be useful to have this paragraph at the beginning of Section 2 so that the reader could better understand how glaciers are represented in the default version of WRF-Hydro and what are the associated challenges. This would provide a very relevant justification for the use of Crocus over glaciers.

Agree. We have moved the paragraph closer to the beginning of Section 2.

P 5 L 149-150: initializing all the layers with pure ice may influence the accuracy of the snowpack simulations when snow starts to fall on the glacier. Indeed, as mentioned by the authors, it forces the model to merge layers as soon as new snow is added. Instead, the initialization proposed by Revuelto et al. (2018) used 6 initial layers to represent the ice and a maximal number of layers set to 50 so that new snow layers can be created as soon as snow is falling on the glacier without forcing the immediate merging of ice layers underneath. This difference should be mentioned in the paper and its impact briefly discussed.

The reviewer is correct, and in future studies the glacier should be initialized in less layers than the maximum layers. Note, we allow 2 months for spin-up time before using October 1$^{st}$ for calculating winter balance. At this time the glacier has started to merge. By January 1 the glacier is merged down to about 6-8 layers and remain as such for the rest of the simulation. Will add a short discussion regarding this point in the paper.

P 5 L 150: what are the snow grain properties for ice layers used in the paper?

We use the Brun 92 scheme. We are not sure where to explicitly find the grain properties for ice layers in the code, but Snowgrain1=99 and Snowgrain2 = 0.003415 in our output files.

P 6 L 162-167: I recommend the authors to move this paragraph at the end of Section 2.1 when they are mentioning the absence of lateral snow redistribution due to wind in Crocus. This would help the reader to better understand how WRH-Hydro/Glacier accounts for the impact of wind-induced snow transport.

We have followed the suggestion and moved the paragraph.

P 6 L 167: it would be interesting to add here a brief description of the configuration of the routing model. Is the routing simulated at the same resolution as Crocus (100 m)? How were derived the routing parameters? It is also important to mention here that no calibration was applied to the routing model.

We have added this to the manuscript:

*The routing is run at the same resolution as Crocus (i.e. at 100m). And no calibrations were applied to the routing model.*

But instead of adding this in section 2.1, Crocus initialization, we added it to section 3, Experimental Description.

P 6 L 170: a table summarizing information about the simulation domains (size, number of grid points, ....) would be useful. This table could also include information on the 100-m simulation domain.

This was suggested by the other reviewer as well, and we have added a table summarizing the simulations

P6 L 171: what is the height above the ground of the lowest atmospheric prognostic level?

The lowest model level height is about 25 m. We have added this information in the text.

P 7 L 183: how is computed the rain/snow partitioning in WRF-Hydro/Glacier?

The rain/snow partitioning in WRF-Hydro/Glacier is the same as one of the Noah-MP options, which is called the Jordan (1991) scheme. Snowfraction (SF) = 0 above 2.5C.  SF = 0.6 between 2 and 2.5C. SF = 1 below 0.5C. Between 0.5 and 2C, SF is a linear function. We added a comment and reference on which scheme is used

Jordan, R. (1991), A one-dimensional temperature model for a snow cover, *Spec. Rep. 91–16*, Cold Reg. Res. and Eng. Lab., U.S. Army Corps of Eng., Hanover, N. H.

P7 L 188: which method it used to obtain the simulated values at the location of the AWS from the output of WRF 1 km? Nearest-neighbor interpolation, bilinear interpolaion? Did they author consider the elevation difference between the simulated station elevation and the actual station elevation when selecting the stations used for model evaluation?

Initially we used nearest neighbor and did not consider differences in elevation. After investigating the differences between model elevation and actual elevation, we decided to take the four closest grid points that also were closest to the actual elevation. The average elevation of these four locations moved closer to the actual elevation than where only using nearest grid point. However, the results did not change much in Figure 4 and conclusions still stay. We did however remove any stations where the difference in elevation were more than 100 m in the new figure 4. Note that the model elevations of the two closest stations to Hardangerjøkulen (Finse and Fet) are only respectively 12 and 6 m higher than the actual elevation.

We added this sentence:

*The values shown in Figure 4 are obtained from finding the closest grid point to the actual location, then from there take the 4 closest model grid points relative to the selected grid point closest to actual AWS elevation.*

P8 L 216-217: At this stage of the paper, it is not clear that the wind direction and the wind speed are well simulated by WRF. This information is mainly confirmed by Figure 7 which is presented later.

In these lines, we were discussing the wind speed and direction during the specific underpredicted precipitation event. We have clarified this in the manuscript:

*We note that the observed storm sequences were captured in the simulation, wind direction was well simulated as well as the wind speed during this precipitation event (not shown), just not the precipitation amount.*

P 9 L 239: at how many locations is measured the mass balance of the glaciers?

For the winter balance, all the green dots in Figure 1 is used (about 60 locations). For the Summer balance, there are 4 locations. We have clarified this in the manuscript.

P 9 L 247-249: it is not clear why the authors did not use the actual dates when the observations were gathered to compute the simulated mass balance, contrary to Vionnet et al. (2019). Could they add a justification?

For the winter balance we did. For the summer balance we did not because the observations are adjusted for summer surface, not when observations are conducted. We now state this in the text:

*Often summer balance observations are conducted when new snow has accumulated on the upper part of the glacier. However, the summer surface can be identified in shallow snow pits. Therefore, to determine mass balance from the WRF-Hydro/Glacier simulations, we use the date with the smallest simulated glacier mass to determine the end of the summer and start of the winter season instead of using the actual date for when observations were gathered.*

P 9 L 256-257: it is not clear to me how a device transported at 15-20 km/h (approx. 4.1-5.5 m/s) with a sampling interval of 1 s can generate a 1-m spacing between data-points. Note that I am not familiar with GPR postprocessing, so ignore my comment if the 1-m spacing is obtained from data postprocessing.

Thanks for point out this error.  We checked our survey coordinates, and there is indeed a GPR trace every ~1 m; it is instead the GPS system which records a positional datapoint every 1 s.  The wording has been corrected in the manuscript:

*The GPR systems were towed behind a snowmobile, at ~15-20 km/h. The interval between successive GPR recordings is ~0.2 s, giving a distance sampling interval of ~ 1 m (regularized to exactly 1 m in processing). A GPS system was also mounted on the snowmobile, recording positions every 1-2 s, to locate the GPR recordings*

P 10 L 273: snow albedo in the version of Crocus used in this paper also depends on the snow age. Indeed, the snow age is used to parameterize the influence of light absorbing impurities on snow albedo in the UV and visible range. A more recent version of Crocus explicitly simulates the direct and indirect radiative impacts of light-absorbing impurities in snow (Tuzet et al., 2017).

We added the information regarding parameterization of light absorbing impurities in the manuscript (see response above)

P10 L 277: what is the size of the two river catchments considered in the study?

The catchment size of Fineseelvi is about 16 km$^2$ and Middalselvi is about 12 km$^2$. We have added this information in the manuscript.

P 12 L 335-347: Figure 10 shows that the GPR provided an excellent coverage of the glacier for winter 2017. It would be very interesting to use these data to compare the simulated and observed elevation-snow depth relation and to compare the simulated and observed variability of snow depth per elevation bands. This would complement well Fig. 8.

We added plots of model and GPR snow depth in Figure 11

P 12 L 349: the evaluation of the simulated albedo is based on a comparison at two representative points selected over the glacier. This choice should be justified in the method section when the authors are describing the MODIS albedo data. Indeed, at L 273-276 the authors mention that MODIS data are available at 500 m resolution over the glacier. This suggest that the full MODIS albedo dataset would be used for model evaluation.

Yes, we agree and we could have used the entire MODIS dataset. We added this sentence:

*To investigate different regions of the glacier (accumulation versus ablation area), we picked two different locations of the glacier.*

P 15 L 440: I am not sure that the term "excellent" can be used to qualify the WRF winter precipitation. Indeed, the authors have shown that they cannot be directly evaluated due to wind undercatch at the high-elevation stations which are the most relevant for this study. Maybe the authors should make here the link between the WRF winter precipitation and the winter mass balance simulated by WRF-Hydro/Glacier.

We changed excellent to great. We also added this sentence:

*This can also be seen in the generally good agreement of winter mass balance.*

P 16 L 454-455: the author should mention here that the model at 100-m resolution cannot capture the spatial variability of the snowpack on the glacier due (i) the absence of proper meteorological downscaling and (ii) the non-representation of lateral snow redistribution. It is mentioned later in the conclusion as a perspective, but it should appear in the bullet list containing the major conclusions of the study as well.

Regarding the absence of proper meteorological downscaling, since the region is not very complex, we believe for this study, the downscaling is proper over Hardangerjøkulen (ads discussed earlier).

We added this sentence to the bullet discussing the GPR observations. "Some of the bias between observations and model results are also likely a result of not accounting for lateral snow distribution in the model."

P 16 L 467: information on the code availability would be very interesting for the readers. It could potentially serve a basis for Crocus users who want to implement the model in their own land surface model.

Currently we are implementing the Crocus into the National Water Model (a version of WRF-Hydro) and this version is now available on git.

https://github.com/trudeeidhammer/wrf_hydro_nwm_public/tree/trude_crocus_merged

Technical comments
Text
P2 L 35: "ablation-season"? done
P2 L 38: a parenthesis ")" is missing  done

P4 L 94: I recommend the authors to use "initially developed" instead of "specifically developed" Indeed, since the 90's, Crocus has been used in many different applications (https://www.umr-cnrm.fr/spip.php?article268&lang=en). done

P4 L 99: use "metamorphism" instead of "metamorphosis" done
P 4 L 100: Ân´ Vionnet Âz˙ instead of "Vionett". done
P5 L 144: It is surprising to have a Section 2.1 without a Section 2.2. Agreed, and we will move the text before Section 2.1 into a new Section 2.1 and moce Section 2.1 to Section 2.2 We will do the same with Section 3.1 (since there are no section 3.2

P5 L 147: use kg mˆ{-3} instead of kg/m3 done
P 5 L 208: use m sˆ{-1} instead of m/s done
P15 L 427: maybe add "and ice" after "physical properties of snow" done
P21 L 598: the reference to Willemet (2008) is not included in the text. This is removed
Table
Table 1: what is the signification of the values appearing in bold in the table? The bold font represents the case with the highest correlation for each year and each location. We have added this in the text.

Figure 7: different scales are used for the observed and simulated wind data on the wind rose. Using the same scale would allow q more direct comparison between model and observations. In addition, could the authors provide simple errors metrics such as bias and RMSE for wind speed at Finse?

We completely agree. The author tried to make the scale the same, but with the software used, it was difficult. However, we found the solution, and the scales are now same. Mean bias for the model results over the entire period is -0.17 m/s.

References

D'Amboise, C. J. L., Müller, K., Oxarango, L., Morin, S., and Schuler, T. V.: Implemen- tation of a physically based water percolation routine in the Crocus/SURFEX (V7.3) snowpack model, Geosci. Model Dev., 10, 3547–3566, https://doi.org/10.5194/gmd- 10-3547-2017, 2017.

Jarosch, A. H., Anslow, F. S., & Clarke, G. K. (2012). High-resolution precipitation and temperature downscaling for glacier models. Climate Dynamics, 38(1-2), 391-409.

Fiddes, J., & Gruber, S. (2014). TopoSCALE v. 1.0: downscaling gridded climate data in complex terrain. Geoscientific Model Development, 7(1), 387-405.

Pietroniro, A., Fortin, V., Kouwen, N., Neal, C., Turcotte, R., Davison, B., ... & Pellerin, P. (2007). Development of the MESH modelling system for hydrological ensemble fore- casting of the Laurentian Great Lakes at the regional scale. Hydrol. Earth Syst. Sci, 11, 1279-1294.

Réveillet, M., Six, D., Vincent, C., Rabatel, A., Dumont, M., Lafaysse, M., Morin, S., Vionnet, V., and Litt, M.: Relative performance of empirical and physical models in as- sessing the seasonal and annual glacier surface mass balance of Saint-Sorlin Glacier (French Alps), The Cryosphere, 12, 1367–1386, https://doi.org/10.5194/tc-12-1367-2018, 2018.

Schirle, C. E., S. J. Cooper, M. A. Wolff, C. Pettersen, N. B. Wood, T. S. L'Ecuyer, T. Ilmo, and K. Nygård, 2019: Estimation of Snowfall Properties at a Mountainous Site in Norway Using Combined Radar and In Situ Microphysical Observations. J. Appl. Meteor. Climatol., 58, 1337–1352, https://doi.org/10.1175/JAMC-D-18-0281.1.

Tuzet, F., Dumont, M., Lafaysse, M., Picard, G., Arnaud, L., Voisin, D., Lejeune, Y., Charrois, L., Nabat, P., and Morin, S.: A multilayer physically based snowpack model simulating direct and indirect radiative impacts of light-absorbing impurities in snow, The Cryosphere, 11, 2633–2653, https://doi.org/10.5194/tc-11-2633-2017, 2017.

Wever, N., Schmid, L., Heilig, A., Eisen, O., Fierz, C., & Lehning, M. (2015). Verifica- tion of the multi-layer SNOWPACK model with different water transport schemes. The Cryosphere, 9(2), 2655-2707.

Wever, N., Würzer, S., Fierz, C., & Lehning, M. (2016). Simulating ice layer formation under the presence of preferential flow in layered snowpacks. The Cryosphere, 10, 2731-2744. Interactive comment on Hydrol. Earth Syst. Sci. Discuss., https://doi.org/10.5194/hess-2020- 119, 2020.

---

## Author Response (AR2)

Response to Dr Collier

Thank you for the comments and great suggestions. Below in blue are our responses and actions.

Thank you to the authors for their responses and for the revisions, which have improved the manuscript. I support the publication of the revised manuscript subject to a few small clarifications and changes:

1. Lines 43-45: "Therefore, the proper simulation of the non-homogenous, non-stationary evolution of a glacier requires atmospheric processes at much finer resolution than typical global or regional climate models can provide (Collier et al., 2013, Collier et al., 2015, Aas et al., 2016)."
I appreciate that the authors have cited the additional studies. However, only Aas et al. (2016) looked at the resolution requirements of atmospheric forcing fields for adequate glacier simulations, as cited. The important, but still missing, point to contextualize the authors' work is that these studies provided the first efforts to integrate a physically based glacier mass balance model into WRF for improved simulations.

Thank you for pointing this out. We have now added this sentence (underline indicate changes)

"Glacier mass balance parameterizations have been implemented in atmospheric models such as the regional climate model (REMO, Kotlarski et al. 2010b) and a climate mass balance model with feedback to the atmosphere was implemented into WRF by Collier et al. (2013)."

We have also removed the Collier et al citations in the sentence the reviewer pointed to.

2. Line 109: "Furthermore, this exposed glacier ice cannot melt as the glacier is only a land surface category"
Please rephrase to be consistent with the revised lines 74-75.
We added to this sentence: "Furthermore, this exposed glacier ice cannot melt as the glacier is only a land surface category (though the glacier is represented in the soil layer with a two-meter layer of water/ice but does not provide runoff to WRF-Hydro)."

3. Lines 134 to 138: "Importantly, the Crocus model interacts with the atmosphere by providing fluxes between the surface of the glacier and the atmosphere. These fluxes are total absorbed solar radiation, total reflected solar radiation, total net longwave radiation, total sensible heat, evaporation heat flux (and rate) from snow, and ground heat flux. Some diagnostic outputs, such as the 2m temperature and 2m vapor mixing ratio are calculated by the original Noah-MP snow model, but with the snow information (snow surface temperature and albedo) provided by Crocus."

Since the forcing data come from an offline WRF simulation, there is no interaction between the Crocus solution and the atmosphere. This sentence can mislead readers into thinking interactive simulations have been performed between all three [atmospheric-cryospheric-hydrological] components. Please remove or rephrase to indicate that atmospheric interactions are planned for future work.

We agree with this comment, and this paragraph has been removed.

4. Line 442 "This is likely due to lack of using the baseflow/groundwater module in these specific WRF-Hydro/Glacier simulations."
Why was this module not used and what is the potential impact?
The potential impact is that some groundwater/baseflow that could potential contribute to the streamflow is not included in the model. I did not use this option because I was not provided with the necessary input data for running this option. Any new simulation should include this option.

We have added to this sentence: "This is likely due to lack of using the baseflow/groundwater module in these specific WRF-Hydro/Glacier simulations, which could add some water to the surface streamflow."

5. Line 477: "And while we have calibrated some parameters in Crocus, we used all default values in Noah-MP"
Can the authors provide the calibrated parameter values or include the information about finding the code on GitHub in the manuscript, so that the results could be reproduced?

Thank you for this question. In our first model results we had to adjust the roughness length to a rather low value for both snow and ice in order to compare better with observations. These are the results used when writing the initial drafts of the manuscript. However, a bug was found in the code (from the coupling, not in the Crocus code itself). When we reran the simulations, we went back to the original roughness values (but the text about calibration remained in the manuscript). We have therefore removed this paragraph.

6. Figure 13: I suggest merging the relevant panels with Figure 11 and 12, to aid with visual comparison.

This is a good suggestion. However, while merging the figure together, I found the figure to be "overwhelming". I also believe that the two merged figures have the potential to being smaller in the final manuscript, than two individual figures. So I will keep them separate. However, I rearranged the rows and columns, so that the two figures now represent the year and model in the same row/column format.

Response to reviewer #2

Thank you for the comments and great suggestions. Below in blue are our responses and actions.

The authors have carefully revised the manuscript based on the comments raised during the first round of review. Therefore, the manuscript had been made clearer in many respects. I would like to thank the authors for this work. I have listed some comments about their answers followed by more specific and technical comments

Comments about the answers

- WRF-Hydro Glacier at 100-m.
I understand the argumentation about the potential limited impact of meteorological downscaling for glacier mass balance simulation in the context of this study. I still think that it would be good in the conclusion to have a paragraph about the benefits and limitations of the configuration at 100-m resolution compared to a configuration at 1-km. From my understanding, it allows a high-resolution glacier initialization and routing. On the other hand, it does not benefit from any topographic-based meteorological downscaling and does not represent physical processes affecting snowpack evolution at 100-m grid spacing.

We added this sentence in the conclusion
"Finally, the forcing at 1 km does not account for any topographic variations in the 100 m domain, thus snowpack evolution at 100 m scale is not included."

- Evaluation of winter precipitation
The comparison between WRF and the DFAR data from Haukeliseter reveals similar precip. biases to those obtained in Finse and Midstova. However, the DFAR is supposed to be less impacted by wind-undercatch of precipitation than the precip. gauge in Finse and Midstova. The paragraph (P 10 L 294-305) added by the authors suggests that the comparison with the DFAR data has little value and is not really reliable. Does it mean that the authors question the quality of the DFAR data? I think the argumentation could be improved in this paragraph.

This is a very good and interesting question. I took a closer at the DFAR data and the bias, and the bias is lower than initial stated in the manuscript.

Regarding the DFAR, the bias in windy conditions is about 5-10 %. This comes from comparisons with bush-sheltered Tretyakov gauges (e.g. Rasmussen et al. 2012): " The DFIR configuration was extensively compared to a bush-sheltered Tretyakov gauge, considered to be a true representation of snowfall, at the hydrological research station near Valdai, Russia, from 1970 to 1990. Although the large octagonal double fence was shown to catch less snowfall than the bush gauge, the differences were relatively small (<10%) "

This underestimation does not explain the about 20% higher overestimation in WRF. After conversation with other modelers that focus on Norway, it is often seen that in westerly conditions, models at times underestimate precipitation at the coast and therefore produce more precipitation further inland.

We have now rephrased the text so the quality of the DFAR is not questioned:

"We compared the WRF model results with the DFAR data (Smith et al. 2019), and WRF is predicting more precipitation compared to these observations, with a bias typically at ~30% (not show). About 10% of this bias could potentially be attributed to underestimation with the DFAR (Rasmussen et al. 2012). The bias in WRF is opposite and higher compared to what is found at locations with little impact of snow. In regards to transfer functions (correcting for under catch in observations), Smith et al (2020) stated this is their study: "Although the application of transfer functions is necessary to mitigate wind bias in solid precipitation measurements, especially at windy sites and for unshielded gauges, the inconsistency in the performance metrics among sites suggests that the functions be applied with caution." We are therefore not adjusting the observed observations on Finse for our evaluation, and rather stress the well comparison between model and observations at Fet and summer season precipitation on at Finse. "

- Variability of snow accumulation
The new figure 13 added to the manuscript is quite interesting and illustrates well the impact of blowing snow sublimation on the simulated snow depth at the glacier scale. However, it also raises a question which is not answered by the authors. Figure 11 shows a clear difference between Noah-MP and Crocus in terms of pixel-to pixel variability of simulated snow depth in 2017 and 2018. Figure 13 (left) shows that the simulated snow depth also presents this pixel-to pixel variability (alternance of orange and yellow/green colors) when the blowing snow sublimation is not activated. This variability is at sub-kilometer scale. In these experiments, Crocus at 100 m is driven by bilinearly interpolated smooth atmospheric forcing obtained from WRF at 1-km grid spacing. In addition, Crocus does not account for lateral-snow redistribution that could create small-scale variability of snow accumulation. Therefore, the author should better explain which processes are generating this small-scale variability in the model. Does it result from the implementation of Crocus in WRH-Hydro? It seems the blowing snow sublimation is not the explanation. Or maybe, it is just a visual effect of the plotting library.

I agree with this comment. This is an issue that is under investigation and my suspicion is that some error constraints in the model had to be relaxed in the current version when implemented into WRF-Hydro. Though this is something that should be looked into, the overall conclusion in the manuscript does not change.

- Station selection
The authors considered in their new figure 4 the differences between the station elevation in the model and the actual station elevation when selecting the stations used for evaluation.

However, the criteria of 100-m mentioned in the response to the reviewers is not mentioned in the revised manuscript. This should be added.

We have added this sentence to the text: "Furthermore, stations that are located in the model over 100 m above the actual elevation are not included."

Specific comments

P 16 L 477: the author mentions here that some Crocus parameters have been calibrated. Could they list the parameters that has been calibrated and which calibration strategy was used? Maybe it could be briefly described in Sect. 2.1.

Thank you for this question. In our first model results we had to adjust the roughness length to a rather low value for both snow and ice in order to compare better with observations. These are the results used when writing the initial drafts of the manuscript. However, a bug was found in the code (from the coupling, not in the Crocus code itself). When we reran the simulations, we went back to the original roughness values (but the text about calibration remained in the manuscript). We have therefore removed this paragraph.

P 17 L 502-507: I think it is important to mention here that the observed density has been measured at one single point over the glacier. This is a limitation for the comparison between observed and simulated snow density.

We added this sentence: "Also note that the snow density is only measured at one location and assumed to be the same over the entire glacier."

P 25 Table 2: the signification of the values appearing in bold is still not really clear to me. It seems that for some year and some location, the highest correlation value is not written in bold. See for example the edge site in 2015 and the value of 0.90 for Noah-MP with respect to MODIS Terra.
We have removed the bold font as we do not mention this in the text

Technical comments

P11 L 323: space between "-0.13" and "m ..". use "m s^{-1}" instead of "m/s"
Done

P13 L 400: "turning off"
Done

P 24 Table 1: "Domain 2"
Done

P 24 Table 1: "grid points"
Done

---

## Author Response (AR3)

Dear Dr Jim Freer

Thank you for working with us through our submission of our manuscript. We have added this text regarding the pixel-to-pixel variations. Unfortunately, we do not have a solid solution yet, but the issue is being investigated further.

"We need to note that Figures 11 and 13 show pixel-to-pixel variations in the Crocus output. This is not due to variations of atmospheric forcing (which has a 1 km grid spacing compared the 100 m grid spacing of the WRF-Hydro/Glacier simulations) or blowing snow. We suspect the pixel-to-pixel variations arise from small vertical resizing errors of the very thick glacier layers, for where we relaxed some of the test requirements when resizing. This does not change the conclusions in this paper, and work is in progress to address this issue. "